# Lzts1 controls both neuronal delamination and outer radial glial-like cell generation during mammalian cerebral development

T. Kawaue[1,4], A. Shitamukai [2], A. Nagasaka[1,5], Y. Tsunekawa[2], T. Shinoda[1], K. Saito[1], R. Terada[1], M. Bilgic[2,3], T. Miyata [1], F. Matsuzaki [2,3] & A. Kawaguchi [1]

In the developing central nervous system, cell departure from the apical surface is the initial and fundamental step to form the 3D, organized architecture. Both delamination of differentiating cells and repositioning of progenitors to generate outer radial glial cells (oRGs) contribute to mammalian neocortical expansion; however, a comprehensive understanding of their mechanisms is lacking. Here, we demonstrate that Lzts1, a molecule associated with microtubule components, promotes both cell departure events. In neuronally committed cells, Lzts1 functions in apical delamination by altering apical junctional organization. In apical RGs (aRGs), Lzts1 expression is variable, depending on Hes1 expression levels. According to its differential levels, Lzts1 induces diverse RG behaviors: planar division, oblique divisions of aRGs that generate oRGs, and their mitotic somal translocation. Loss-of-function of *lzts1* impairs all these cell departure processes. Thus, Lzts1 functions as a master modulator of cellular dynamics, contributing to increasing complexity of the cerebral architecture during evolution.

[1] Department of Anatomy and Cell Biology, Nagoya University Graduate School of Medicine, 65 Tsurumai, Showa-ku, Nagoya, Aichi 466-8550, Japan. [2] Laboratory for Cell Asymmetry, RIKEN Center for Biosystems Dynamics Research, 2-2-3 Minatojima-minamimachi, Chuo-ku, Kobe, Hyogo 650-0047, Japan. [3] Graduate School of Biostudies, Kyoto University, Yoshida-Konoe-cho, Sakyo-ku, Kyoto 606-8501, Japan. [4] Present address: Mechanobiology Institute, National University of Singapore, T-Lab, 5 A Engineering Drive 1, Singapore 117411, Singapore. [5] Present address: Division of Anatomy, Department of Human Development and Fostering, Meikai University School of Dentistry, 1-1 Keyakidai, Sakado, Saitama 350-0283, Japan. Correspondence and requests for materials should be addressed to A.K. (email: akawa@med.nagoya-u.ac.jp)

During mammalian neocortical development, radially elongating apical radial glial cells (aRGs = apical progenitors, APs or vRGs) undergo interkinetic nuclear migration (INM) and divide to produce two daughter cells[1,2] at the apical surface. Two major pathways have been identified for daughter cells to disconnect and depart from the apical surface (Fig. 1a). One is neuronal delamination by cells committed to differentiate into the neuronal lineage (mitotically active intermediate/basal progenitor (IP = BP) cells and postmitotic neurons)[2–4], and the other is oblique (or perpendicular) division related to the generation of outer radial glial cells (oRGs = bRGs)[2,5]. oRGs are undifferentiated neural progenitor cells that divide multiple times in the subventricular zone (SVZ). These cells are more abundant and proliferative in the species with gyrencephalic brains, such as humans, than in mice with lissencephalic brains[6–11], indicating that the cell departure process that repositions the neural progenitors is also critical for the evolutionary expansion of the neocortex.

Neuronally differentiating cells, i.e., neurons or IPs[3,4], are typically generated by the horizontal division of aRGs, by which they inherit the apical membrane at birth[12,13]. Next, neuronally differentiating cells retract their apical processes to delaminate from the cadherin-based adherens junction (AJ) belt[14] that packs the apical endfeet of ventricular zone (VZ) cells together[2]. A recent study revealed a centrosome-nucleated wheel-like microtubule configuration aligned with the apical actin cable and AJs at the apical endfeet of aRGs[15]. This cytoskeletal configuration maintains AJs, and during neuronal delamination, it shows dynamic changes, including constriction of the actomyosin ring of apical processes[16]. Along with this, epithelial–mesenchymal transition (EMT)-related transcription factors induce neuronal delamination[17]. However, researchers have not clearly elucidated the mechanism by which only differentiating cells delaminate from the apical surface within several hours after birth[3,18].

oRGs are typically produced by the oblique (or perpendicular) division of a subset of aRGs (Fig. 1a). In this case, basal daughter cells, i.e., newly generated oRGs, do not inherit the apical AJ belt and can migrate to the basal side[5,12,13,19,20], exhibiting mitotic somal translocation (MST) in which the soma rapidly translocates basally/forward before cytokinesis[6,21,22]. These unique cellular behaviors, oblique division and MST, show evolutionary changes in their frequency and distance with relation to the size of the germinal zone in the species[20,21]. Currently, the regulatory mechanisms that evoke oblique division and the potential molecular mechanisms underlying the correlation with oblique aRG division and MST are unknown[23].

Our previous single-cell transcriptome analysis[24,25] identified leucine zipper putative tumor suppressor 1 (lzts1; also known as FEZ1 and PSD-Zip70)[26–29] as a gene displaying significantly higher expression in nascent IPs than in aRGs. Therefore, lzts1 is a candidate gene that regulates neuronal delamination. In addition, lzts1 is a tumor suppressor gene that has been implicated in several human cancers[26]. Lzts1 is associated with microtubule components and is involved in microtubule assembly[27], further suggesting a possible function in cytoskeletal dynamics.

Here, we report that Lzts1 positively controls both neuronal delamination and oRG generation in an expression level-dependent manner. Our findings support the hypothesis that these different events are both aspects of the same process, continuously varying cellular dynamics controlled by Lzts1. We propose that Lzts1 functions as a master modulator of the cytoskeleton, including both the actomyosin system and microtubules, to produce diverse cell behaviors in the cell departure processes.

## Results

**Lzts1 is expressed at AJ of neuronally differentiating cells**. Our previous single-cell transcriptome profiling of mouse embryonic day 14 (E14) cortical progenitor cells[24,25] revealed that lzts1 is expressed in a subset of aRGs and all IPs and neurons, including nascent IPs (Fig. 1b). Consistent with the in situ expression pattern (Supplementary Figure 1), Lzts1 immunoreactivity was observed in some VZ cells, in addition to the SVZ, intermediate zone (IZ) and cortical plate (Fig. 1c, d). These Lzts1+ VZ cells were Tbr2 (Eomes::EGFP+ [30] (Fig. 1e) and Gadd45g::d4Venus+ [31] (Fig. 1f). The simultaneous expression of both a transcription factor, tbr2, and a negative cell-cycle regulator, gadd45g, in neuronally differentiating cells (nascent neurons and IPs)[24] suggests that the Lzts1+ VZ cells were nascent differentiating cells.

Particularly strong Lzts1 immunoreactivity was detected at the apical endfeet of some cells in a ring-like pattern (Fig. 1d). Lzts1 co-localized with ZO1, a scaffolding protein in AJs, in Gadd45g::d4Venus+ apical endfeet (Fig. 1g, h) by en face analysis, indicating that Lzts1 is expressed close to the AJ belt of nascent differentiating cells[31]. The Lzts1 immunofluorescence signal values at the AJs were negatively correlated with the values (apex area^[1/2] μm) that are proportional to the circumferential length of the ZO1+AJ ring (Fig. 1i), raising the possibility that Lzts1 has a role in delamination based on its localization to the AJ. If this hypothesis is true, the dot-like Lzts1 immunoreactivity observed among the ZO1+ meshwork (Fig. 1g) represents the apical endfeet of departing cells, because separate live images of the neuronally differentiating cells revealed that their apices become progressively smaller to form a vertex through delamination[31].

Lzts1 immunolabeling by electron microscopy showed intracellular gold particles that were closely associated with the AJ belts (Fig. 1j, arrowheads), suggesting that Lzts1 may interact with AJ-related proteins, including the associated cytoskeleton. Most particles were located within 3~4 μm of the apical surface, but a subset of cells contained particles located more basal, intracellular, or adjacent to the plasma membrane (Fig. 1k, arrows, ~100 μm from the apical surface).

**Lzts1 knockdown impairs Neurog1/2-induced delamination**. We next examined whether Lzts1 knockdown (KD) perturbs radial neuronal migration from the apical surface. To increase the incidence of neuronal differentiation[4,32], we introduced neurog1 and neurog2, key regulators of neuronal differentiation, into the E13 mouse cerebral wall through in vivo electroporation. Lzts1 expression was induced 18 h after electroporation (Fig. 2a), and by 24 h after electroporation >90% of electroporated cells were positioned outside the region within 0–60 μm from the apical surface (Fig. 2b, c, +Neurog1/2 case). We then knocked down Lzts1 in cells overexpressing Neurog1/2 (Fig. 2b, c), using two siRNAs with different target sequences; siRNA#1 more strongly suppressed Lzts1 expression than siRNA#2 (Supplementary Figure 2). One day after the electroporation of Lzts1-siRNA#1 (and #2, to a more moderate degree), a significant number of EGFP+ cells remained in the region within 0–60 μm from the apical surface (Fig. 2c). These EGFP+ VZ cells were Tbr2+, and RG markers[33,34] Sox2− and Pax6− (Supplementary Figure 3), indicating that Lzts1 KD did not inhibit neurogenin-induced differentiation. Furthermore, co-expression of siRNA#1-resistant Lzts1 (Supplementary Figure 2), but not wild-type Lzts1, antagonized the effect of siRNA#1 (Fig. 2b, c). These results suggest that the loss of Lzts1 retards neurogenin-induced neuronal migration from the apical surface.

Notably, co-expression of membrane-targeted Lyn-EGFP visualized many of the Lzts1 KD cells retained long apical

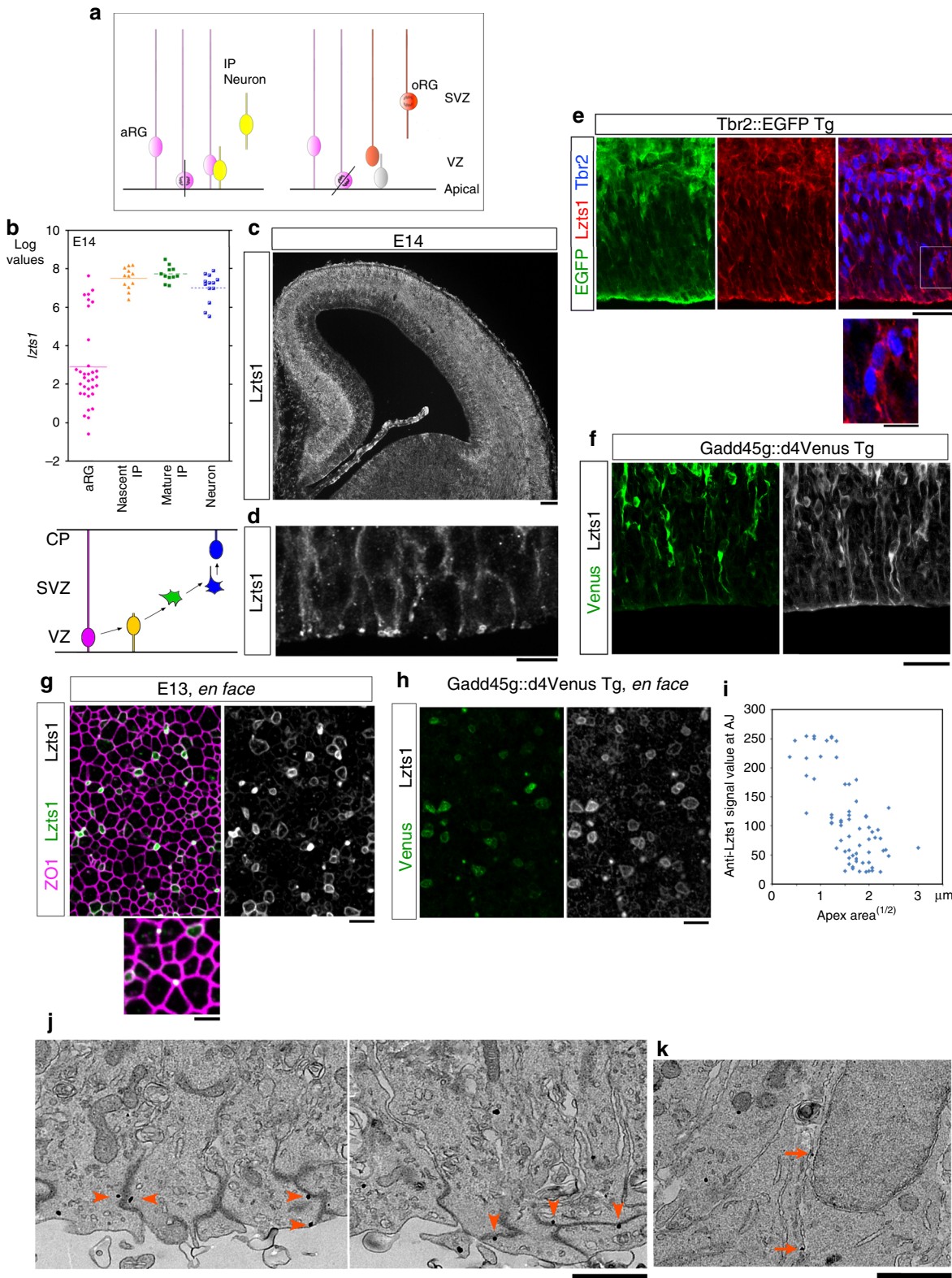

processes attached to the apical surface (Fig. 2d, arrowheads). Therefore, the retardation of neuronal migration induced by Lzts1 KD is most likely caused by a delamination defect in the cells.

**Loss-of Lzts1 impairs cell departure from the VZ.** Lzts1 KD did not significantly alter the percentage of Pax6+ cells (Fig. 3a),

BrdU incorporation after 30 min of labeling (Fig. 3b), Tbr2+ cells (Fig. 3c) or the mitotic index in the electroporated cells (Fig. 3d), suggesting that the loss of Lzts1 did not primarily affect cellular differentiation. Then, we examined whether the Lzts1 loss-of-function affected neuronal cell migration under physiological conditions. The introduction of the CRISPR/Cas9 system by in vivo electroporation[35] at E13 successfully abolished Lzts1 expression after 2 days (Fig. 3e, f). These Lzts1 knockout (KO)

**Fig. 1** Lzts1 is expressed at high levels at the AJs of delaminating cells. **a** Neuronal delamination and oRG generation by oblique aRG division are two different ways for departure from the apical surface. IP, intermediate progenitor cell. **b** Single-cell transcriptome profiles[24] of E14 cells show increased expression of the *lzts1* mRNA during neuronal differentiation. (Affymetrix ID: 1433988_s_at, annotated as *C230098O21Rik*, a transcript variant of *lzts1*). **c** Anti-Lzts1 immunohistochemistry (IHC) of the E14 mouse brain (see also Supplementary Figure 1). **d** Magnified view of the E13 brain section stained with the anti-Lzts1 antibody showing both dot-like and ring-like expression of Lzts1 at high levels in some apical endfeet. Z-projection images of 8-μm thick slices. **e** and **f** Lzts1 IHC of the E14 dorsal forebrain of Tbr2::EGFP Tg mice, with magnified view of Tbr2+Lzts1+ cells in the VZ **e** and Gadd45g::d4Venus Tg mice **f** showing that GFP+, Tbr2+, or Venus+ differentiating cells are Lzts1+. **g** and **h** Lzts1 is expressed at high levels at the AJs of the apical endfeet of differentiating cells. *En face* observations of anti-Lzts1 and anti-ZO1 IHC of the E13 dorsal forebrain **g** or anti-Lzts1 and anti-GFP IHC of E13 Gadd45g::d4Venus Tg mouse dorsal forebrain **h** from the apical surface. In the magnified view **g**, dot-like signals may represent the apical endfeet of cells that had almost completed the delamination from the apical surface. **i** Anti-Lzts1 signal intensities along the cellular junctions were negatively correlated with the apex (apical) area^(1/2), which is proportional to the planar circumferential length of the AJ ring. **j** and **k** Ultrastructural localization of Lzts1 in the E14 dorsolateral cerebrum. Immunoelectron microscopy using an anti-Lzts1 antibody shows that intracellular Lzts1 gold particles were closely located to the electron-dense zone of AJs (**j**, arrowheads). Particles with an intracellular distribution or located adjacent to the plasma membrane were also observed in a subset of the cells (**k**, arrows, ~ 100 μm from the apical surface). Bars, 100 μm in **c**, 10 μm in **d**, magnified view **e**, 30 μm in **e**, **f**, 5 μm in **g**, **h**, and 1 μm in magnified view **g**, **j**, **k**

cells were positioned more apically than negative control cells (Fig. 3g), and similar results were obtained with Lzts1 KD cells (Supplementary Figure 4). Moreover, both Lzts1 KD and Lzts1 KO increased the number of apically positioned Tbr2+ EGFP+ cells, and this KD-induced phenotype was rescued by the expression of siRNA-resistant Lzts1 (Fig. 3h–j). Thus, the loss of Lzts1 disturbs radial neuronal migration without affecting differentiation.

Because neuronally differentiating cells are known to delaminate from the apical surface with contraction of their AJ rings[15,16], we performed an en face observation of the apical endfeet of the Lzts1 KO cells. We introduced a guide RNA (gRNA) for *lzts1*, hCas9, and RFP into the E13 Gadd45::d4Venus mice. As a control, we expressed hCas9 and RFP. After 2 days (Fig. 4a–c), the size of the AJ ring in the differentiating Venus+ cells was significantly smaller than that in the Venus− cells both in the control and the Lzts1 KO cells ($p < 1.0 \times 10^{-7}$ and $p < 1.0 \times 10^{-7}$). Furthermore, no significant difference in the AJ ring length was observed between control and Lzts1 KO Venus+ cells ($p = 0.76$, Steel–Dwass test) (Fig. 4b). These results suggest that the contraction of the AJ ring in the differentiating cells occurs even in the absence of Lzts1. On the other hand, Lzts1 KO significantly increased the percentage of Venus+ apical endfeet among the RFP+ endfeet ($p = 0.0072$, Wilcoxson rank sum test) (Fig. 4c). Because the loss-of-function of Lzts1 did not increase neuronal differentiation (Fig. 3a–d), this result suggests that the detachment of the apical process from the apical surface is retarded by Lzts1 KO.

Together, these results indicate that Lzts1 is necessary for appropriate neuronal delamination from the apical surface.

**Lzts1 overexpression (OE) induces delamination**. Next, we performed an Lzts1 gain-of-function study. We forcibly expressed Lzts1 (at a plasmid concentration of $1.0\,\mu g\,\mu l^{-1}$, indicated as "Lzts1 1.0") with Lyn-EGFP, a membrane-targeted EGFP used to visualize cell morphology, under the control of a ubiquitous CAG promoter (Fig. 5a). After 2 days, almost all EGFP+ Lzts1-expressing cells lost their apical processes and were positioned outside of the VZ (Fig. 5b). Interestingly, some Lzts1-overexpressing cells in the SVZ showed a monopolar morphology with long basal processes and were Sox2+, which are both characteristic features of oRGs (Fig. 5c). An Lzts1-mutant study revealed that both localization mediated by N-terminal myristoylation[28] and the C-terminal sequence are necessary for the observed cellular localization outside the VZ (Supplementary Figure 5).

Lzts1 overexpression positioned the cells outside the VZ (Fig. 5d), with the emergence of many Ki67+ EGFP+ progenitor

cells in the SVZ and IZ (Fig. 5e), and these ectopic progenitors included both Sox2+ cells (Fig. 5f) and Tbr2+ cells (Fig. 5g). Almost all (> 98%) Lzts1-expressing PH3+ M-phase cells were present in the non-apical surface region (Fig. 5h, i). However, we did not observe significant changes in the frequencies of BrdU+ cells (30-min labeling) (Fig. 5j), Pax6+ cells (Fig. 5k), and Tbr2+ cells (Fig. 5l), or the mitotic index in electroporated cells (Fig. 5m). Thus, Lzts1 does not primarily affect the proliferation and differentiation of neural progenitor cells, but affects the cellular dynamics.

The Lzts1-overexpressing brains had also an increased number of EGFP− SVZ progenitors, which included both Sox2+ and Tbr2+ cells (Fig. 5e–h). Therefore, Lzts1 overexpression may exert a non-cell autonomous effect on the cell-positioning, possibly through the homophilic cell adhesion among neural progenitor cells, in addition to the possible cell-extrinsic factors that function in regulating the proliferation and maintenance of the surrounding cells.

A previous study suggested that Scrt1, an EMT-related transcription factor, starts to express at an early time point of neuronal differentiation and promotes apical process detachment[17]. Therefore, Scrt1 may induce Lzts1 expression to evoke neuronal delamination. To examine this possibility, we overexpressed Scrt1 in the brain; Scrt1 overexpression resulted in cells positioned outside the VZ[17] without increasing Lzts1 immunoreactivity (Supplementary Figure 6), suggesting that Scrt1 does not induce Lzts1 expression.

**Lzts1 OE induces apical process retraction and MST**. Next, we performed live imaging to observe the behaviors of aRG-like cells expressing exogenous Lzts1 (at the concentration $1.0$–$2.0\,\mu g\,\mu l^{-1}$) with EGFP and Lyn-EGFP (Fig. 6a–c). In the control experiment, the nuclei of all EGFP-labeled aRG-like cells (63 cells that we observed) moved apically and soon divided at the apical surface (designated as "normal") (Fig. 6d). In contrast, the aRG-like cells with forced Lzts1 expression behaved differently, showing the following three patterns: retraction of the apical process (Type A), retraction of the apical process followed by MST (Type B), and MST from the apical surface after apical INM (Type C) (Fig. 6a–c and Supplementary Movies 1–4).

The first pattern was the simple retraction of the apical process without preceding nucleosomal movements and subsequent MST or division (Type A) (Fig. 6a and Supplementary Movie 1). The observed retraction of the apical process appeared similar to differentiating cells[3,31].

The second pattern was characterized by a brief MST following the retraction of apical processes. The cell retracted its apical process, moved its cell body basally, and soon after divided into

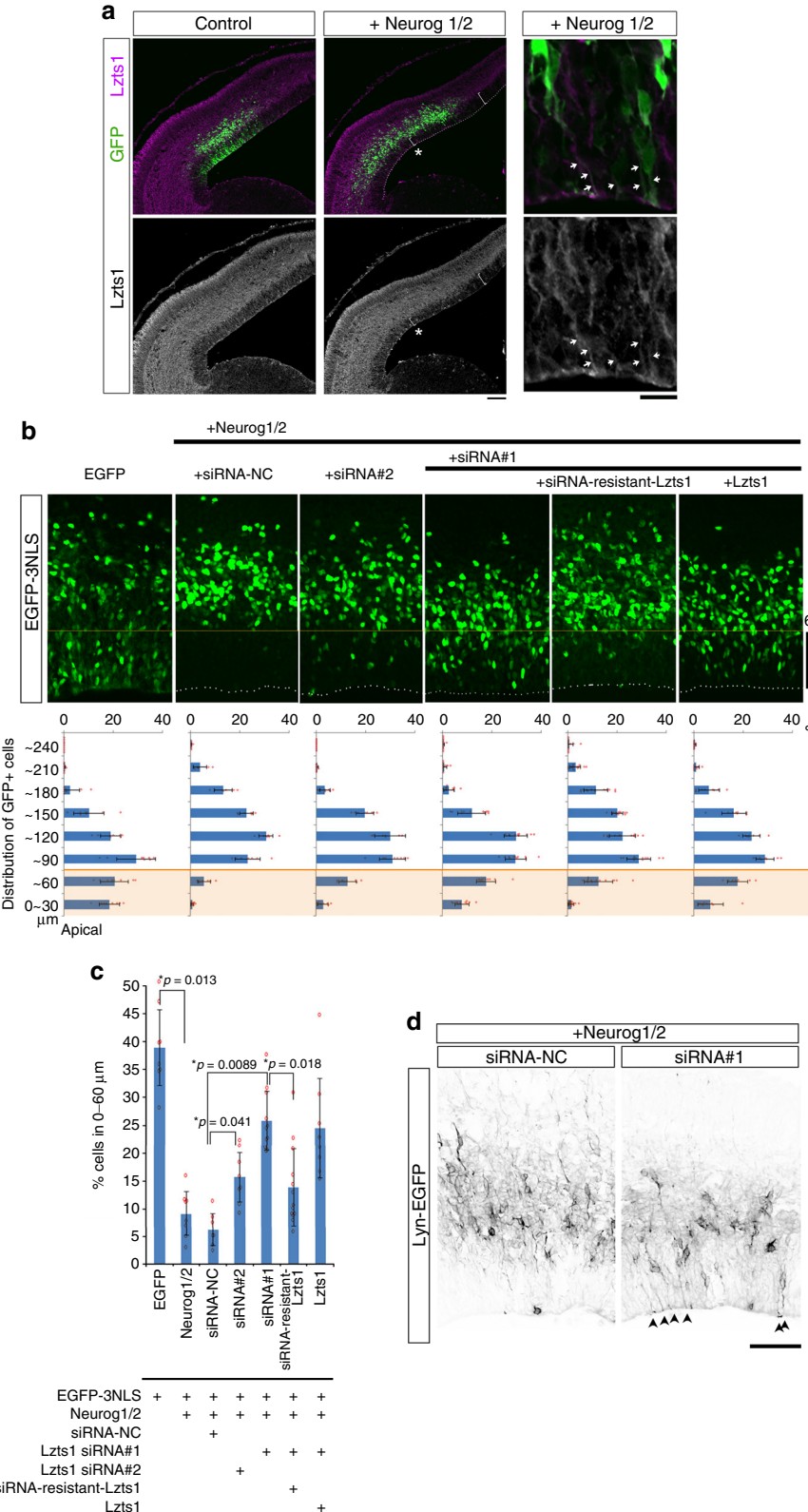

daughter cells (Type B) (Fig. 6b and Supplementary Movie 2). This behavior appeared similar to that of some IPs in mouse cerebral slice cultures[3,36].

The third pattern was MST from the apical surface after apical INM (Type C) (Fig. 6c and Supplementary Movie 3). The progenitor cell underwent apparently normal apical INM during G2 phase and arrived at the apical surface. However, once the cell entered M phase, its cell body moved to the basal side at an unusually fast rate (~ 58 μm in 20 min in Fig. 6c), going through the VZ, and underwent cytokinesis in the SVZ. We have never seen this type of cellular behavior in wild-type mouse brain slices. This pattern resembles that of some aRGs in ferrets and humans that undergo MST to generate an oRG daughter cell following INM to the apical surface during G2 phase[22]. Although our

**Fig. 2** Lzts1 KD perturbs the delamination of neurog1/2-expressing cells. **a** Lzts1 expression is upregulated throughout neuronal differentiation following the forced co-expression of *neurog1* and *neurog2*. Neurog1/2, and GFP were co-expressed at E13 by in vivo electroporation, and sections were examined after 18 h. Neurog1/2 co-expression expands the apico-basal width of Lzts1+ area and reduces the depth of the VZ (shown by asterisk) compared with that in the non-electroporated region. Magnified view shows that the Neurog1/2-expressing EGFP+ cells are Lzts1+. Arrows indicate the GFP+Lzts1+ apical processes. **b** and **c** Neurog1/2-induced migration from the VZ is partially inhibited by Lzts1 KD induced by siRNA#1 (and more moderately by siRNA#2). This phenotype was rescued by the expression of siRNA#1-resistant Lzts1 but not wild-type Lzts1. In vivo electroporation of *neurog1/2* and *egfp* was performed at E13, and brains were examined after 24 h (see also Supplementary Figure 2). siRNA-NC, negative control siRNA. **b** Distributions of EGFP+ cells. **c** Percentage of the EGFP+-electroporated cells within the area of 0–60 μm from the apical surface (orange colored region in **b**) among total EGFP+ cells (Steel–Dwass test, $N = 8, 8, 7, 8, 11, 13,$ and 8 sections from 4, 4, 4, 4, 7, and 4 embryos, respectively). **d** Cre-mediated Lyn-EGFP (EGFP with a membrane targeting sequence) labeling shows that forced expression of Neurog1/2 causes cells to retract their apical processes, whereas many Lzts1 KD cells retain their processes (arrowheads). In vivo electroporation of *neurog1/2*, *lyn-egfp*, and siRNAs was performed at E13, and brains were examined at E14. Bars, 100 μm in **a**, 10 μm in magnified view **a**, 60 μm in **b**, 50 μm in **d**. Means ± s.d. Source data are provided as a Source Data file

observations apparently differed from these reported cell behaviors in that the apical daughter cell of division retained the thin apical process for a while, use of a membrane-targeted form of fluorescent protein, such as Lyn-EGFP, may be able to be used to visualize potentially retained apical processes in ferrets/humans.

Notably, Lzts1-expressing aRG-like cells showed these various behaviors according to the concentration of the *lzts1* expression vectors introduced during electroporation (Fig. 6e); Type A behaviors were more frequently observed in "Lzts1 2.0" (34/75 = 45.3%) than in "Lzts1 1.0" (10/94 = 10.6%) ($p = 4.0 \times 10^{-7}$, Fisher's exact test, two-sided). As the concentration of the expression vectors introduced during electroporation was positively correlated with the Lzts1 expression levels in the electroporated cells (Supplementary Figure 7), these diverse cellular behaviors are postulated to correspond to the Lzts1 expression levels in the cells, although it might also include a non-cell autonomous effect.

**Lzts1 activates the actomyosin system in delamination**. As the MST of oRGs is mediated by the Rho–ROCK–myosin pathway[5,21], we performed its inhibitor experiment. The administration of myosin II inhibitor blebbistatin decreased the distance of Lzts1-induced MST (Fig. 7a) in slice culture, indicating the Lzts1-induced MST is mediated by the activation of the actomyosin system.

To determine whether the activation of the actomyosin system is also involved in Lzts1-induced delamination, we next examined the contraction of the apical endfeet. Forced expression of Lzts1 strikingly contracted the AJ ring (Fig. 7b), and, furthermore, the in vivo administration of blebbistatin into the lateral ventricles significantly decreased this Lzts1-induced contraction of the AJ ring (Fig. 7c, d). These results suggest that Lzts1 expression induces apical contraction by activating the actomyosin system, which is similar to the apical abscission observed in neuronal delamination[15,16]

Importantly, it has been reported that the activation of myosin II contracts the apical endfeet but is not sufficient to induce delamination from the apical surface[16]. As the onset of neuronal delamination is characterized by the downregulation of N-cadherin at the AJs[16,37], we examined its expression at the Lzts1-overexpressing apical endfeet. En face observations revealed the intensity of the N-cadherin immunofluorescence was weaker at the small AJ rings ($< 1$ μm apex area^[1/2]) (Fig. 7e, f), indicating that the Lzts1-mediated apical contraction was accompanied by the downregulation of N-cadherin.

To further clarify the effect of Lzts1 on the cytoskeletons, we used the NIH3T3 mouse fibroblast cell line to minimize the noise based on the cellular heterogeneity. NIH3T3 cells overexpressing Lzts1 exhibited increased immunoreactivity for phospho-myosin light chain 2 accompanied by a change in morphology to a narrow shape (Fig. 7g). Our atomic force microscope (AFM) indentation measurement[38] indicated that exogenous Lzts1 expression increased the stiffness of the NIH3T3 cells ($p = 2.2 \times 10^{-4}$, Steel–Dwass test), and blebbistatin antagonized this effect (Fig. 7h). These results suggest that the exogenous Lzts1 activates the actomyosin system in NIH3T3 cells to increase the stiffness of the cells.

Then, to examine whether the reported inhibitory effect of Lzts1 on microtubule assembly[27] was necessary for the Lzts1-mediated actomyosin activation, we added taxol to the medium. The taxol treatment itself increased cellular stiffness both in the control and the Lzts1-overexpression NIH3T3 cells. In Lzts1-overexpressing cells, this taxol-induced increase in stiffness was attenuated by the addition of blebbistatin, suggesting that Lzts1 activates myosin II even in response to microtubule stabilization (Fig. 7h). Therefore, Lzts1 may have a general function to enhance the activity of the actomyosin system, contributing to cellular stiffness or contraction, which might not necessarily require the inhibitory effect of Lzts1 on microtubule assembly.

Taken together, we proposed a model that Lzts1 in neuronally differentiating cells modulates the microtubule-actin-AJ system at the apical endfeet to evoke apical contraction and reduce N-cadherin expression, both of which ensure the quick delamination from the apical surface (Fig. 7i).

**Weak Lzts1 expression induces oblique aRG division**. The Type C departure, i.e., the MST from the apical surface (Fig. 6c), suggests that Lzts1 inhibits the anchoring of centrosomes to the apical portion of the process forming the AJs during M phase. If this is the case, weak expression of Lzts1 in aRGs will weaken the anchoring machinery of centrosomes and give rise to oblique aRG division, which may mimic the physiological condition of oRG production. We electroporated *lzts1* expression vectors at a low concentration (0.2 μg μl$^{-1}$) and measured the mitotic spindle orientation of aRGs (Fig. 8a–e)[39]. Approximately 45% of Lzts1-expressing aRGs divided at the apical surface (Fig. 5i), and among them, a significant number of aRGs showed an oblique spindle orientation that was rare in control cells (Fig. 8e).

In obliquely dividing cells, Lzts1-FLAG was localized to the cell cortex (Fig. 8b). Next, we examined the localization of LGN because LGN is localized to the cell cortex and binds Numa to orient the mitotic spindle by anchoring spindle astral microtubules[40]. The basolateral localization of LGN was maintained, but it showed a slightly diagonal pattern along the direction of the division (Fig. 8a). In some cases, the apical AJ ring was inherited asymmetrically by apical daughter cells (Fig. 8c, arrow).

As both MST and oblique divisions are characteristic features of oRGs or oRG-producing aRG division[5], our results raise an intriguing possibility that Lzts1 positively regulates oRG generation in vivo.

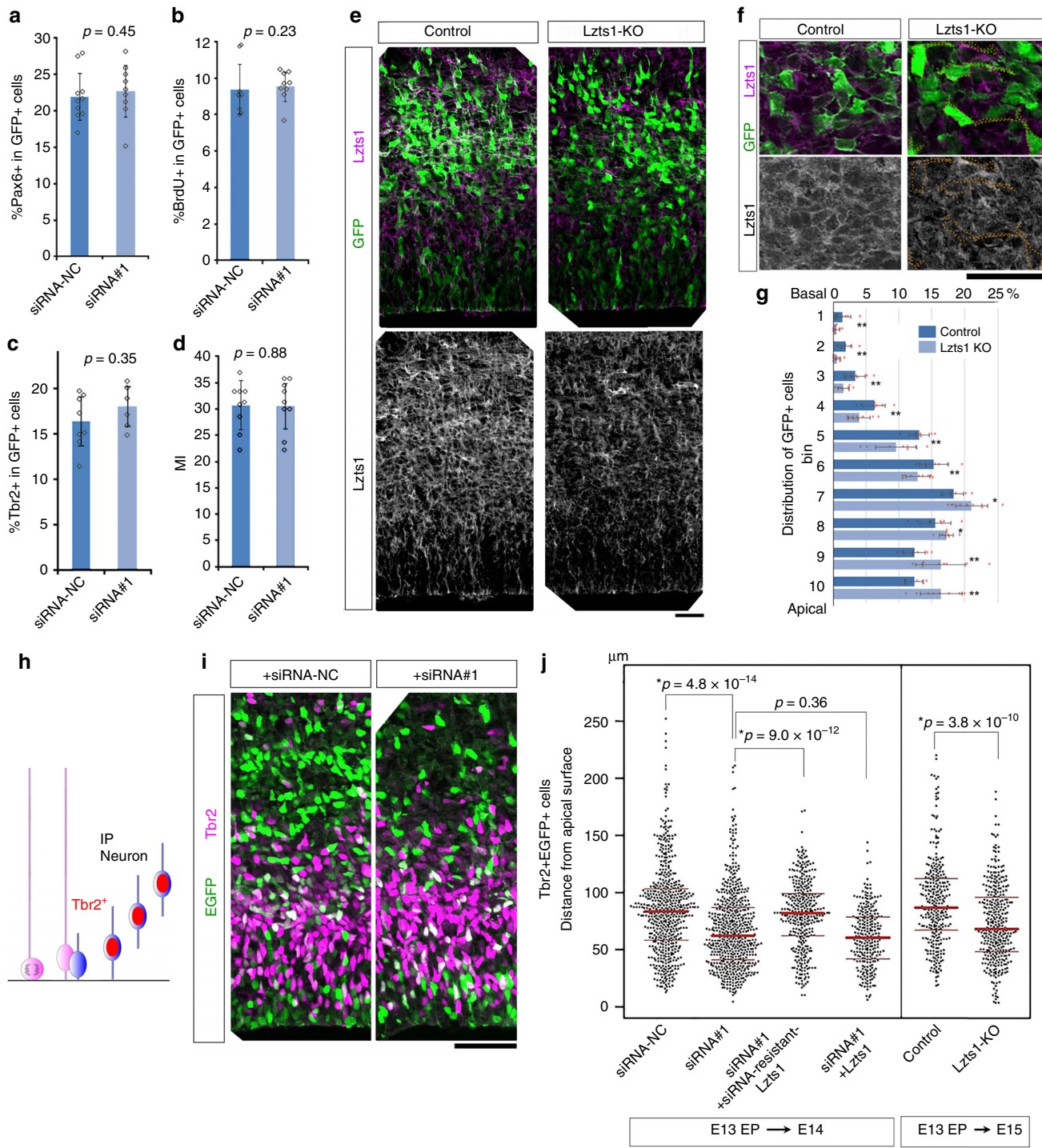

**Lzts1 expression in aRGs varies with developmental time**. We first carefully examined *lzts1* expression levels in individual aRGs during development. The single-cell transcriptome profiles[24,25] showed that some E14 aRGs (7 of 33 cells) expressed the *lzts1* transcript at slightly lower levels than IPs or neurons (Figs 1b and 9a). E11 and E16 IPs expressed *lzts1* at high levels, but we did not observe *lzts1*+ aRGs at E11 and E16 (Fig. 9a). Thus, *lzts1* expression in aRGs might depend on the developmental time window, although its regulatory mechanism is currently unknown. To characterize *lzts1*+ aRGs at E14, we compared the expression levels of six transcription factors and co-regulators related to the differentiation or maintenance of progenitors (*hes1*,

*pax6*, *sox2*, *neurog2*, *btg2* and *tbr2*)[24,33,34,36,41] between *lzts1*+ aRGs (*N* = 7) and *lzts1*− aRGs (*N* = 26) (Fig. 9b–d and Supplementary Figure 8). Among these genes, *hes1* was expressed at low levels in *lzts1*+ aRGs and high levels in *lzts1*− aRGs, and a significant difference was observed between the two cell populations (*p* = 0.045, permuted Brunner–Munzel test) (Fig. 9b). Because these transcriptome profiles provide a snapshot of the cells at a certain time point, the low level of *hes1* expression in aRGs does not necessarily mean that the cells do not express Hes1 or are differentiating. Rather, *hes1* has been reported to exhibit oscillatory expression in single aRGs[41] and variable expression in the aRG population[24]. Because Hes1 represses Neurogenin

**Fig. 3** Loss-of-function of Lzts1 retards radial migration through differentiation. **a–d** Lzts1-siRNA#1 or a negative control siRNA (siRNA-NC) was in vivo electroporated at E13, and sections of the E14 **c** or E15 **a, b, d** brain were examined using IHC with anti-Pax6 **a**, anti-BrdU (30-min BrdU pulse labeling) **b**, anti-Tbr2 **c**, anti-Ki67 and anti-BrdU **d** antibodies. The mitotic index (MI) **d** indicates the percentage of Ki67[+] cells among BrdU[+] GFP[+] cells that received BrdU 20 h before fixation. Means ± s.d., **a** $N = 9, 9$; **b** $N = 10, 9$; **d** $N = 8, 6$; **d** $N = 9$, nine sections from three embryos per experiment, Wilcoxon rank sum test. **e** and **f** CRISPR/Cas9-induced disruption (KO) of *lzts1* successfully reduces Lzts1 expression. The hCas9 and guide RNA for *lzts1* were co-expressed with EGFP by performing in vivo electroporation at E13, and brains were examined at E15. The majority of EGFP[+] cells were negative or exhibit weak Lzts1 immunoreactivity. **f** Magnified view. Bars, 30 μm. **g** Lzts1 KO retards the overall radial migration of cells from the apical surface. In vivo electroporation was performed at E13, and the distribution of EGFP[+] cells in the cerebral wall was examined at E15 using 10 bins. Means ± s.d., Brunner–Munzel test, *$p < 0.05$, **$p < 0.01$, $N = 8$ (control) and 9 (KO) sections from three embryos per experiment (see also Supplementary Figure 4). **h** Scheme for Tbr2 expression in differentiating cells. **i** and **j** Both Lzts1 KD and KO significantly slowed the migration of Tbr2[+] cells from the apical surface. This KD effect was rescued by the siRNA-resistant Lzts1, but not by wild-type Lzts1. In vivo electroporation was performed at E13, and the migration of EGFP[+] Tbr2[+] cells from the apical surface was examined at E14 in KD experiments or at E15 in KO experiments (Steel–Dwass test for multiple comparisons among the four conditions in KD experiments; Brunner–Munzel test for KO experiments; $N = 547, 568, 351, 257, 306$, and 344 cells from 8, 8, 11, 9, 7, and 8 sections of 4, 4, 4, 3, 3, and 3 embryos, respectively; Medians with Q1 and Q3 values). Bar, 50 μm. Source data are provided as a Source Data file

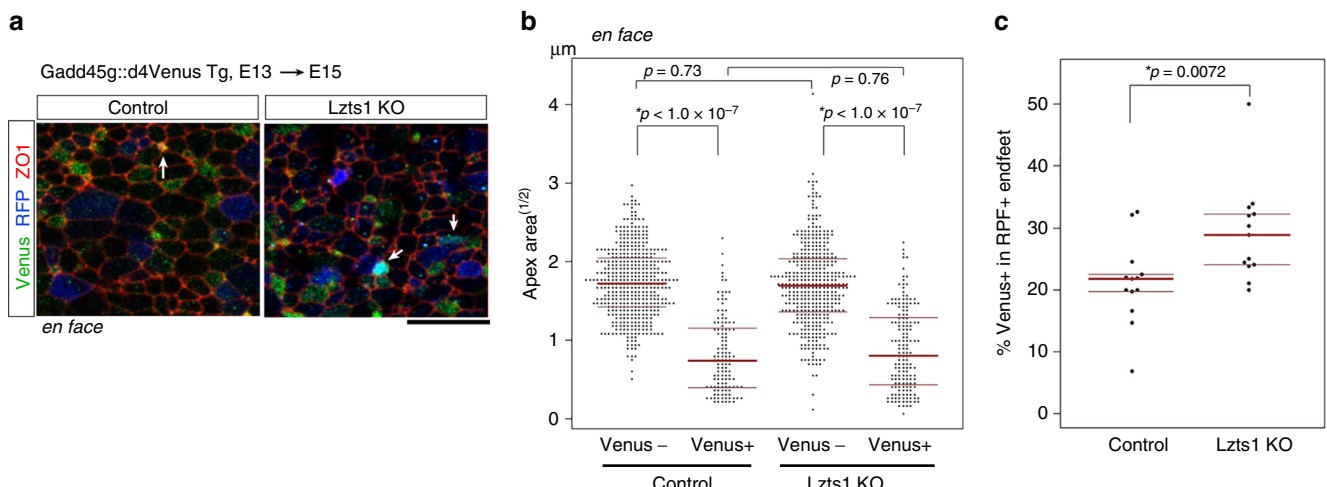

**Fig. 4** Lzts1 is dispensable for apical contraction but ensures rapid delamination. **a–c** CRISPR/Cas9-mediated Lzts1 KO was performed by electroporating the plasmids into E13 Gadd45g::d4Venus Tg mice in vivo, and the apical endfeet were examined at E15 by performing en face observations. **a** Arrows indicate the RFP[+] (electroporated) cells that are Venus[+] (neuronally differentiating). **b** Lzts1 KO did not change the distribution of the circumferential length of AJ rings in both Venus[+] and Venus[−] cell populations. $N = 445, 124, 404$, and 162 cells from five (control) and six (KO) embryos, respectively, Steel–Dwass test. **c** Lzts1 KO increased the percentage of Venus[+] apical endfeet among RFP[+] endfeet. $N = 13$ and 13 fields from six embryos per experiment, Wilcoxson rank sum test. Bar, 5 μm. Medians with Q1 and Q3 values are reported. Source data are provided as a Source Data file

expression, which induces Lzts1 expression (Fig. 2a), we speculated that the variable *hes1* expression explains the variable *lzts1* expression in the aRG population (discussed below).

**Lzts1 is responsible for oRG-like cell generation**. Then we investigate whether Lzts1 is involved in generation of oRG-like cells in vivo. Lzts1 KO by in vivo electroporation significantly decreased the angle (θ) of spindle orientation in the mitotic aRGs ($p = 0.016$, Brunner–Munzel test); however, the difference in the angles between control and KO was small (Fig. 9e). Therefore, we next examined whether Lzts1 KO actually affected the frequency of the oblique division of aRGs that induces asymmetrical inheritance of the apical junction by one of the daughter cells[19] (Fig. 9f). We performed time-lapse en face imaging of aRG divisions using EGFP-ZO1 and PACT-mCyRFP to visualize the apical junction and centrosomes, respectively. In 24 h, the Lzts1 KO cells showed significantly fewer oblique divisions ($p = 0.0067$, Fisher's exact test, two-sided), suggesting that Lzts1 KO decreased the generation of the daughter cells that did not inherit apical junctions.

Then, we examined the effect of the loss-of-function of Lzts1 on the number of newly generated oRG-like cells. We electroporated Lzts1-siRNA#1 or NC-siRNA along with an EGFP-3NLS expression vector into the E13 or E14 cerebrum. After 2 days, the percentage of Sox2[+] cells among EGFP[+] SVZ cells was significantly reduced by Lzts1 KD in animals electroporated at E14 ($p = 0.046$, Steel–Dwass test) (Fig. 9g–i). More Sox2[+] GFP[+] SVZ cells were observed at E14 → E16 than at E13 → E15, consistent with previous reports[7]. The observed decrease induced by Lzts1-siRNA#1 at E14 → E16 was rescued by the siRNA-resistant Lzts1, but not wild-type Lzts1, further supporting the hypothesis that this decrease was induced by the loss of Lzts1 expression.

Although the decrease in Sox2[+] oRG-like cells induced by Lzts1 KD might be explained by the delayed migration of newly generated Sox2[+] oRG-like cells, the distribution of Sox2[+] GFP[+] cells outside the VZ did not show a clear difference between control and KD cells ($p = 0.97$, Wilcoxon rank sum test) (Supplementary Figure 9), suggesting that this explanation is not the primary mechanism. Rather, these results together with the gain-of-function phenotypes (Fig. 8e) suggest that Lzts1 positively controls oRG generation by modulating aRG division angles.

On the other hand, we also found that Lzts1 KD at a later stage (E16 → E18 experiment) did not perturb the generation of Sox2[+]GFP[+] cells in the SVZ (Supplementary Figure 10). Thus, the generation of SVZ progenitor cells at the late, gliogenic stage might be controlled by a different molecular mechanism from the mid, neurogenic stage, at least in mice.

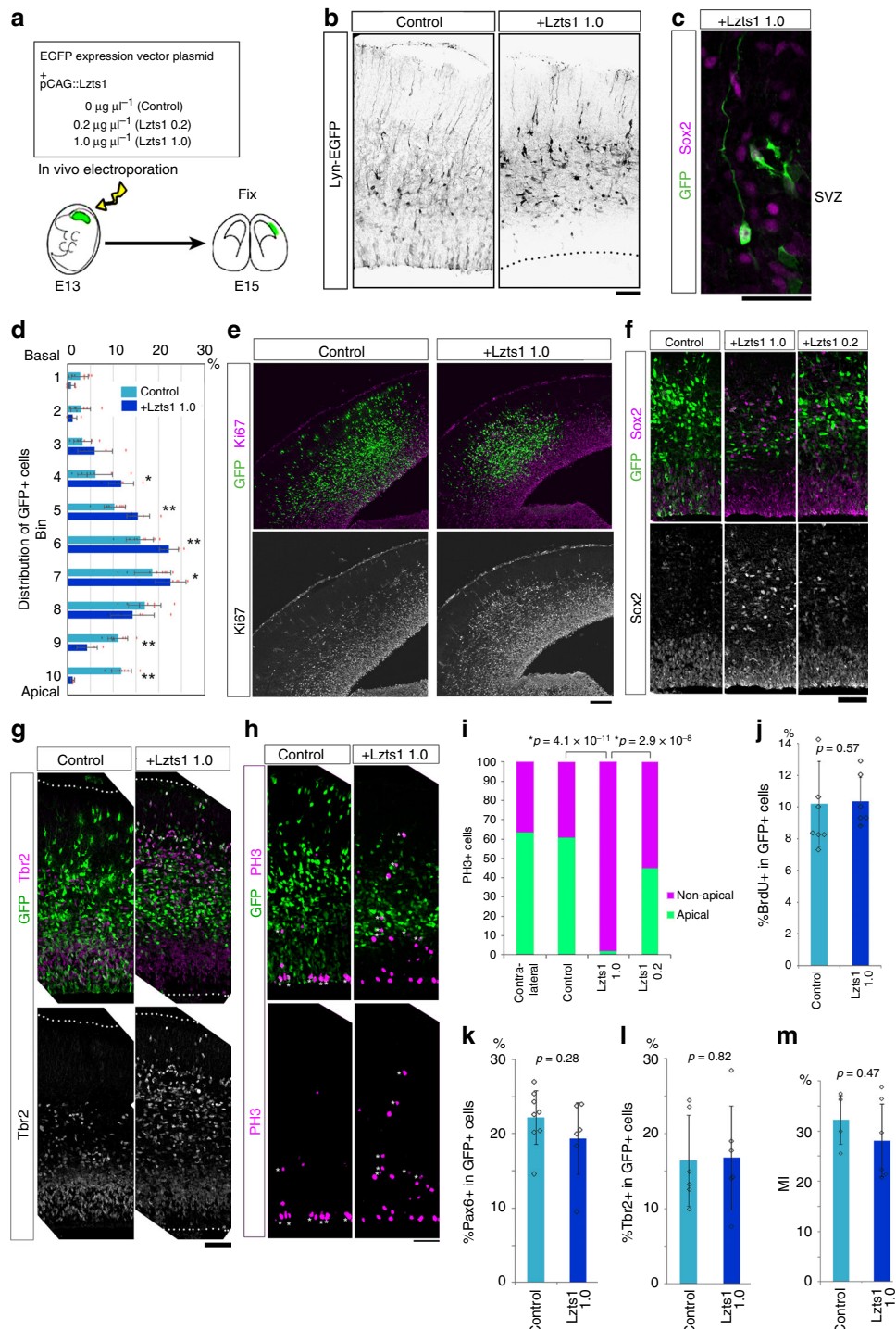

**Fig. 5** Lzts1 overexpression positions cells outside the VZ. **a** Experimental design. **b** Lzts1 overexpression (1.0 μg μl⁻¹) promotes the detachment of cells from the VZ surface. Cell morphology was visualized by Cre-mediated Lyn-EGFP (membrane-targeted form of EGFP), which was co-electroporated (2 days after E13 electroporation). **c** An example of GFP⁺ Sox2⁺ cells in the SVZ that have long basal processes, which is the typical morphology of oRGs. **d**–**m** The Lzts1 expression vector (1.0 μg μl⁻¹, indicated as "Lzts1 1.0" or 0.2 μg μl⁻¹, indicated as "Lzts1 0.2") was in vivo electroporated at E13, and E15 brain sections were examined by IHC with anti-Ki67 **e**, anti-Sox2 **f**, anti-Tbr2 **g** and **l**, anti-PH3 **h** and **i**, anti-BrdU (30 min of BrdU pulse labeling) **j**, and anti-Pax6 **k** antibodies. **d** Distribution of Lzts1-overexpressing cells in 10 bins throughout the cerebral wall. **f** Forced expression of Lzts1 at both high (1.0 μg μl⁻¹) and moderate (0.2 μg μl⁻¹) levels increased the number of Sox2⁺ GFP⁺ cells in the SVZ. **h** and **i** Lzts1 significantly increases the percentage of non-apical PH3⁺ mitotic cells in a dose-dependent manner (Fisher's exact test, two-sided, with Bonferroni-adjusted P value p < 0.05/6 = 0.0083). The asterisk in **h** indicates a PH3⁺ GFP⁺ cell. **m** The mitotic index (MI) indicates the percentage of Ki67⁺ cells among BrdU⁺ GFP⁺ cells that received BrdU 20 h before fixation. **d** N = 8 sections from 4 embryos per experiment; **i** N = 343, 56, 46, and 91 cells from 11, 9, 6, and 12 brain sections from 7, 4, 3, 4 embryos, respectively; **j** N = 8, 6; **k** N = 8, 6; **l** N = 6, 6; and **m** N = 4, 6 sections from three embryos per experiment, **d** *p < 0.05 and **p < 0.01, **d** and **j**–**m** Wilcoxon rank sum test. Bars, 50 μm in **b**, and **f**–**h**, 30 μm in **c**, 100 μm in **e**. Means ± s.d. Source data are provided as a Source Data file

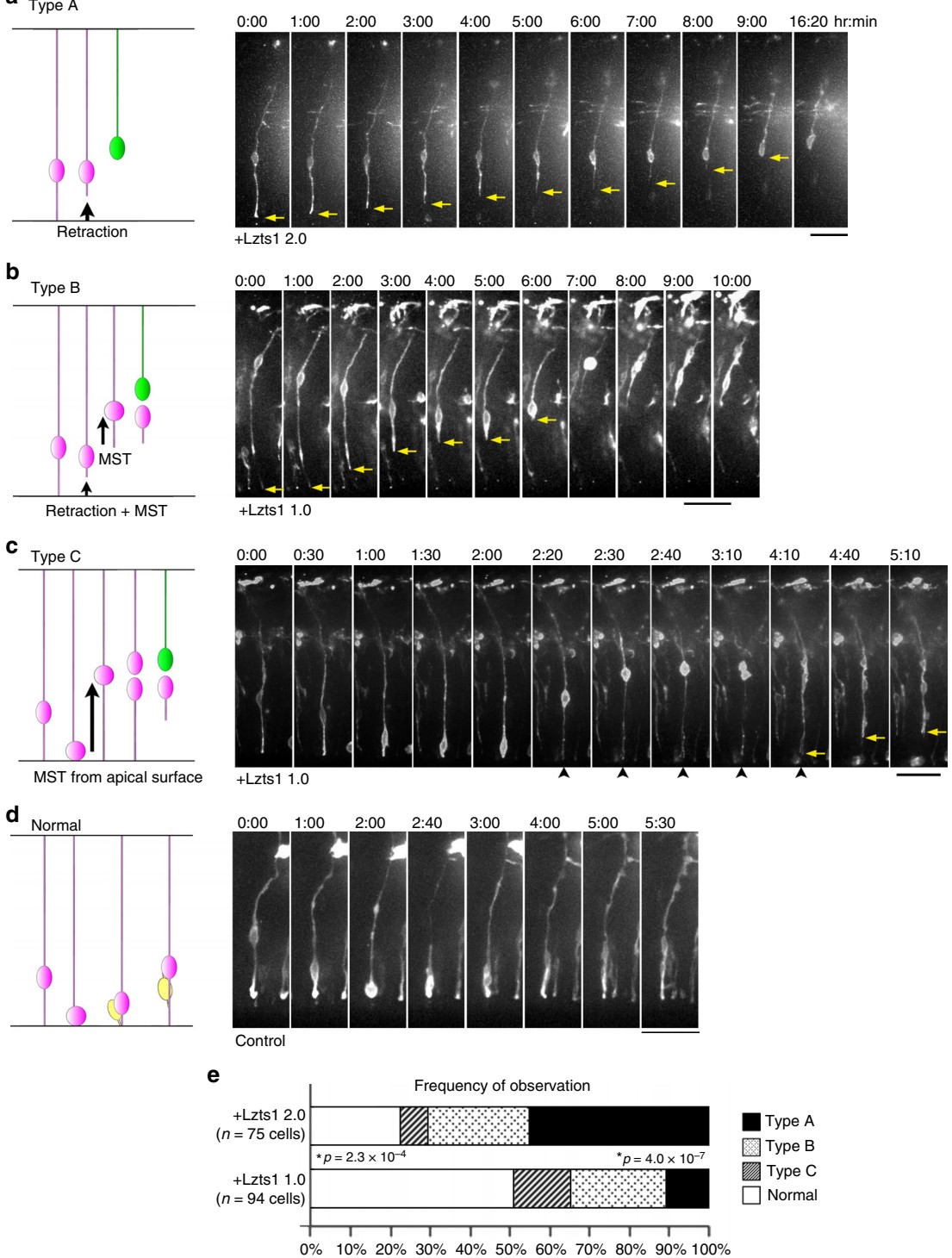

**Fig. 6** Lzts1 overexpression induces delamination and MST. Time-lapse images of Lzts1-overexpressing aRG-like cells (pCAG::Lzts1: 2.0 μg μl⁻¹ in **a**, 1.0 μg μl⁻¹ in **b** and **c** or control EGFP-expressing cells **d**. The endfeet of the apical processes are indicated by arrows. Electroporation was performed at E12, and 8–24 h later, brain slices were cultured and images of aRG-like cells were captured for 20 h. See also Supplementary Movies 1–4. **a** Image of an *lzts1*-overexpressing cell showing the retraction of its apical process without mitosis (Type A). **b** Image of an *lzts1*-overexpressing cell that retracted its apical process from the apical surface, presumably during G2 phase, and underwent MST after retraction (Type B). **c** Time-lapse images of an *lzts1*-overexpressing cell showing the "MST from the apical surface" behavior after apical INM during G2 phase (Type C). Note that the apical process was maintained during mitosis (arrowheads). **d** Time-lapse images of the aRG-like cell in the control experiment showing normal INM and apical division. In control experiments, all aRG-like cells that we observed (63 cells from five embryos) exhibited this pattern. **e** The *lzts1* plasmid concentration in the electroporation was related to the type of behaviors exhibited by aRG-like cells (N = 75 and 94 cells from six and four embryos). Type A was predominantly observed in cells electroporated with 2.0 μg μl⁻¹ pCAG::Lzts1 (see also Supplementary Figure 7) (Fisher's exact test, two-sided, with Bonferroni-adjusted P value p < 0.05/4 = 0.0125). Bars, 50 μm in **a**–**d**

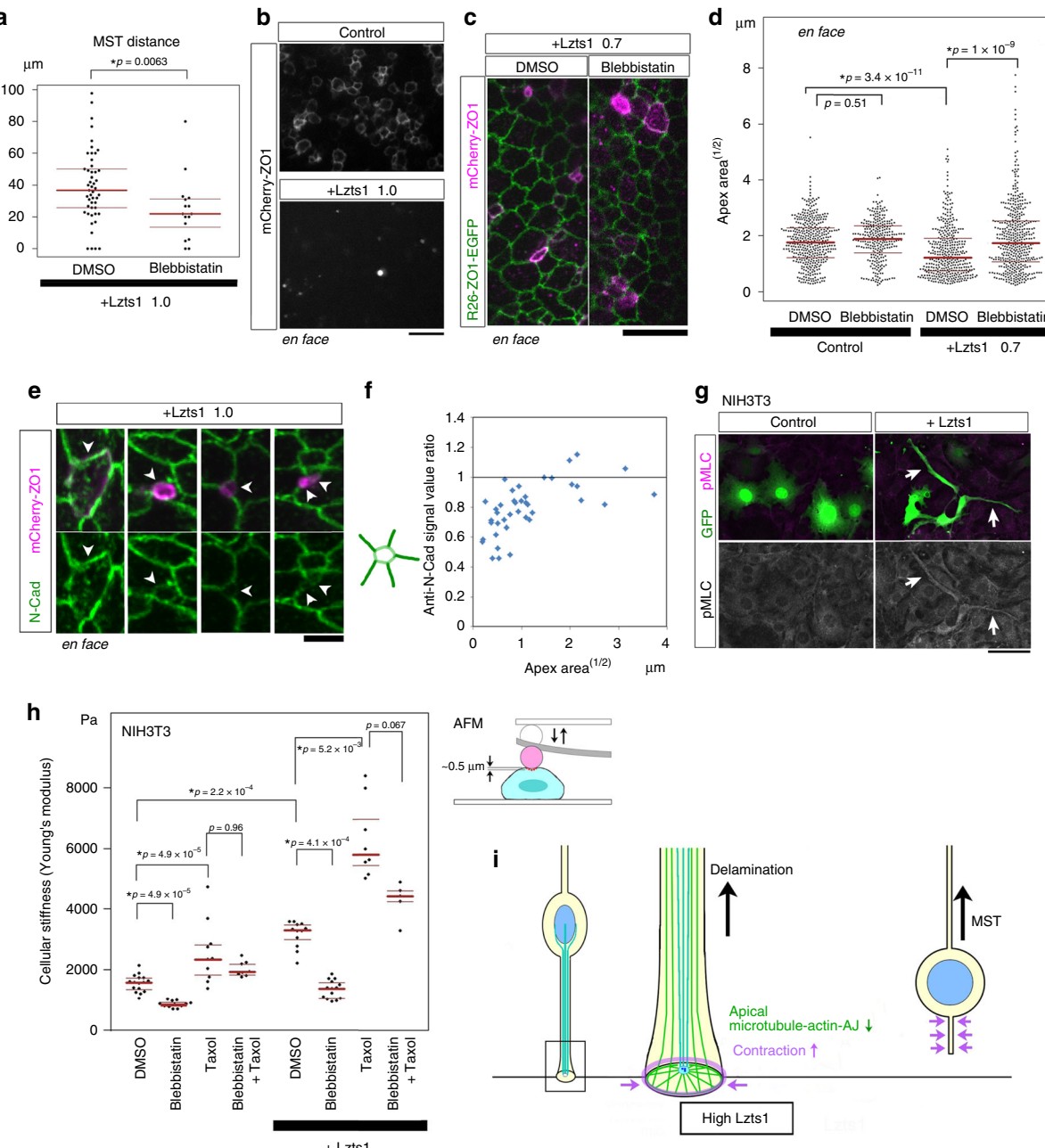

**Fig. 7** Lzts1 induces MST and apical contraction with dwonregulation of N-cadherin. **a** The distance of the MST in the SVZ induced by Lzts1 overexpression was decreased by blebbistatin, suggesting that the Lzts1-induced MST was mediated by myosin II ($N = 49$ and 16 cases, Brunner–Munzel test). **b** Lzts1 expression promotes the constriction of the apical endfoot. ZO1–mCherry with or without Lzts1 ($1.0\ \mu g\ \mu l^{-1}$) was expressed at E13 following in vivo electroporation, and after 1 day, en face observations of the apical surface were performed. **c, d** The myosin II inhibitor blebbistatin significantly decreased Lzts1-induced constriction of the apical endfoot. ZO1–mCherry with or without Lzts1 (pCAG::Lzts1: $0.7\ \mu g\ \mu l^{-1}$) was electroporated into E13 R26-ZO1-EGFP mice, and after 1 day, blebbistatin or an equal concentration of DMSO was injected into the lateral ventricle for 30 min. En face observations of the apical surface are shown. **d** $N = 380$, 364, 378, and 433 cells from three embryos per experiment. Steel–Dwass test. **e, f** N-cadherin expression at AJs was decreased along with the contraction of the AJ rings in cells overexpressing Lzts1 (arrowheads). The signals for the anti-N-cadherin antibody at the Lzts1-expressing apical endfeet were compared to the signals at the neighboring cellular junctions. $N = 40$ cells from two embryos in **f**. **g** Anti-phospho-myosin light chain 2 (pMLC) staining of control or Lzts1-expressing mouse fibroblast NIH3T3 cells (arrows). **h** Exogenous Lzts1 expression makes NIH3T3 cells stiffer than control cells by activating the actomyosin system, and this activation occurs even in the cells with the taxol-induced stabilization of microtubules. AFM indentation measurements of cellular stiffness were performed at 2 h after cells were plated. $N = 16$, 12, 10, 8, 12, 14, 8, and 5 cells. Steel–Dwass test. **i** Lzts1 controls neuronal delamination (model). In differentiating cells, high levels of Lzts1 activate the Rho–ROCK-Myosin II pathway to enhance contraction and induce MST. Lzts1 concurrently perturbs the apical microtubule-actin-AJ complex at the apical endfoot by inhibiting microtubule polymerization to ensure the rapid delamination. Bars, $10\ \mu m$ in **b**, **c**, $3\ \mu m$ in **e**, and $50\ \mu m$ in **g**. Medians with Q1 and Q3 values

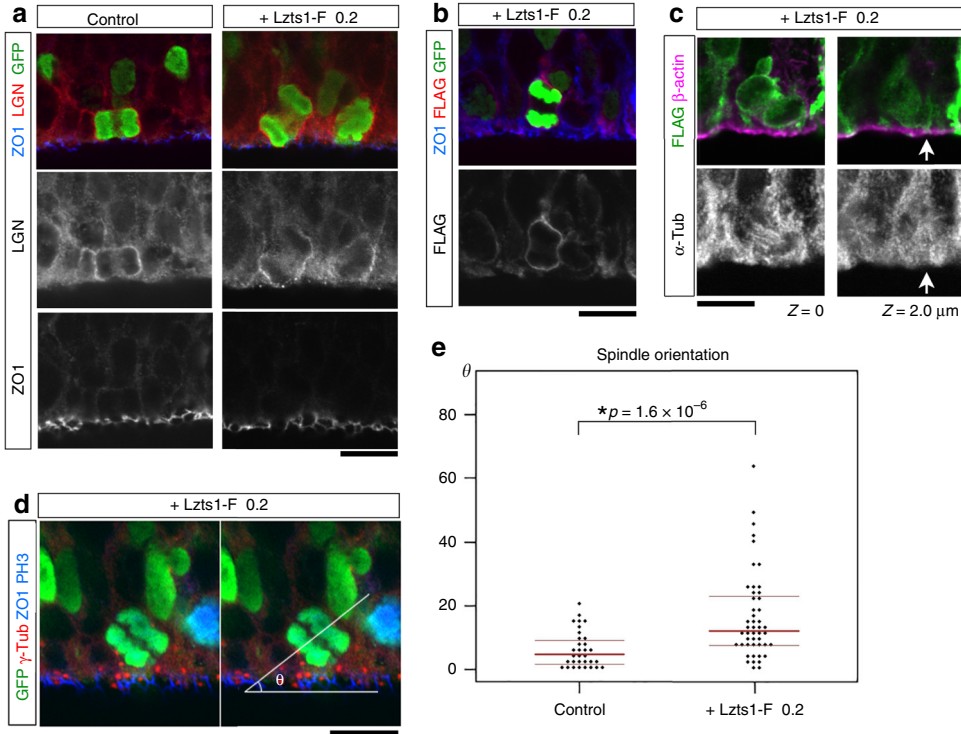

**Fig. 8** Low Lzts1 expression gives rise to oblique aRG division. In vivo electroporation was performed at E13 to induce Lzts1 expression at a low level (pCAG::Lzts1-Flag [F]: 0.2 μg μl$^{-1}$) along with H2B-GFP to visualize the DNA, and brain sections were examined at E14. **a** Triple staining for LGN, ZO1, and GFP in control or Lzts1-F-expressing cells undergoing mitosis at the apical surface. **b** Triple staining for Flag, ZO1, and GFP in Lzts1-F-expressing cells undergoing mitosis at the apical surface. **c** Triple immunostaining for α-tubulin, β-actin, and Flag in an Lzts1-F-expressing cell undergoing mitosis. Panels are shown with $z = 2.0$-μm intervals. The arrow indicates the connection to the apical surface. **d** Quadruple staining for GFP, γ-tubulin, ZO1 and PH3. **e** The number of cells displaying the oblique spindle orientation was significantly increased following Lzts1-F expression compared with control cells at the apical surface. We measured the spindle orientation $\theta$ in fixed sections **d** using a 3D-measurement method[39] ($N = 32$ and 48 cells from three and four embryos, $p = 1.6 \times 10^{-6}$, Brunner–Munzel test, medians with Q1 and Q3 values are shown) to ensure the accuracy of the values

**Functions of Lzts1 in the gyrencephalic brains**. Lzts1 is a highly evolutionarily conserved protein from mice to ferrets and humans. Because ferret has a gyrencephalic brain, and oRGs are more abundant in ferret than in mouse[10,11], the developing ferret brain is a suitable organism to study Lzts1 function in oRG generation (Fig. 10a). Ferret Lzts1 overexpression (1.0 μg μl$^{-1}$) in the E28 ferret brain by ex vivo electroporation significantly increased the percentage of Sox2$^+$ cells among SVZ EGFP$^+$ cells (Supplementary Figure 11), suggesting that Lzts1 overexpression positions ferret progenitor cells outside the VZ. Immunohistochemistry (IHC) of the E36 ferret brain revealed Lzts1 immunoreactivity in the SVZ, which included Sox2$^+$ oRG-like cells and at least some Sox2$^+$ mitotic cells (Supplementary Figure 11).

To investigate whether Lzts1 controls oRG generation in the ferret brain, we performed CRISPR/Cas9-induced KO studies. As oRG production from aRGs most frequently occurs at approximately E34[42], we expressed gRNAs for ferret *lzts1*, hCas9, and EGFP-3NLS at E32 through in vivo electroporation and examined the tissue at E38 (Fig. 10 and Supplementary Figure 12). To sustain EGFP-3NLS expression in the electroporated cells and their progenies, we took advantage of the transposon system (piggyBac)[43], in which CAG promoter-EGFP-3NLS sequence was integrated into the genome by a transposase. We generated four types of samples, i.e., "Cas9" (negative control, without gRNA), "control" (with gRNA for ferret genomic sequence not related to *lzts1*), "KO#1" and "KO#2" (each was the mixture of three different gRNAs for ferret *lzts1*). The gRNAs of KO#1 more strongly suppressed Lzts1 expression than those of KO#2 (Supplementary Figure 12).

The observed distribution of EGFP$^+$ cells suggested that Lzts1 KO delays cellular migration from the apical surface (Fig. 10b). Then, we examined whether Lzts1 KO affects oRG production, using Hes1 as a marker of undifferentiated cells. Lzts1 KO #1 significantly (and KO#2, more moderately) reduced the percentage of Hes1$^+$ cells among EGFP + cells in the OSVZ, which represent the newly generated oRGs (Fig. 10c, d). We also confirmed that these Hes1$^+$GFP$^+$ OSVZ cells were also Hopx$^{+ 44}$ and that Lzts1 KO did not reduce the expression of Hes1 in individual oRGs (Supplementary Figure 13). Based on these results, Lzts1 positively controls oRG generation in the ferret brain.

***Lzts1* is expressed in a subset of human oRGs**. A New World monkey, the marmoset, expresses Lzts1 in the developing cerebrum in a pattern similar to mice (Supplementary Figure 14), suggesting that the function of Lzts1 is evolutionarily conserved from rodents to primates.

To evaluate *lzts1* expression in human neural progenitor cells, we used two previously reported independent data sets of fluorescence-activated cell sorting-based RNA-seq[45,46]. In both data sets, *lzts1* was expressed in the aRG and oRG populations with the pattern of aRG < oRG, as well as in the neuronal (+IP) population (Supplementary Figure 15). Moreover, single-cell RNA-seq of fixed human fetal cortical cells[47] indicated that a subset of putative oRG cells (sox2$^+$pax6$^+$ hopx$^+$ tbr2$^-$ cells in the side population) expresses *lzts1* (10/63, 15.8%), although the detection of gene expression at low copy numbers may be difficult

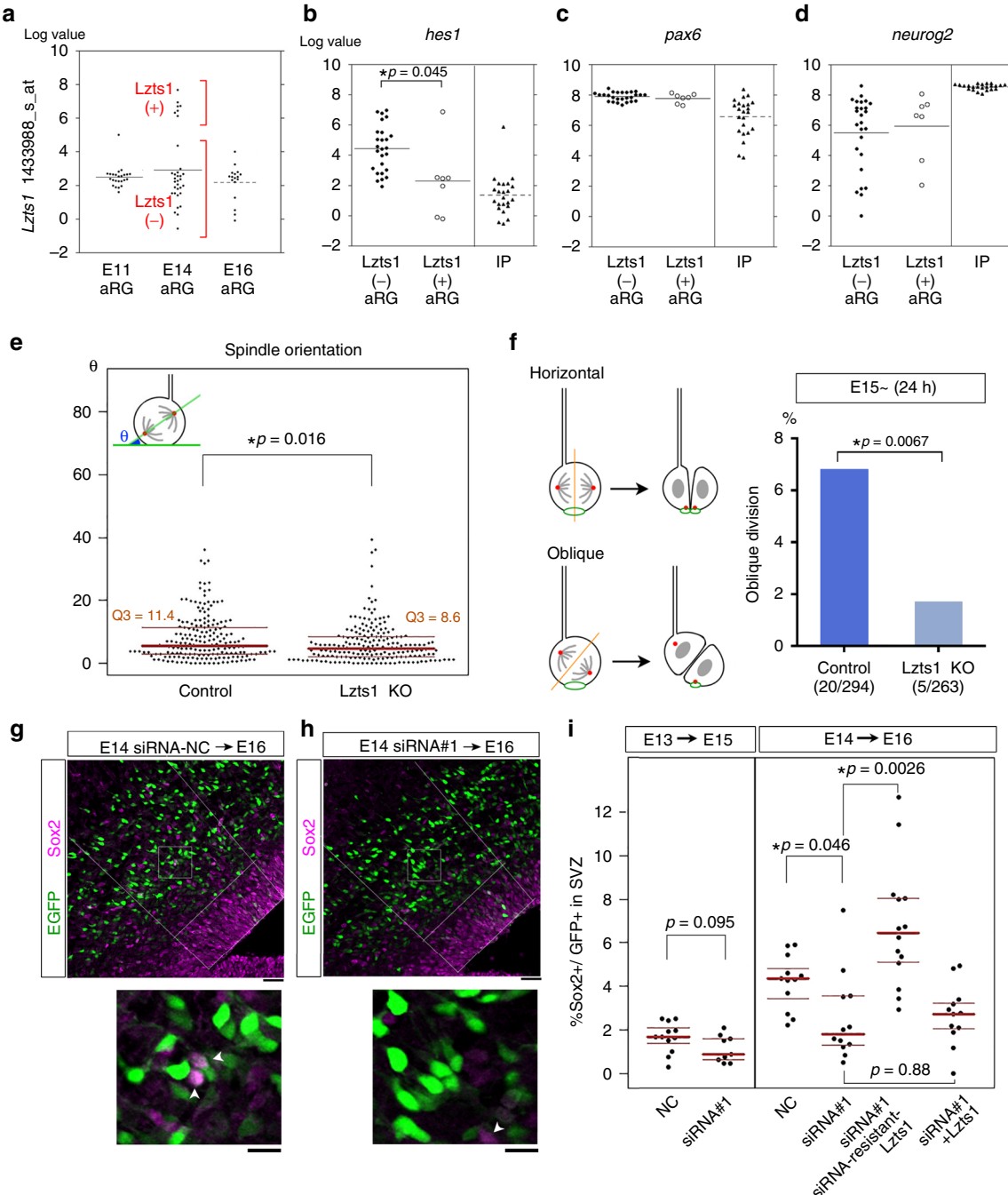

**Fig. 9** Variable expression of Lzts1 is responsible for generation of oRG-like cells. **a** *lzts1* is expressed in a subset of aRGs within the developmental time window. Single-cell transcriptome profiles[24, 25] revealed that 7/33 aRGs are *lzts1*+ (Log value > 6.0) at E14. Variances of the values are significantly different between E11 and E14, as well as between E14 and E16 aRGs (F-test, $p = 4.1 \times 10^{-8}$, $p = 0.0038$, $N = 24$, 33 and 17, respectively, bars indicate means). **b–d** Variations in gene expression in *lzts1*+ aRGs ($N = 7$) and *lzts1*− aRGs ($N = 26$). Significantly lower *hes1* expression was observed in *lzts1*+ aRGs than in *lzts1*− aRGs ($p = 0.045$), but *pax6* ($p = 0.36$) (**c**) and *neurog2* ($p = 0.68$) **d** expression were not significantly altered (permuted Brunner–Munzel test). As the references, data from IPs are also shown. Data from single-cell transcriptome profiles[24]. See also Supplementary Figure 8. **e** Lzts1 KO decreases the division angles ($\theta$) of aRGs. $N = 193$ and 193 cells from 10 and 9 embryos, respectively, Brunner–Munzel test, medians with Q1 and Q3 values. **f** Lzts1 KO reduces the oblique division of aRGs. hCas9, the guide RNA for Lzts1, ZO1-EGFP, and PACT1-mCyRFP were expressed by electroporation at E13, and after 2 days, en face imaging of the apical surface was performed to observe the aRG division pattern (images were obtained from four and three embryos, two-sided Fisher's exact test). **g–i** Lzts1 KD impairs the generation of Sox2+ GFP+ cells outside the VZ. The percentage of Sox2+ cells among GFP+ cells present in the SVZ (and IZ) was examined 2 days after electroporation. Bar, 30 µm. Arrowheads in the magnified views (**g**, **h**; Bar, 10 µm) indicate Sox2+ GFP+ cells. **i** The percentage of Sox2+ GFP+ cells in the SVZ (and IZ) was significantly reduced by Lzts1 KD following electroporation at E14, and this reduction was rescued by the introduction of the siRNA-resistant Lzts1, but not wild-type Lzts1 ($N = 12$, 9, 12, 12, 14, and 12 sections from 4, 3, 4, 4, 5, and 4 embryos, respectively; medians with Q1 and Q3 values are shown; Wilcoxon exact test at E13 and Steel–Dwass test at E14). Source data are provided as a Source Data file

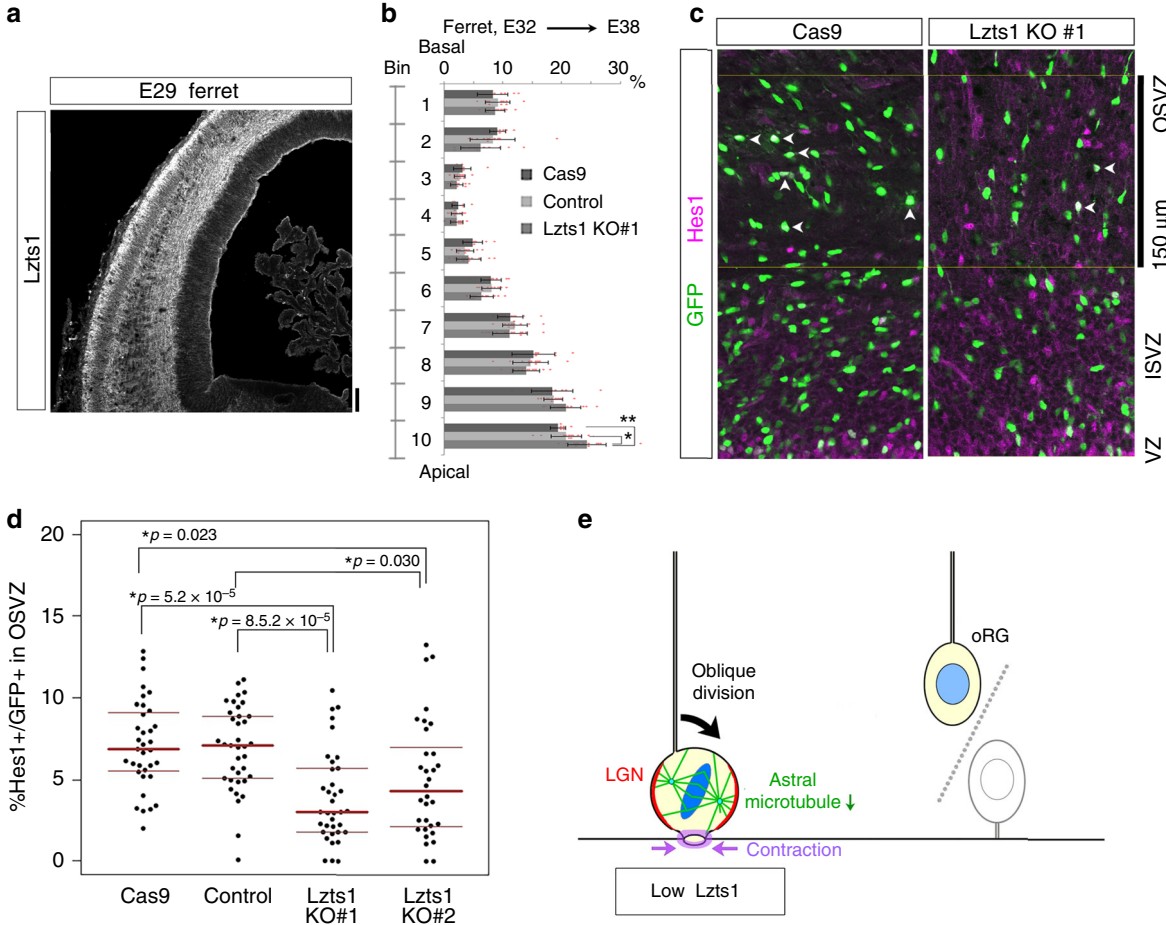

**Fig. 10** Lzts1 controls oRG generation in the gyrencephalic brain. **a** IHC for Lzts1 in an E29 ferret dorsal forebrain. Bar, 100 μm. **b–d** CRISPR/Cas9-induced disruption (KO) of *lzts1* in the ferret brain. hCas9 and the gRNA for ferret *lzts1* were co-expressed with EGFP by in vivo electroporation at E32, and brains were examined at E38. See also Supplementary Figure 12. **b** Lzts1 KO retards cellular migration from the apical surface. Distributions of Cas9 only (no gRNA), negative control gRNA or Lzts1-gRNA (KO#1) electroporated cells in 10 bins separating the cerebral wall were determined ($N = 11, 11,$ and 13 hemispheres, respectively, mean ± s.d., Steel–Dwass test, *$p = 0.023$, **$p = 5.5 \times 10^{-4}$). **c** and **d** Lzts1 KO (KO#1, and KO#2 more moderately) reduced the percentage of Hes1+ cells among the electroporated cells in the OSVZ (we examined a 150-μm-depth area from the basal side of ISVZ, approximately corresponding to Bin 8 and the apical half of Bin 7) ($N = 35, 37, 36,$ and 32 sections from 16, 20, 18, and 14 hemispheres, respectively; Steel–Dwass test, medians with Q1 and Q3 values are shown). Bar, 150 μm. Source data are provided as a Source Data file. **e** Low Lzts1 expression in aRGs induces the oRG-producing division (model). Low levels of Lzts1 are sufficient to inhibit astral microtubule–LGN–AJ anchoring in M-phase cells to induce oblique division

using RNA-seq of fixed single cells. These results suggest that *lzts1* is expressed in at least in a subset of human oRGs and may affect their morphological dynamics.

## Discussion

In this study, we demonstrate that Lzts1 positively regulates neuronal delamination, MST, and oblique aRG division to generate oRGs in an expression level-dependent manner. Our data support the hypothesis that neuronal delamination and oRG generation are two aspects of the same process, continuously variable cellular dynamics controlled by Lzts1 (Supplementary Figure 16).

In this report, we show that Lzts1-induced apical contraction by activating the actomyosin system. However, the activation of the actomyosin system is insufficient to evoke delamination from AJs[16]. Lzts1 is associated with microtubule components and is involved in microtubule assembly; moreover, the purified Lzts1 protein without PKA phosphorylation inhibits mitogen-activated protein 2-induced tubulin polymerization[27]. Thus, we hypothesize that Lzts1 induces neuronal delamination through its inhibitory effect on microtubules concurrent with the activation of the actomyosin system.

In epithelial cells, the minus ends of noncentrosomal microtubules are anchored at the adhesion belt of the cadherin-based AJs and regulate AJ integrity[48]. In addition, a centrosome-nucleated wheel-like microtubule configuration aligned with the apical actin cable maintains AJs of the neuroepithelium[15,16]. Therefore, high levels of Lzts1 expression possibly perturb this cytoskeletal configuration at the apical endfoot via an inhibitory effect on microtubules. We speculate that this modulation would disintegrate AJs of differentiating cells with the downregulation of N-cadherin.

When Lzts1 expression is silenced, the departure of differentiating cells from the apical surface is retarded, but not completely inhibited. This result indicates that other mechanisms, such as *scrt1/2*-mediated and *foxp*-mediated downregulation of cadherin[17,37], *Insm1*-mediated repression of AJ ring protein[49], contraction of the F-actin ring[16] and passive forces from neighboring crowding cells[50], also regulate the departure of differentiating cells in conjunction with Lzts1. Among the molecules related to neuronal delamination, only Lzts1 shows a unique

expression pattern that is clearly confined to the AJs of cells that will delaminate. Furthermore, *lzts1* expression is rapidly upregulated during neuronal differentiation[24]. Thus, Lzts1 is responsible for the quick cellular departure during neuronal differentiation.

Our results suggest the spindle orientation of aRGs may be controlled by a mechanism similar to that of neuronal delamination in the context of regulation of the microtubule-AJ complex. Consistent with our observations, Btg2::GFP+ neuronal progenitors, which express *lzts1*[24,51], show more dynamic deviations in the spindle orientation than proliferating progenitors[52]. Interestingly, these Btg2::GFP+ mitotic progenitors have relatively small astral microtubules[52]. As astral microtubules are sensitive to inhibitors of microtubule polymerization owing to their highly dynamic nature, low Lzts1 expression might be sufficient to perturb the formation of astral microtubules and inhibit the microtubule–LGN–AJ interaction[52], resulting in various spindle orientations (Fig. 10e). In addition, the moderate apical contraction induced by Lzts1 might also be involved in oblique division. Future investigations will determine how Lzts1 coordinates dynamic cytoskeletal remodeling to regulate both neuronal delamination and mitotic spindle orientation.

As Lzts1 expression is induced by neurogenin, the unified regulation of cell departure, i.e., neuronal delamination and oRG generation, by Lzts1 ensures the generation of oRGs with neurogenic potential within the time window of neurogenesis[6,7,42]. The common molecular features shared between neuronally differentiating cells and oRGs have been reported in a previous study using RNA-seq analysis[45], in which the activation of Neurog2 induces the generation of oRG-like cells in the ferret embryonic brain.

Even at the peak timing for oRG generation, the majority of aRGs continue to undergo horizontal division[42]. The regulatory mechanisms that evoke oblique division in only a subset of aRGs are still obscure[23]. In the present study, a subset of aRGs expressed *lzts1* but did not exhibit apparent differences other than in the expression of *hes1*. Therefore, we propose that E14 aRGs express *lzts1* only transiently, giving rise to low Lzts1 expression in a limited number of aRGs at a particular time point. This subset of aRGs will give rise to the oblique division that produces oRGs. This phenomenon in which variable expression of a molecule that is typically expressed at low levels induces an irreversible change in cell characteristics might represent a useful strategy to build a diverse cellular system, as proposed by a previously reported mathematical model[53]. According to our results, the modulation of the mitotic orientation is a trigger to enhance small differences among the aRG population and subsequent generate oRGs as a new cell population.

We found that forced expression of Lzts1 in the mouse brain gives rise to MST from the apical surface. Interestingly, a similar cell behavior has been reported in ferrets and humans[22], but not in mice. These results raise the intriguing possibility that Lzts1 expression or function is enhanced in ferret and human neural progenitor cells and might explain why the frequency and distance of MST and oRG generation are greater in human and ferret brains than in mouse brains[5]; these processes contribute to the evolutionary expansion of the neocortex.

In humans, *lzts1* is located on chromosome 8p22; the 8p region is associated with susceptibility to psychiatric disorders and cancer[54]. These psychiatric and neurodevelopmental disorders have been explained by impaired synaptic transmission from glutamatergic neurons, as suggested by previous studies in Lzts1 KO mice[55]. Moreover, the disorganized production of oRGs induced by altered Lzts1 function might be involved in psychiatric disorders. Studies of 3D-organized developing brain tissues, such as ES-derived human cerebral organoids[56], would provide useful information about whether and how Lzts1

contributes to oRG generation and the formation of the cerebral architecture.

## Methods

**Animals**. CD1 mice (Crlj:ICR and Slc:ICR) were used throughout the mouse experiments. Tbr2::EGFP (Eomes::EGFP) BAC transgenic mice (Fig. 1e) were generated by the GENSAT Project[30], NINDS Contract #N01NS02331 to Rockefeller University (New York, USA). The generation and characterization of Gadd45g::d4Venus transgenic mice has been reported[31]. R26-ZO1-EGFP mice (Accession No. CDB0260K, http://www2.clst.riken.jp/arg/reporter_mice.html) were used to visualize the localization of ZO1[57]. Ferrets were purchased from Marshall Bioresources (New York, USA). To time animal pregnancies, we defined the date when a vaginal plug was observed as E0. The sex of the embryos used was not examined. All animal experiments were performed in accordance with institutional guidelines.

**Plasmid DNA**. The primers used to construct plasmid DNAs are listed in Supplementary Table 1. To generate pCAG::Lzts1, the mouse *lzts1* cDNA (AF288601.1) was cloned from cDNAs obtained from the dorsal forebrain of E14 CD1 into the TOPO-blunt II vector (Thermo Fisher Scientific, MA, USA) using the primers kzk-Lzts1-F1 and Lzts1-B1, and the EcoRI fragment of the *lzts1* cDNA was subcloned into pCAGGS. pCAG::furoLzts1-F was generated by cloning the ferret *lzts1* cDNA without a stop codon (XM_004763331.1, 110–2087) into pCAG-FlagN1 via a SalI/AgeI site using the primers FuroLz-F1 and R1, F2 and R2, F3 and R3.

To generate pCAG::Scrt1, the mouse *scrt1* cDNA was cloned from mouse brain extracts by RT-PCR using the primers kzk-Scrt1-F and Scrt1-R. To generate pCAG::Neurog1, the *neurog1* cDNA (BC062148) was obtained from MGC clones. These cDNAs were subcloned into expression vectors with a CAG promoter. pEF::Neurog2 was previously generated[31] using EF1α-promoter expression vectors (pEF-BOS)[58].

pCAG::siRNA-resistant-Lzts1, which expresses mouse Lzts1 including 4 point mutations with no amino acid changes within the siRNA#1 target sequence, was generated by PCR from a pCAG::Lzts1 template using the primers kzk-Lzts1-mut-F1 and Lzts1-B1, and the fragment was cloned into the TOPO-blunt II vector (Thermo Fisher Scientific). Next, the EcoRI fragment of mutated-*lzts1* cDNA was subcloned into pCAGGS.

To generate pCAG::Lzts1-F, the mouse Lzts1 cDNA lacking a stop codon was cloned into the pCAG-Flag-N1 vector modified pEGFP-N1 (Thermo Fisher Scientific) using the primers kzk-Lzts1-F1 and KpnI-B3-Lzts1. To generate pCAG::G2A-Lzts1-F and pCAG::ΔC-Lzts1-F, the G2A mutation in the myristoylation consensus sequence of the mouse *lzts1* cDNA lacking a stop codon or the *lzts1* cDNA lacking the C-terminal half (1–255 a.a.) was generated by PCR using the primers kzk-F4-G2A and KpnI-B3-Lzts1, kzk-Lzts1-F1, and Lzts1-B765, respectively, and cloned into pCAG-Flag-N1.

pCAG::H2B-EGFP or pBA-LPL-H2B-EGFP was used to visualize DNA[19], pEF::LPL-Lyn-EGFP was transfected with a low concentration of pEF::Cre or pCAG::Cre to visualize the morphology of electroporated cells[19,50], and pCAG::EGFP-3NLS[40] or pCAG::mCherry3NLS[25] was used to visualize nuclei. pCAG::EGFP-ZO1 or pCAG::mCherry-ZO1 was used to visualize AJs. The pPB-CAG-PACT-mCyRFP1 vector used to visualize centrosomes, pPB-CAG-EGFP-ZO1 vector used to visualize AJs, and pPB-CAG-2EGFP-3NLS vector used to visualize the nucleus were generated by subcloning CAG-PACT-mCyRFP1, CAG-EGFP-ZO1, and CAG-2EGFP-3NLS into the piggyBac donor vector pPB-LR5[59] (Sanger Institute, Cambridgeshire, UK), respectively. mCyRPF1 ORF was obtained from Addgene (#84545). pCAG-hyPBase generated from pCMV-hyPBase[43] (Sanger Institute) was used to express the hyperactive piggyBac transposase.

**Knockdown experiments**. Stealth RNAi (Thermo Fisher Scientific) was used for Lzts1 knockdown experiments. The targeting sequences were as follows: Lzts1-siRNA#1, 5'-AACACUGUGGCCUGAGAUAAGGCUGC-3' (MSS210114) and Lzts1-siRNA#2, 5'-UAAUUGGGACUCUUUGAGCUGUUGC-3' (MSS210113). The negative control Hi GC siRNA was used as a control (NC-siRNA). For in vivo electroporations, 200 μM RNAi solution in normal saline was prepared as a stock solution.

**CRISPR/Cas9-induced Lzts1 KO**. The guide RNAs (gRNAs) against target genes (mouse Lzts1: GCCGTTTTTGGGGGTTCAGCT; ferret Lzts1 KO#1: CCGGGCT TCACGATACAAGT, TAGTGGGGACATAGGCGGCC, and GGAGAGTTC TCTGCGTACCA; ferret Lzts1 KO#2: GAGCTCAATCGGTATTCAGA, CAAG CTCAGGTCCTACGAGAAG, and CCCACCAGCCGTTTTGGAGGCTC; ferret negative control [intron of GDF5]: TCCCGGGTTGACGATAAAAAT) were designed using the Zhang lab website (http://crispr.mit.edu)[60]. The gRNA fragment was amplified by PCR through self-amplification of the designed primer set and cloned into AflII-digested gRNA vectors (gRNA backbone-YT210)[61], which were modified from the original vector generated by the Church lab[62]. The primers used to construct gRNAs are listed in Supplementary Table 2. A mixture of gRNA vectors, pCAX::hCas9, and pCAG::EGFP(3NLS), pCAG::Strawberry, or pPB-CAG-2EGFP-3NLS with pCAG-hyPBase, was transfected into the embryonic brain by in vivo electroporation[35] (Supplementary Table 3).

**In vivo and ex vivo electroporation**. For in vivo (in utero) electroporation[50,63], the abdomen of an anesthetized pregnant CD1 or Tg mouse was dissected with fine scissors, and the uterine horns were exposed. A DNA solution containing 0.005% Fast Green FCF (Wako Pure Chemical Industries, Osaka, Japan) was injected into the lateral ventricle using a pulled glass capillary. When ex vivo electroporation was performed (Supplementary Figure 11), the DNA solution was injected into the lateral ventricle of the head of an E28 ferret embryo in HEPES. The head of the embryo was then placed between the discs of a forceps-type electrode (3-mm disc electrodes, CUY650P3; NEPA GENE, Chiba, Japan), and electric pulses (35 V for E13 mice; and 53 V for E28 ferret) were discharged four times, resulting in gene transfection into the cerebral wall. For the in vivo electroporation experiments in ferrets[61] shown in Fig. 10 and Supplementary Figure 12, E32 pregnant ferrets were anesthetized with isoflurane, the embryonic brain hemispheres were injected with 2 µl of the DNA solution along with 0.0002% Fast Green FCF, and then the embryos were placed between the paddles of the 10-mm electrodes and subjected to 5 × 100 ms 45 V electric pulses. For in vivo electroporation, the uterus was immediately returned to the abdominal cavity, and the wall and skin of the abdominal cavity were sutured. The final siRNA, gRNA and plasmid DNA concentrations used for electroporation are listed in Supplementary Table 3. We confirmed a high co-electroporation efficiency (> 98% overlap) using our co-electroporation protocol[25].

**Immunohistochemistry**. Brains were fixed in 4% paraformaldehyde (PFA), 1% PFA (for anti-LGN and anti-Hes1 staining), PLP fixative, or 3% v/v glyoxal solution[64] (pH 4, for anti-N-cadherin staining); immersed in 20% sucrose or 30% sucrose (E38 ferret); embedded in OCT compound (Miles, Elkhart, IN, USA); frozen and sectioned. Frozen sections, fixed-brain tissues or fixed cells were immunostained with rabbit anti-Lzts1 pAb (HPA006294, Sigma-Aldrich, 1:800 [Fig. 1c]) (20878-1-AP, 0.36 mg ml⁻¹, 1:1000, Proteintech Group, Rosemont, USA; recommended), mouse anti-ZO1 mAb (33–9100, 0.5 mg ml⁻¹, 1:200, Life Technologies), mouse anti-BrdU mAb (B2531, 4.4 mg ml⁻¹, 1:200, Sigma-Aldrich), rat anti-BrdU mAb (NB500-169, 0.5 mg⁻¹, 1:300, Novus Biologicals), chicken anti-GFP pAb (GFP-1020, 10 mg ml⁻¹, 1:1000, Aves Labs, Tigard, USA), rabbit anti-GFP pAb (598, 1:1000, MBL, Nagoya, Japan), rat anti-GFP mAb (GF090R, 1:500, Nacalai Tesque, Kyoto, Japan), rabbit anti-Tbr2 pAb (ab23345, 0.5 mg ml⁻¹, 1:300, Abcam, Cambridge, UK), rabbit anti-Tbr2 mAb (ab183991, 0.643 mg ml⁻¹, 1:1000, Abcam), rabbit anti-Sox2 pAb (ab97959, 1 mg ml⁻¹, 1:500, Abcam), mouse eFluor 660-conjugated anti-Sox2 mAb (50-9811-82, 0.2 mg ml⁻¹, 1:800, Thermo Fisher Scientific), rabbit anti-Pax6 pAb (PRB-278P, 2 mg ml⁻¹, 1:500, Covance, Princeton, USA), rabbit anti-Pax6 mAb (ab195045, 2 mg ml⁻¹, 1:400, Abcam), rat anti-PH3 mAb (HTA28, ab10543, 1:400, Abcam), rabbit anti-PH3 pAb (06–570, 1 mg ml⁻¹, 1:300, Sigma-Aldrich), mouse anti-Ki67 mAb (NCL-L-Ki67-MM1, 1:50, Leica Biosystems, Wetzlar, Germany), rabbit anti-RFP pAb (PM005, 1:1000, MBL), rat anti-RFP mAb (5F8, 1 mg ml⁻¹, 1:200, Chromo Tek, Planegg, Germany), rabbit anti-Scrt1 pAb (HPA045265, 0.1 mg ml⁻¹, 1:400, Sigma-Aldrich), rabbit anti-N-cadherin pAb (M142, 2.0 mg ml⁻¹, 1:400, TAKARA Bio Inc, Kusatsu, Japan), mouse anti-phospho-myosin light chain 2 mAb (S19, 1:200, Cell Signaling Technology, Danvers, USA.), rabbit anti-LGN pAb (1:1000)[40], rabbit anti-γ-tubulin pAb (T5192, 1:500, Sigma-Aldrich), goat anti-γ-tubulin pAb (sc-7396, 0.2 mg ml⁻¹, 1:1000, Santa Cruz Biotechnology), mouse anti-α-tubulin mAb (clone DM1A, T6199, 1 mg ml⁻¹, 1:1000, Sigma-Aldrich), mouse anti-β-actin mAb (clone AC-74, A2228, 2 mg ml⁻¹, 1:500, Sigma-Aldrich), rat anti-Hes1 mAb (NM1, 1.0 mg ml⁻¹, 1:800, MBL), rabbit anti-Flag pAb (DYKDDDK Tag antibody, PA1-984B, 1 mg ml⁻¹, 1:1000, Thermo Fisher Scientific), rabbit anti-HOPX pAb (FL-73, sc-30216, 1:800, Santa Cruz Biotechnology), rabbit anti-Tenascin C mAb (ab108930, 0.4 mg ml⁻¹, 1:800, Abcam), and mouse anti-PTP zeata mAb (ab126497, 0.3 mg ml⁻¹, 1:500, Abcam).

Secondary antibodies were conjugated to Alexa Fluor 488, 546, 555, or 647 (A11039, A21206, A10043, A31573, A21202, A110536, A31571, A11056, A21208, and A21434; 2 mg ml⁻¹, 1:1000, Thermo Fisher Scientific). A Zenon Alexa Fluor 647 Rabbit IgG labeling kit (Z25308, Thermo Fisher Scientific) was used to label anti-Tbr2 antibodies in Fig. 1 and anti-Sox2 antibodies in Supplementary Figure 11. Antigen retrieval by Histo-VT (Nacalai Tesque) or 5 N HCl was performed before staining as needed. Immunostained sections were imaged on a laser-scanning confocal microscope (FV1000, Olympus, Tokyo, Japan). In the analyses of Sox2⁺ GFP⁺ cells outside the VZ (Fig. 9g–i), we examined GFP⁺ cells in the distinct area (> 100 µm from the apical surface and a 150-µm-wide area from the angle at E15; > 70 µm and 150 µm at E16).

**Immunoelectron microscopy**. For immunogold EM labeling, cerebral hemispheres were fixed with 4% paraformaldehyde for 3 h at 4 °C, immersed in a graded series of sucrose solutions (10%, 15%, and 20%), and embedded in OCT compound. Frozen sections (10 µm thick) were cut on a cryostat and mounted on silane-coated slides. After blocking, sections were incubated with the primary antibody (20878-1-AP, 1:1000, rabbit anti-Lzts1 pAb, Proteintech Group) in PBS overnight at 4 °C. Sections were incubated with 1.4-nm gold-coupled goat anti-rabbit Fab fragment secondary antibodies (7204, 1:100, Nanoprobes Inc, NY, US) at room temperature for 2 h and postfixed with 2.5% glutaraldehyde, followed by gold enhancement of the immunogold particles using GoldEnhance EM (Nanoprobes). After treatment with 0.5% OsO₄, the sections were stained with 1% uranyl acetate, dehydrated and embedded in Epon 812. Ultrathin sections (70-nm thick) were cut using a diamond knife (DiATOME, Hatfield, US) on an ultramicrotome (Leica EM UC7), collected on formvar-coated single-slot

copper grids, stained with lead citrate, and examined under a JEM-1400Plus electron microscope (JEOL, Tokyo, Japan).

**Cultures and live observation of cell behaviors**. For cross-sectional and en face cultures[19,50], brain walls and slices were mounted in a type Ia collagen gel (Koken, Tokyo, Japan) with growth media: DMEM/F12 containing N2 supplement (1:100, Thermo Fisher Scientific), B27 supplement without vitamin A (1:50, Thermo Fisher Scientific), 5% FBS (v/v), 5% HS (v/v), 10 ng ml⁻¹ EGF, and 10 ng ml⁻¹ FGF. Time-lapse confocal microscopy was performed using an upright CSU-X1 microscope (Yokogawa, Musashino, Japan) equipped with an iXon + CCD camera (Andor, Belfast, UK) with an on-stage chamber filled with 5% CO₂ and 40% O₂ or a CV1000 system (Yokogawa) with 5% CO₂. In Fig. 6 experiments, since we very sporadically visualized aRGs to obtain clear images, only a small fraction of Lzts1-overexpressing VZ cells was monitored. For inhibitor experiments, 10 µM (−)-blebbistatin (6.8 mM stock in DMSO, Merck Millipore) or an equal volume of DMSO was added to the growth medium before imaging began.

For time-lapse analysis of apical junction inheritance (Fig. 9f)[19], the dorsal part of cerebral walls was immersed in type Ia collagen and then transferred to a circular 30-mm membrane filter (Merck Millipore, Darmstadt, Germany). After a 15-min incubation at 37 °C, growth medium was added, and the sample was observed under an inverted confocal microscope (FV1000, FV1000 MPE, Olympus: A1R MP+, Nikon, Tokyo, Japan) with an on-stage chamber filled with 5% CO₂ and 40% O₂.

**En face analysis of the AJ ring**. For the in vivo administration of myosin II inhibitor (Fig. 7c, d), E13 R26-ZO1-EGFP mouse embryos were electroporated with pCAG:: mCherry-ZO1 and pCAG::Lzts1 (0.7 µg µg⁻¹) in vivo. After 24 h, 20 µM (−)-blebbistatin (6.8 mM stock in DMSO, Merck Millipore) or an equal volume of DMSO in normal saline was injected into the lateral ventricles of the mouse embryos. At 30 min after the injection, the embryos were removed and brain tissues were fixed with 4% PFA. For analysis of the effect of Lzts1 KO (Fig. 4), the guide RNA for Lzts1, hCas9 and dsRed were expressed in Gadd45g::Venus Tg mouse embryos at E13, and the brain tissues were fixed at E15. After rinses with PBS, tissues were stained with a rabbit anti-RFP pAb (PM005, 1:1000, MBL), mouse anti-ZO1 mAb (33–9100, 1:200, Life Technologies), and rat anti-GFP pAb (GF090R, 1:500, Nacalai Tesque). En face images were captured with an inverted confocal microscope (FV1000, Olympus) (Figs 4, 7c, e) or an upright confocal microscope (CSU-X1, Yokogawa) (Fig. 7b), and the inner area surrounded by the ZO1⁺ AJ ring was measured by ImageJ software. We obtained the area (µm²) of the apical endfeet by ImageJ software, and calculated the square root of the area value as the apex area^(1/2) (µm), which was assumed to be proportional to the circumference of each AJ ring.

**En face analysis of anti-Lzts1 and anti-N-cadherin signals**. For the estimation of signals from anti-Lzts1 or anti-N-cadherin immunoreactivity (Figs 1i and 7f), the average values were calculated from the values obtained from 10 points of a cellular junction per cell. For the estimation of anti-N-cadherin signal ratio, the signal values were compared with the average values obtained from the neighboring cellular junctions (five points per cellular junction). Signal values and the area of the apical endfoot were obtained by using ImageJ software.

**Analysis of anti-Lzts1 signals to confirm KO efficiency**. The ferret brain samples that had been electroporated at E32 were fixed at E38, cryosectioned, and subjected to IHC with anti-Lzts1 (20878-1-AP, 1:1000, Proteintech Group) and anti-GFP (GFP-1020, 1:1000, Aves Labs) antibodies. Confocal images (FV1000) were examined by ImageJ software to obtain the Lzts1 expression ratio, which was calculated by dividing the maximum fluorescence intensity of Lzts1 signals at cellular junctions between two GFP⁺ cells in the ISVZ by the mean value of the maximum fluorescence intensity at the neighboring five cellular junctions (Supplementary Figure 12).

**Immunocytochemistry and anti-Flag/GFP signal ratio**. We co-transfected pCAG::lzts1-F (2.0 µg per dish) and pCAG::EGFP (0.5 µg per dish) into NIH3T3 cells (from RIKEN BRC) cultured in medium (DMEM/F12 with 10% FBS) in a 35-mm dish using 3.75 µl of the Lipofectamine 3000 reagent (Thermo Fisher Scientific). After two days, cells were fixed with 4% PFA containing 1% TCA, and stained with mouse anti-phosho-myosin light chain 2 antibody (S19, 1:200, Cell Signaling Technology) and chick anti-GFP antibody (GFP-1020, 1:1000, Aves Labs) (Fig. 7g).

To evaluate the ratio of the anti-Flag/GFP signals shown in Supplementary Figure 7, E12 mouse embryos were co-electroporated with different concentrations of pCAG::EGFP and pCAG::lzts1-F, and after 1 day, cerebral cells were dissociated and cultured in DMEM/F12 with N2 and B27 supplement. After 1 h, the cells were fixed with 4% PFA, immunostained with anti-Flag (PA1-984B, 1:1000, Thermo Fisher Scientific) and anti-GFP antibodies (GFP-1020, 1:1000, Aves Labs), and images were obtained using an FV1000 microscope. The integrated values for the anti-Flag and anti-GFP fluorescent signals in a defined area in each cell were determined using ImageJ software, and the Flag/GFP ratio was calculated.

**AFM indentation measurements of NIH3T3 cell stiffness**. We co-transfected pCAG::Lzts1-F (2.0 µg per dish) and pCAG::EGFP-3NLS (0.5 µg per dish) into

NIH3T3 cells cultured in medium (DMEM/F12 with 10% fetal bovine serum; FBS) in a 35-mm dish using 3.75 µl of Lipofectamine 3000 reagent (Thermo Fisher Scientific). We evaluated the effect of Lzts1 on cellular stiffness by calculating the difference in Young's modulus [Pa] (Fig. 7h). Two days after transfection, cells were treated with 0.05% trypsin-EDTA (Thermo Fisher Scientific) and resuspended in medium; 2 h after the cells were plated on a PEI-coated dish, the positions of EGFP$^+$ (Lzts1$^+$) and EGFP$^-$ (Lzts1$^-$) cells were recorded using an inverted fluorescence microscope (IX71, Olympus), and AFM indentation measurements[38] were initiated when no clear difference in the morphology of EGFP$^+$ (Lzts1$^+$) cells and EGFP$^-$ (Lzts1$^-$) cells was observed. All measurements were performed with a Cellhesion200 (JPK Instruments, Berlin, Germany) mounted on an IX71 inverted microscope (Olympus) equipped with a cantilever and a borosilicate bead (sQUBE, 5 µm diameter), with 1 nN of applied force and 1 µm/s of approach and retraction velocities. Each measurement point was set at the top of each cell. Force–distance curves were acquired using contact mode, and analyses of the obtained force–distance curves were performed with JPK DP software v.5 (JPK Instruments). In all cases, the indentation depth was ~ 0.5 µm, and the spring constant of each cantilever was determined before measurements using the thermal noise method in air (nominal value, 0.2 N/m). For inhibitor experiments, 5 µM (−)-blebbistatin (Merck Millipore), 30 nM taxol (Adipogen, San Diego, CA, USA) or an equal volume of DMSO was added to the medium 30 min (blebbistatin, DMSO) or 2 h (taxol) before AFM measurements began.

**3D calculation of spindle orientation**. Brain sections were co-immunostained with a goat (sc-7396, 1:1000, Santa Cruz Biotechnology) or rabbit (T5192, 1:500, Sigma-Aldrich) anti-γ-tubulin pAb, mouse anti-ZO1 mAb (33–9100, 1:200, Life Technologies) and chick (GFP-1020, 1:1000, Aves Labs) or rabbit (598, 1:1000, MBL) anti-GFP pAb, along with DAPI nuclear staining (Thermo Fisher Scientific). Confocal images were captured by using an FV1000 microscope (Olympus) with a ×40 or ×60 objective lens and 0.5-µm interval-Z scanning. Images of mitotic spindles in sections were 3D reconstructed by ImageJ software, and spindle orientation was measured by using the R package[39].

**RNA in situ hybridization**. Nonradioactive in situ hybridization of frozen sections of CD1 mouse brains, spinal cords, or ferret brains was performed using digoxigenin (DIG) (Roche, Basel, Switzerland)-labeled antisense RNA probes. The probe for *C230098O21Rik*, a transcript variant of *lzts1*, was located at 430–1541 of AK082735, and the probe for *Tnc* was located at 5593–6082 of ferret Tnc cDNA (transcript of TNC-201, ENSMPUT00000001705.1).

**Assessment of the siRNA efficiency**. To evaluate the silencing efficiency of siRNAs targeting *lzts1* (Supplementary Figure 2), we co-transfected pCAG::Lzts1 or pCAG:: siRNA-resistant-Lzts1 (0.5 µg per well) and pCAG::EGFP (0.1 µg per well) along with 0.5 µl of 20 µM siRNA (siRNA#1, #2 or control siRNA) into COS7 cells (from RIKEN BRC) growing in 24-well plates using 1.5 µl of Lipofectamine 2000 (Thermo Fisher Scientific). After 1 day, cells were collected, and SDS-PAGE and immunoblotting were performed with a rabbit anti-Lzts1 polyclonal antibody (HPA006294, 1:2000, Sigma-Aldrich) or rabbit anti-GFP polyclonal antibody (598, 1:2000, MBL) and HRP-conjugated anti-rabbit IgG (NA934, 1:2000, GE Healthcare, Little Chalfont, UK). Chemiluminescence (Western Lightning Plus-ECL, Perkin Elmer Inc., Waltham, USA) was detected using an ImageQuant LAS4010 instrument (GE Healthcare).

**Statistical analysis**. Differences were analyzed with the R package using Wilcoxon rank sum tests with two-tailed *p* values and Brunner–Munzel tests when equal variances were not assumed. Permuted Brunner–Munzel tests[65] were used for small sample sizes. Steel–Dwass tests were used for multiple comparisons of nonparametric data. Fisher's exact test was used in the analysis of contingency tables. No randomization was used, and no samples were excluded from the analysis. No statistical methods were used to predetermine the sample size owing to experimental limitations, but the sample sizes used here were similar to those described in related previous studies[24,50]. Blindings were performed for capture images and quantification of Sox2$^+$ GFP$^+$ cells (control and KD) in Fig. 9h–j and of Hes1$^+$GFP$^+$ cells in Fig. 10c, d. The sample size used in each analysis is shown in the legends.

**Reporting summary**. Further information on research design is available in the Nature Research Reporting Summary linked to this article.

## Code availabity

Spindle orientation was measured by the R package "utilFuncs.R" script[39] that has been provided as text format at https://doi.org/10.3389/fncel.2015.00033.

## Data availability

The source data underlying Fig. 2b, c, 3a–d, g, 4c, 5d, i–m, 6e, 9i, 10b, d and Supplementary Figs. 4–6, 10, and 11 are provided as a Source Data file. Other data sets generated during the current study are available from the corresponding authors on reasonable request.

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

## Acknowledgements

We thank N. Noguchi, M. Masaoka, T. Suetsugu, K. Murase, T. Ikawa, and K. Ohta for providing excellent assistance; D. Konno, M. Furuse, Y. Izumi, and T. Otani for participating in valuable discussions; T. Fujimori and S. Tsukita for the gift of the R26-ZO1-EGFP mouse; T. Shimogori for the gift of marmoset brains; and I. Fujita for the gift of the pPB-CAG-EGFP-ZO1 plasmid. This study was supported by JSPS KAKENHI grant numbers JP16K06990 and JP17H05765, a Grant-in-Aid for Scientific Research on Innovative Areas 'Interplay of developmental clock and extracellular environment in brain formation', and THE HORI SCIENCES AND ARTS FOUNDATION (A.K.).

## Author contributions

A.K. designed and performed most experiments and analyses. T.K., T.S., and K.S. performed the time-lapse imaging, and T.K. performed immuno-EM and en face imaging of IHC. A.S., and A.K. performed 3D estimations of the spindle orientation, and A.S. performed the time-lapse analysis of apical junction inheritance. R.T. performed time-lapse studies with blebbistatin. M.B. performed in situ hybridization of the ferret brain. A.N. performed AFM measurements. F.M., Y.T., and T.M. provided technical and intellectual support. A.K. wrote the manuscript.

## Additional information

**Competing interests:** The authors declare no competing interests.

