## [Peer Review File · Nature Communications]

Reviewers' comments:

Reviewer #2 (Remarks to the Author):

In this manuscript, the authors propose that *Lzts1* is an important regulator of delamination of neural progenitors and the generation of oRGs through a dose-dependent regulation of actomyosin behavior in apical endfeet in the developing cortex. The use of live imaging to monitor cellular behaviors *ex vivo* reveals that *Lzts1* can induce a mitotic behavior in mice previously only seen in gyrencephalic species. Also of interest is the finding that *Lzts1* expression may be oscillating out of phase with *Hes1*, and that the levels of *Lzts1* may be important to determine if a delaminating progenitor becomes an oRG.

Overall the findings are novel and hint at an interesting mechanism by which *Lzts1* regulates these processes. However, there are a few issues with the manuscript:

Fig. 1E-Indicating overlap of *Lzts1* and *Tbr2*EGFP in apical processes connected to nuclei that do not seem to be *Tbr2*⁺, possible bleed through of fluorophores? A composite image pointing to cells with nuclear *Tbr2*/cytoplasmic *Lzts1* would be more powerful.

Figure 1g'-How are the authors certain that the *Lzts1* punctae are the apical endfeet of departing cells?

Figure 2 legend is not clear, 2b should be a one-way ANOVA since they are making multiple comparisons.

Figure 3b and supplemental figure 3 measure distance of cells from VZ, but it might be more appropriate to equally bin the areas being measured to control for differences in cortical width.

Fig 4l is mentioned before fig 4h-k, and the image in fig 4l doesn't show the VZ for either condition, it would aid the story to show a lack of GFP+PH3⁺ cells in the VZ rather than cropping out that part of the image.

Fig 5- it would aid the manuscript to give the frequency the authors observed the MST/apical process behaviors at the different concentrations to make a more convincing argument that the change in plasmid concentration affects the final concentration in the electroporated cells and that this has a consistent effect on behavior.

Figure 7c-Font/style of this panel is different from others

Supplementary Figure 4-uses permuted Brunner-Munzel test which is for comparison of two samples (similar to t-test), need to do a one-way ANOVA.

Line 273-*Lzts1* is typed as "*Lzt1*"

Lines 287 and 290- The authors say the anchoring of chromosomes but I think they mean the anchoring of centrosomes.

General comments:

-How can the authors be sure that plasmid concentration correlates to the amount of protein expressed in a cell? They need to show that there is an increase in plasmid delivery/protein amount upon higher plasmid concentration otherwise it might just be that there are more cells expressing the construct resulting in a potential cell non-autonomous effect. I'm not sure if this technique is sensitive enough to definitively say that this is a dose-dependent effect.

-The figures are crowded and at times difficult to follow.

-I'm unsure how the authors obtained the units $\text{area}^{(1/2)}$. Shouldn't this be the square root of the area? Why wouldn't they use $(\text{units})^2$?

-The authors routinely (figs 2b, 3e, 3f, 4b, 6b, s4) use statistical analyses that are for comparison of two samples on multiple samples. The appropriate tests for nonparametric multiple comparisons should be used.

-The dot plots in figs 3f and 6g are very crowded, particularly 6g, it is hard to differentiate the columns from each other.

Reviewer #3 (Remarks to the Author):

Lzts1 controls both neuronal delamination and oRG generation during mammalian development

In this study by Kawaguchi's group, the authors investigate mechanisms that regulate delamination of newborn neural stem cell progeny and generation of newborn outer radial glia (oRG) cells during embryonic mammalian brain development. Delamination is currently associated with abolition of gap junctions, whereby daughter cells retract their apical process and migrate away from the ventricular border, as well as generation of cells using oblique cleavage plane orientation at mitosis. These processes are further implicated in differentiation and in the generation of outer radial glial cells (oRGs, which are predominant in upper species). But the mechanisms remain poorly understood. The authors examine the molecule Lzts1 for roles in these processes. They first show that Lzts1 is expressed in the mouse, mainly in new neurons and intermediate progenitors (and in primates is expressed in oRGs). Using knockdown in mice they show that Lzts1 controls delamination and RG cleavage plane orientation but not generation of intermediate progenitors or neurons. Overexpression has an opposing phenotype, promoting more basal progeny and increased oblique divisions. They also use overexpression in ferrets to support the conclusion that Lzts1 controls production of oRGs. Study of overexpression in the presence of acto-myosin inhibitors in mice, leads them to conclude that Lzts1 promotes acto-myosin to induce delamination.

This is an extensive study that investigates a poorly understood mechanism that could have been coopted during evolution to generate oRGs, which are key progenitors responsible for the generation of a complex cerebral cortex. This study provides novel and significant insights regarding this delamination process which will be of interest to the cortical development field. However, I found some of the main conclusions insufficiently supported, in part due to the overreliance on over-expression. Moreover, the authors need to further clarify mechanisms of action, which likely requires careful editing and reworking of the paper as well as additional experiments.

Major concerns:

1. A major conclusion of the paper is that Lzts1 promotes oRG generation. However this is not

sufficiently supported. A) In Figure 8 they show that *Lzts1* is expressed in mouse RGCs but only transiently at mid-stages. Mice only make small numbers of oRGs and mainly at later stages of development. If *Lzts1* is involved in the generation of oRGs, one would expect it is higher at later timepoints in aRGs (oRG generating cells). They do use mice to show that *Lzts1* impacts spindle orientation (in Figure 7 (LOF) and 8 (OE)) which is consistent with more basal progenitor generation, but only correlative. B) The use of ferrets is more valuable than mice for study of oRG generation. They include one experiment to overexpress *Lzts1* in ferrets, but this is complicated as it is not clear how physiologically relevant this is. Beyond looking at Sox2+ cells in ferret overexpression they should assess canonical oRG markers (as recently defined by Kriegstein groups and others). Use of ferrets would ideally be better exploited to include knockdown studies. If *Lzts1* is required for oRG generation, as they conclude, fewer oRGs should be made in this knockdown. In the absence of such data I think this conclusion is not well substantiated.

2. Generally speaking, the authors invoke many mechanisms to explain the role of *Lzts1* in lamination. Further, the paper is a bit disorganized-as it jumps between knockdown and overexpression making the central conclusions difficult to interpret. As a result the paper suffers somewhat from "data dump" without a clear idea of what is most relevant, making it overly speculative. Figure 9 attempts to bring all of this together but is itself extremely confusing, again perhaps due to inclusion of multiple datasets. Some of this may be related to the over-reliance on GOF studies, which as I suggest in point 1 may not be physiologically relevant. The manuscript would be more impactful and stronger if the authors carefully edit experiments (perhaps a sole focus on lamination), and consider reorganizing the paper to focus on knockdown and overexpression separately, with further focus on a central mechanism.

3. Figures 2, 3, and 8 include loss of function data which show that *Lzts1* controls: more basal location of newborn cells (figs 2 and 3) and spindle orientation (fig 8). Generally these data support their conclusion that *Lzts1* controls delamination. However they need to be better fleshed out. A) Figure 2: Data mentioned in lines 155-156: the authors need to show the data that EGFP+ cells in the VZ after overexpression of *Neurog1/2* together with *Lzts1* shRNA-knockdown are *Neurog2+*, *Sox2-* and *Pax6-*. They should also use a different marker (such as *NeuroD2* or *Tbr1*) to assess differentiated fates (as *Neurog2* was actually overexpressed in that experiment). They also need to show that the total number of GFP+ cells are not affected in these experiments and given that *Lzts1* is expressed in neurons, that there is no change in neuron generation in *Lzts1* loss of function - this is only shown for overexpression currently. B) The graph of GFP distribution is difficult to interpret relative to the figure. Why not represent in bins as in Figure 3B? C) For Figure 3B, the result showing more apically positioned cells in the *Lzts1*-KO is not convincing and lacks error bars and statistical tests. D) The authors need to include siRNA rescue in the absence of *Ngn2* (currently only shown in presence of *Ngn2* overexpression).

4. The authors state that *Lzts1* functions to "enhance actomyosin system" but it is not clear what they mean by this, as it relies on experiments with blebbistatin (showing incomplete rescue). Figure 6c uses an atomic force microscope to measure stiffness in fibroblasts, but I am not sure how much this adds- is this outcome really relevant in a tissue context where there are vastly different pulling forces on the cell? There is also heavy speculation as to how *Lzts1* intersects with microtubules but this cytoskeletal component is never evaluated formally. If acto-myosin activation is important, additional experiments could be included to assess activation of myosin light chain kinase and other differences in contraction via phosphorylation events and perhaps fret? Can they include some experiments to support the interesting idea that *Lzts1* mediates these activities are via interface with microtubules?

5. Figure 5 includes live imaging which correlates levels of *Lzts1* with different behaviors of RGS- but this is all qualitative. A more comprehensive quantitative analysis of the different behaviors is needed to more clearly assess the effect of *Lzts1* expression. Of note there is no wild-type control included here making it difficult to interpret. Again these experiments rely upon overexpression in

apical RGs, making the physiological relevance questionable (given low levels of Lzts1 expression in these cells).

Minor comments:

1. In Figure 1, do the authors observe any co-localization of Lzts1 with microtubules by EM?
2. For Figure 3A, please show a high mag of the Lzts1 expression in GFP knockout cells to convince the reader that the antibody is specific.
3. In Figure 4a the authors need to clarify why they use Lyn-EGFP and what 1.0 and 0.2 are in the figure.
4. Lines 191-194: It is not clear why the authors mention Scrt1 and what the experiments related to this gene bring to the study. Consider removing this result or be more specific on the rationale for this piece of data.
5. Figure 4e-g: It seems that a lot of the ectopic progenitor cells are not EGFP+, suggesting a non-cell autonomous role for Lzts1. The authors should comment on this result.
6. Figure 4E would benefit from graph of bins to support the conclusion that there are more cells in the SVZ/IZ following overexpression.
7. Figures 6C,G: instead of comparing electroporated cells with the surrounding non electroporated cells, it would be better to use electroporated cells without any overexpression plasmid as a control.
8. Line 287: The authors should clarify what they mean with "anchoring of chromosomes to the apical portion of the process forming the AJ"
9. Figure 7e: How many brains were analyzed in this experiment?
10. The statement that Ngn2 upregulates Ltz is not supported by data in figure 2, which only support a genetic relationship between the two.
11. In figure 8K they should quantify spindle orientation as in Figure 7E.
12. General comment: For many results, the authors did not mention the number of samples/animals used for the statistical comparison. Please add those.

6th Dec 2018

Dear Reviewers,

We wish to express our appreciation to the Reviewers for their instructive and insightful comments, which have helped us to greatly improve the manuscript. In accordance with the reviewers' suggestions, we have performed the loss-of-function studies in the ferret brains and extensively edited our manuscript. Our point-by-point responses are provided below.

Sincerely yours,

Ayano Kawaguchi

Department of Anatomy and Cell Biology,
Nagoya University Graduate School of Medicine,
65 Tsurumai, Showa-ku, Nagoya, Aichi 466-8550, Japan

Responses to the Reviewers' Comments

Reviewer #2 (Remarks to the Author):

In this manuscript, the authors propose that Lzts1 is an important regulator of delamination of neural progenitors and the generation of oRGs through a dose-dependent regulation of actomyosin behavior in apical endfeet in the developing cortex. The use of live imaging to monitor cellular behaviors *ex vivo* reveals that Lzts1 can induce a mitotic behavior in mice previously only seen in gyrencephalic species. Also of interest is the finding that Lzts1 expression may be oscillating out of phase with Hes1, and that the levels of Lzts1 may be important to determine if a delaminating progenitor becomes an oRG.

Overall the findings are novel and hint at an interesting mechanism by which Lzts1 regulates these processes. However, there are a few issues with the manuscript:

Minor comments:

(1) Fig. 1E-Indicating overlap of Lzts1 and Tbr2EGFP in apical processes connected to nuclei that do not seem to be Tbr2+, possible bleed through of fluorophores? A composite image pointing to cells with nuclear Tbr2/cytoplasmic Lzts1 would be more powerful.

We appreciate the reviewer's comment on this point. We revised the figures showing the IHC of Tbr2-promoter::EGFP Tg mouse brains. We performed triple staining with anti-Lzts1, anti-GFP, and anti-Tbr2 antibodies, and added a high magnification image to show the Tbr2+Lzts1+ cells in the VZ (Fig. 1e).

We revised the Fig. 1 legend as follows:

e and f) Lzts1 IHC of the E14 dorsal forebrain of Tbr2::EGFP Tg mice (e) and Gadd45g::d4Venus Tg mice (f) showing that GFP⁺, Tbr2⁺ or Venus⁺ differentiating cells are Lzts1⁺. e') Magnified view of Tbr2⁺Lzts1⁺ cells in the VZ.

(2) Figure 1g'-How are the authors certain that the Lzts1 punctate are the apical endfeet of departing cells?

This figure shows an en face apical view of the fixed tissue, not the live imaging of the departing cells. Thus, as the reviewer noted, we did not confirm that the punctate Lzts1 signals were actually the apical endfeet of the departing cells. The identification of the departing cells observed with live imaging using IHC with an anti-Lzts1 antibody was technically difficult; therefore, we revised the text as follows:

Results, page 5

...The values for the Lzts1 immunofluorescence signal at the AJs were negatively correlated with the values ($\text{apex area}^{[1/2]} \mu\text{m}$) that are proportional to the circumferential length of the ZO1⁺ AJ ring (Fig. 1i), raising the possibility that Lzts1 plays a role in delamination based on its localization to the AJ. If this hypothesis is true, the dot-like Lzts1 immunoreactivity observed among the ZO1⁺ meshwork (Fig. 1g') represents the apical endfeet of departing cells, because separate live images of the neuronally differentiating cells revealed that their apices become progressively smaller to form a 'vertex' through delamination ³².

(3) Figure 2 legend is not clear, 2b should be a one-way ANOVA since they are making multiple comparisons.

We apologize that our previous description was not clear. We agree with the Reviewer's point that we should perform the multiple comparisons test. In this case, each sample size was relatively small, and we were not able assume a normal distribution of the data. Therefore, we applied the Steel-Dwass test as the non-parametric multiple comparisons test in Fig. 2c.

Furthermore, as shown in our response to major point (3-B) from Reviewer 3, we revised Fig. 2b and 2c with an orange-colored mark to clearly show that we compare the percentage of cells within 0-60 μm distance from the apical surface, and revised the legend of Fig. 2 as follows:

Figure 2:

b and c) Neurog1/2-induced migration from the VZ is partially inhibited by Lzts1 KD induced by siRNA#1 (and more moderately by siRNA#2). This phenotype was rescued by the expression of siRNA-resistant mutLzts1 but not wild-type (WT) Lzts1. *In vivo* electroporation of *neurog1/2* and *egfp* was performed at E13, and brains were examined after 24 h (see also Supplementary Fig. 2).

b) Distributions of EGFP⁺ cells. Means \pm s.d.

c) Percentage of the EGFP⁺ electroporated cells within the area of 0–60 μm from the apical surface (orange colored region in **b**) among total EGFP⁺ cells. Means \pm s.d. (Steel-Dwass test, N=8, 8, 7, 8, 11, 13, and 8 sections from 4, 4, 4, 4, 7, and 4 embryos, respectively).

(4) Figure 3b and supplemental figure 3 measure distance of cells from VZ, but it might be more appropriate to equally bin the areas being measured to control for differences in cortical width.

We appreciate the reviewer's suggestion. We have generated histograms with 10 bins, which range from the apical to the basal side of the cerebral walls, to show the distribution of the GFP⁺ cells. We showed the data with error bars (s.d.), and the differences were analyzed by using the non-parametric test (Brunner-Munzel test) (new figure Fig. 3f and Supplementary Fig. 3).

Figure 3

f) Lzts1 KO retards the overall radial migration of cells from the apical surface. *In vivo* electroporation was performed at E13, and the distribution of EGFP⁺ cells in the cerebral wall was examined at E15 using 10 bins. Means \pm s.d., Brunner-Munzel test, * p <0.05, ** p <0.01, N=8 (control) and 9 (KO) sections from 3 embryos per experiment (see also Supplementary Fig. 3).

Supplementary Figure 3 (related to Figure 3)

Lzts1 KD retards the radial migration of cells from the apical surface.

In vivo electroporation of the EGFP expression vector with Lzts1-siRNA#1 or siRNA-NC (negative control siRNA) was performed at E13, and the distribution of EGFP⁺ cells in 10

bins of the cerebral wall was examined at E15 (N=8 sections from 4 embryos were analyzed per experiment, ** $p < 0.01$ and * $p < 0.05$, Brunner-Munzel test).

(5) Fig 4l is mentioned before fig 4h-k, and the image in fig 4l doesn't show the VZ for either condition, it would aid the story to show a lack of GFP+PH3+ cells in the VZ rather than cropping out that part of the image.

We appreciate the comment on this point.

We have revised the order in which the images are presented in the figures (new Fig. 5). We also added a figure showing the position of GFP+PH3+ cells (new Fig. 5h), in which we marked the GFP+PH3+ cells with an asterisk to show the lack of GFP+PH3+ cells at the apical surface of the Lzts1 OE brains.

Fig. 5 legend

h and i) Lzts1 increases the percentage of non-apical PH3+ mitotic cells in a dose-dependent manner. The asterisk in (h) indicates a PH3+ GFP+ cell.

(6) Fig 5- it would aid the manuscript to give the frequency the authors observed the MST/apical process behaviors at the different concentrations to make a more convincing argument that the change in plasmid concentration affects the final concentration in the electroporated cells and that this has a consistent effect on behavior.

We sincerely appreciate the Reviewer's suggestion. We have added a figure (Fig 6e) to show the percentage of cells exhibiting each departure pattern following Lzts1 overexpression induced by the electroporation of different plasmid concentrations. We believe that this panel clearly shows the predominant Type A pattern observed in the high-level ($2.0 \mu\text{g } \mu\text{l}^{-1}$) Lzts1-overexpressing cells. We also added Fig 6d, which shows the normal cell departure pattern (normal INM) observed in the control experiment. Then, we revised the text and figure legends as follows:

Results, page 9

Lzts1 overexpression induces apical process retraction and MST

To further understand the abnormal cellular positioning after Lzts1 overexpression, we performed live imaging to observe the behaviors of aRG-like cells expressing exogenous Lzts1 (at the concentration $1.0\text{--}2.0 \mu\text{g } \mu\text{l}^{-1}$) (Fig. 6a–c). Since we very sporadically visualized aRGs to obtain clear images, only a small fraction of Lzts1-overexpressing VZ cells was monitored. In the control experiment, the nuclei of all EGFP-labeled aRG-like cells (63 cells that we observed) moved apically and soon divided at the apical surface (designated as

'normal') (Fig. 6d). In contrast, the aRG-like cells with forced *Lzts1* expression behaved differently, showing the following three patterns: retraction of the apical process (Type A), retraction of the apical process followed by MST (Type B), and MST from the apical surface after apical INM (Type C) (Fig. 6a–c and Supplementary Movies 1–4).

Page 10

Notably, *Lzts1*-expressing aRG-like cells showed these various behaviors according to the concentration of the *lzts1* expression vectors introduced during electroporation (Fig. 6e) ($p = 8.6 \times 10^{-7}$, Fisher's exact test, two-sided). Type A behaviors were more frequently observed in the experimental condition in which $2.0 \mu\text{g } \mu\text{l}^{-1}$ ($34/75 = 45.3\%$) was electroporated than in cells electroporated with $1.0 \mu\text{g } \mu\text{l}^{-1}$ *lzts1* expression vector ($10/94 = 10.6\%$). In cells electroporated with $1.0 \mu\text{g } \mu\text{l}^{-1}$ vector, many aRG-like cells exhibited the normal INM, while 24.5% ($23/94$) showed Type B behaviors and 13.8% ($13/94$) showed Type C movement. Since the concentration of the expression vectors introduced during electroporation was positively correlated with the *Lzts1* expression levels in the electroporated cells (Supplementary Fig. 6), these diverse cellular behaviors are postulated to correspond to the *Lzts1* expression levels in the cells, although it might also include a non-cell autonomous effect, as observed for the positions of the *Lzts1*-overexpressing progenitors in the SVZ (Fig. 5e–g).

Figure 6

***Lzts1* overexpression induces delamination and MST.**

Time-lapse images of *Lzts1*-overexpressing aRG-like cells (pCAG::*Lzts1*: $2.0 \mu\text{g } \mu\text{l}^{-1}$ in **a**, $1.0 \mu\text{g } \mu\text{l}^{-1}$ in **b** and **c**) or control EGFP-expressing cells (**d**). The endfeet of the apical processes are indicated by arrows. Electroporation was performed at E12, and 8–24 hrs later, brain slices were cultured and images of aRG-like cells were captured for 20 hrs. See also Supplementary Movies 1–4.

a) Image of an *lzts1*-overexpressing cell showing the retraction of its apical process without mitosis (Type A).

b) Image of an *lzts1*-overexpressing cell that retracted its apical process from the apical surface, presumably during G2 phase, and underwent MST after retraction (Type B).

c) Time-lapse images of an *lzts1*-overexpressing cell showing the 'MST from the apical surface' behavior after apical INM during G2 phase (Type C). Note that the apical process was maintained during mitosis (arrowheads).

d) Time-lapse images of the aRG-like cell in the control experiment showing normal INM and apical division. In control experiments, all aRG-like cells that we observed (63 cells from 5 embryos) exhibited this pattern.

e) The Lzts1 plasmid concentration in the electroporation was related to the type of behaviors exhibited by aRG-like cells ($p=8.6 \times 10^{-7}$, Fisher's exact test, two-sided, N=75 and 94 cells from 6 and 4 embryos). Type A was predominantly observed in cells electroporated with $2.0 \mu\text{g } \mu\text{l}^{-1}$ pCAG::Lzts1 (see also Supplementary Fig. 6).

Bars, 50 μm in (a–d).

(7) Figure 7c-Font/style of this panel is different from others

We corrected the discrepancy.

(8) Supplementary Figure 4-uses permuted Brunner-Munzel test which is for comparison of two samples (similar to t-test), need to do a one-way ANOVA.

We agree with the point. Since a small sample size was available for this condition and we were not able to assume a normal distribution of the data, we applied the Steel-Dwass test as the non-parametric multiple comparisons test in new Supplementary Fig. 4.

(9) Line 273-Lzts1 is typed as "Lzt1"

We corrected the error.

(10) Lines 287 and 290- The authors say the anchoring of chromosomes but I think they mean the anchoring of centrosomes.

We agree with this point and corrected it as follows:

Result, page 12

The Type C departure, i.e., 'MST from the apical surface' induced by Lzts1 overexpression (Fig. 6c), indicates that Lzts1 inhibits the anchoring of centrosomes to the apical portion of the process forming the AJs during M phase. Thus, we speculated that weak expression of Lzts1 in aRGs might weaken the anchoring machinery of centrosomes and give rise to oblique aRG division, which might mimic the physiological condition of oRG production.

(Major) General comments:

(1) -How can the authors be sure that plasmid concentration correlates to the amount of protein expressed in a cell? They need to show that there is an increase in plasmid delivery/protein amount upon higher plasmid concentration otherwise it might just be that there are more cells expressing the construct resulting in a potential cell non-autonomous effect. I'm not sure if this technique is sensitive enough to definitively say that this is a dose-dependent effect.

We appreciate the Reviewer's comment. In response, we have examined the expression levels of Lzts1-Flag in the electroporated cells, which were electroporated with different concentrations of the expression vector *in vivo*. We added the result to the new Supplementary Fig. 6. As shown in this figure, the overall Flag levels were positively correlated with the concentration of the plasmid introduced during electroporation. Nonetheless, since we did not directly examine the levels of Lzts1 in the cells using live imaging, we were not able to exclude the possibility that the observed phenotypes included a non-cell autonomous effect of the Lzts1 overexpression. (We also discussed the non-cell autonomous effect in our response to minor point 5 from Reviewer #3).

Therefore, we have revised the text and figure legends as follows:

Results, Page 10

...Since the concentration of the expression vectors introduced during electroporation was positively correlated with the Lzts1 expression levels in the electroporated cells (Supplementary Fig. 6), these diverse cellular behaviors are postulated to correspond to the Lzts1 expression levels in the cells, although it might also include a non-cell autonomous effect, as observed for the positions of the Lzts1-overexpressing progenitors in the SVZ (Fig. 5e-g).

Supplementary Figure 6 (related to Figure 6)

Lzts1 expression levels in the cells were positively correlated with the plasmid vector concentration introduced during the *in vivo* electroporation.

Different concentrations of the Flag-tagged Lzts1-expression vector were co-electroporated with EGFP expression vectors. One day after electroporation, the cerebral walls were dissociated, cultured for 1 hr and stained with anti-Flag and anti-GFP antibodies after fixation. The Flag/GFP ratio was calculated in each GFP⁺ cell. * p<0.05 and ** p<0.01 (Steel-Dwass test, N=38, 98, 88, 92 cells from two embryos and 4 wells per condition; medians with Q1 and Q3 values are shown).

(2) The figures are crowded and at times difficult to follow.

As described in our response to major point 2 from Reviewer #3, we extensively edited our text and figures to present our results fluently.

In response, we revised the positions of the panels in Fig 1. Furthermore, we moved several panels (Figs. 4b, 4d, 4n, 4o, 8d, 8f, 8g, and 9a) to the Supplementary information, Fig. 4m to new Fig. 10, and Fig. 6g to new Fig. 4 to reduce the crowding of the figures.

(3) -I'm unsure how the authors obtained the units $\text{area}^{1/2}$. Shouldn't this be the square root of the area? Why wouldn't they use $(\text{units})^2$?

We agree that our previous description was not sufficient to clarify this point. We obtained the area (μm^2) of the apical endfeet by ImageJ software and calculated the square root of the area as $\text{area}^{1/2}$ (μm). The obtained square root value was assumed to be proportional to the circumferential length of each AJ ring. We considered that the scattering drawing of 'area value' was inappropriate to present the data because the plotted points corresponding to the high values (including outlier) affected the apparent distribution pattern of the plotted points on the scatter plot, and thus we were unable to clearly show the apical contraction phenomena (shown by the plot points corresponding to the low values) that was the focus of our investigation.

We want to emphasize that the statistical test returned the exact same p-values regardless of whether we analyzed the 'area value' or the 'square root of the area values' because we applied the tests to the ranked data (Wilcoxon exact test, Brunner-Munzel test, or Steel-Dwass test).

Therefore, we revised the text as follows:

Method, page 25

...We obtained the area (μm^2) of the apical endfeet by ImageJ software, and calculated the square root of the area value as the apex $\text{area}^{1/2}$ (μm), which was assumed to be proportional to the circumference of each AJ ring.

(4)-The authors routinely (figs 2b, 3e, 3f, 4b, 6b, s4) use statistical analyses that are for comparison of two samples on multiple samples. The appropriate tests for nonparametric multiple comparisons should be used.

We agree with the reviewer's point, and we have revised our statistical tests in Fig. 2b (new figure Fig. 2c), Fig. 3e-f (new Fig. 3i), 4b (new Supplementary Fig. 5) and S4 to the nonparametric multiple comparisons Steel-Dwass test. In new Fig. 3i, we added the results of the rescue experiment for the KD cases. Thus, we performed Steel-Dwass test among these four experimental conditions. However, the KO experiment was distinct from the KD experiment, as it used a different approach and time point, and we performed the Brunner-Munzel test to compare KO and KO-control. We also revised Fig 3i to show the results from KD and KO experiments separately.

As for Fig. 6b, statistical data were not presented; therefore, we wondered if we might be mistaken in pointing to Fig 6c, d, and g. These figures correspond to the new Fig. 7d, h, and

4b, respectively, and we have also revised our tests to the Steel-Dwass test.

In summary, we performed Steel-Dwass multiple comparison test in Fig. 2c (among 7 conditions), Fig. 3i (4 conditions), Fig. 4b (4 conditions), Fig. 7d (4 conditions), Fig. 7h (8 conditions), Fig. 9i (4 conditions), Fig. 10b (3 conditions), Fig. 10d (4 conditions), Supplementary Fig. 4d (4 conditions), Supplementary Fig. 5a (3 conditions), Supplementary Fig. 6 (4 conditions) and Supplementary Fig. 11b (3 conditions).

(5) The dot plots in figs 3f and 6g are very crowded, particularly 6g, it is hard to differentiate the columns from each other.

We have revised Figs 3f and 6g to use the small plotted points (new figure 3i and 4b).

Reviewer #3 (Remarks to the Author):

In this study by Kawaguchi's group, the authors investigate mechanisms that regulate delamination of newborn neural stem cell progeny and generation of newborn outer radial glia (oRG) cells during embryonic mammalian brain development. Delamination is currently associated with abolition of gap junctions, whereby daughter cells retract their apical process and migrate away from the ventricular border, as well as generation of cells using oblique cleavage plane orientation at mitosis. These processes are further implicated in differentiation and in the generation of outer radial glial cells (oRGs, which are predominant in upper species). But the mechanisms remain poorly understood. The authors examine the molecule *Lzts1* for roles in these processes. They first show that *Lzts1* is expressed in the mouse, mainly in new neurons and intermediate progenitors (and in primates is expressed in oRGs). Using knockdown in mice they show that *Lzts1* controls delamination and RG cleavage plane orientation but not generation of intermediate progenitors or neurons. Overexpression has a opposing phenotype, promoting more basal progeny and increased oblique divisions. They also use overexpression in ferrets to support the conclusion that *Lzts1* controls production of oRGs. Study of overexpression in the presence of acto-myosin inhibitors in mice, leads them to conclude that *Lzts1* promotes acto-myosin to induce delamination.

This is an extensive study that investigates a poorly understood mechanism that could have been coopted during evolution to generate oRGs, which are key progenitors responsible for the generation of a complex cerebral cortex. This study provides novel and significant insights regarding this delamination process which will be of interest to the cortical development field. However, I found some of the main conclusions insufficiently supported,

in part due to the overreliance on over-expression. Moreover, the authors need to further clarify mechanisms of action, which likely requires careful editing and reworking of the paper as well as additional experiments.

Major concerns:

1. A major conclusion of the paper is that *Lzts1* promotes oRG generation. However this is not sufficiently supported.

A) In Figure 8 they show that *Lzts1* is expressed in mouse RGCs but only transiently at mid-stages. Mice only make small numbers of oRGs and mainly at later stages of development. If *Lzts1* is involved in the generation of oRGs, one would expect it is higher at later timepoints in aRGs (oRG generating cells). They do use mice to show that *Lzts1* impacts spindle orientation (in Figure 7 (LOF) and 8 (OE)) which is consistent with more basal progenitor generation, but only correlative.

We appreciate the reviewer's comment. First, we agree with the point that the mouse is not the ideal animal model to investigate oRG generation because oRGs are a minor population in mice and have a more limited proliferative potential than in the brains of gyrencephalic species. Therefore, as we wrote in our response to major point 1(B), we performed additional LOF studies in the ferret brain.

Regarding the effect *Lzts1* on the later stages of mouse development, we performed an additional LOF study. We expressed *Lzts1*-siRNA with an EGFP plasmid by performing *in vivo* electroporation at E16, and after 2 days, we examined the emergence of Sox2+GFP+ cells in the SVZ. The results are shown in new Supplementary Fig. 9, suggesting that *Lzts1*-KD does not perturb the generation of Sox2+ SVZ cells from the late progenitor cells. Thus, we postulate that *Lzts1* controls oRG generation only during the neurogenic stage, and other mechanisms likely control the later stage.

Interestingly, a recent paper (Zweifel et al., 2018) has observed high HOPX expression from the late embryonic stage in mice (~E16), and the highest expression was observed in subsets of NSCs biased to acquire an astroglial fate. Since HOPX expression is known to induce SVZ progenitor cells, these results suggest that different molecular mechanisms control the generation of the SVZ progenitors at the late (gliogenic) stage from the mid (neurogenic) stage, at least in mice. In contrast, as the proneural genes induce *Lzts1* expression, *Lzts1* expression would ensure the generation of the SVZ progenitors with neurogenic potential during the mid-embryonic stage.

Therefore, we have revised our manuscript as follows:

Results, page 15

On the other hand, we also found that *Lzts1* KD at a later stage (E16 → E18 experiment) did not perturb the generation of Sox2⁺GFP⁺ cells in the SVZ (Supplementary Fig. 9). Thus, the generation of SVZ progenitor cells at the late, gliogenic stage might be controlled by a different molecular mechanism from the mid, neurogenic stage, at least in mice.

B) The use of ferrets is more valuable than mice for study of oRG generation. They include one experiment to overexpress *Lzts1* in ferrets, but this is complicated as it is not clear how physiologically relevant this is. Beyond looking at Sox2⁺ cells in ferret overexpression they should assess canonical oRG markers (as recently defined by Kriegstein groups and others). Use of ferrets would ideally be better exploited to include knockdown studies. If *Lzts1* is required for oRG generation, as they conclude, fewer oRGs should be made in this knockdown. In the absence of such data I think this conclusion is not well substantiated.

We appreciate the Reviewer's comment on this important point.

In the ferret experiments shown in Fig. 4, we overexpressed ferret *Lzts1* in the E28 brains using *ex vivo* electroporation. This developmental stage was earlier than the timing of physiological oRG generation and too early to express HOPX, a marker of oRGs used in humans (Pollen et al., 2015; Nowakowski et al., 2016). In these panels, we did not argue that *Lzts1*-OE increased the oRGs but simply showed that it induces the ectopic formation of progenitor cells in the SVZ. Therefore, we agree with the reviewer's suggestion that we should perform a LOF study in the ferret embryonic brains to examine whether *Lzts1* is involved in oRG generation. In response, we have moved these results from the GOF studies to the Supplementary information (new Supplementary Fig. 10) and performed the additional LOF experiments (new Figure 10 and Supplementary Fig. 11).

Practically, at the developmental stage when oRG production from aRGs frequently occurs in the ferret brain (Martínez-Martínez et al., 2016), daughter cells require approximately 4 days to migrate into the OSVZ after electroporation, and thus KD is not appropriate/ideal because of this relatively long lag time, which may allow several rounds of divisions and inevitably dilute KD vectors or siRNAs. Thus, we performed the KO studies using the CRISPR/Cas9 system with *in vivo* electroporation. Since oRG generation most frequently occurs at approximately E34 in the ferret, we performed the electroporation at E32, and fixed and analyzed the brains after 6 days (E38). We took advantage of the transposon system (piggyBac system) to sustain the labeling of the electroporated cells and their progenies for 6 days. In this system, the CAG promoter-EGFP-3NLS sequence is integrated into the genome of the electroporated cells by transposase, thereby enabling the sustained

labeling by EGFP-3NLS (Yusa et al., 2011).

In the ferret KO experiments, we prepared two types of negative control samples, i.e., 'Cas9' and 'Control.' 'Cas9' used no gRNA. In the 'Control' samples, the gRNA for a genomic sequence unrelated to *Lzts1* (ferret intron of *GDF5*) was used as the negative control gRNA, in which we examined the effects of genome editing itself on the cell behaviors that we analyzed. To knock out ferret *Lzts1*, we also prepared two types of samples, KO#1 and KO#2, each of them was the mixture of three different gRNAs for ferret *Lzts1* (Supplementary Table 1). By examining the anti-*Lzts1* immunoreactivity of the EGFP+ cells in the ISVZ, we confirmed our gRNAs of KO#1 more strongly suppressed *Lzts1* expression than those of KO#2 (Supplementary Fig. 11).

Then, we used *Hes1* as the marker to identify oRG-like cells in OSVZ in these LOF studies. Since the anti-*Hes1* antibody stains the nucleus of the undifferentiated neural progenitors, we used the EGFP with nuclear localization signals (EGFP-3NLS) to label the cells, accurately determine the level of *Hes1* immunoreactivity and ensure the reliability of the quantification of cell numbers. Unfortunately, the morphology of the labeled cells was difficult to assess using this method. However, based on the results from separate experiments, strong *Hes1* immunoreactivity was exclusively observed in the undifferentiated neural progenitor cells at E38 (rather than *Sox2* and *HOPX* at this developmental stage), and thus *Hes1*+ cells in OSVZ were assumed to be the undifferentiated progenitors that represent oRGs.

The results of the KO experiments are shown in new Fig. 10, which overall show similar phenotypes to the LOF mouse embryos. We believe that these results further strengthen our conclusion that *Lzts1* positively controls oRG generation, and again we sincerely appreciate the Reviewer's insightful suggestions to improve our manuscript.

Therefore, we have revised our manuscript as follows:

Results, page 15

To investigate whether *Lzts1* controls oRG generation in the ferret brain, we performed CRISPR/Cas9-induced KO studies. Since oRG production from aRGs most frequently occurs at approximately E34¹², we expressed gRNAs for ferret *lzts1*, hCas9, and EGFP-3NLS in the ferret cerebral wall at E32 through *in vivo* electroporation and examined the tissue at E38 (Fig. 10 and Supplementary Fig. 11). To sustain EGFP-3NLS expression in the electroporated cells and their progenies, we took advantage of the transposon system (piggyBac)⁴⁴, in which CAG promoter-EGFP-3NLS sequence was integrated into the

genome by a transposase. We generated four types of samples, i.e., 'Cas9' (negative control, without gRNA), 'control' (with gRNA for ferret genomic sequence not related to *lzts1*), 'KO#1' and 'KO#2' (each was the mixture of three different gRNAs for ferret *lzts1*). The gRNAs of KO#1 more strongly suppressed *Lzts1* expression than those of KO#2 (Supplementary Fig.11).

The observed distribution of EGFP⁺ cells suggested that *Lzts1* KO delays cellular migration from the apical surface (Fig. 10d), similar to mice (Fig. 3f). Moreover, *Lzts1* KO (KO#1) significantly (and KO#2, more moderately) reduced the percentage of Hes1⁺ undifferentiated neural progenitor cells among EGFP⁺ cells in the OSZ, which represent the newly generated oRGs (Fig. 10c and d). Based on these results, *Lzts1* positively controls oRG generation in the ferret brain.

Figure legends

Fig.10

b–d) CRISPR/Cas9-induced disruption (KO) of *lzts1* in the ferret brain. hCas9 and the guide RNA for ferret *lzts1* were co-expressed with EGFP by *in vivo* electroporation at E32, and brains were examined at E38. See also Supplementary Fig. 11.

b) *Lzts1* KO retards cellular migration from the apical surface. Distributions of Cas9 only (no gRNA), negative control gRNA or *Lzts1*-gRNA (KO#1) electroporated cells in 10 bins separating the cerebral wall were determined (N=11, 11 and 13 hemispheres, respectively, mean ± s.d., Steel-Dwass test, * $p=0.023$, ** $p=5.5 \times 10^{-4}$).

c and d) *Lzts1* KO (KO#1, and KO#2 more moderately) reduced the percentage of Hes1⁺ cells among the electroporated cells in the OSVZ (we examined a 150- μ m-depth area from the basal side of ISVZ) (N=35, 37, 36 and 32 sections from 16, 20, 18 and 14 hemispheres, respectively; Steel-Dwass test, medians with Q1 and Q3 values are shown). Bar, 150 μ m.

Supplementary Fig.11

CRISPR/Cas9-mediated KO of *Lzts1* in the embryonic ferret brain.

In vivo electroporation of Cas9, gRNA and EGFP-3NLS (EGFP with nuclear localization signals) expression vectors was performed at E32 in ferret embryos, and brains were examined at E38. Four types of experiments were performed, i.e., Cas9 (negative control, without gRNA), control (with the gRNA for a ferret genomic sequence not related to *lzts1*), 'KO#1' and 'KO#2' (each with mixture of 3 different gRNAs for ferret *lzts1*).

a) *Lzts1* and GFP immunostaining of the ISVZ in the E38 brains. Bar, 10 μ m.

b) Since *Lzts1* is mainly localized to the cytoplasm or cell cortex, we examined the anti-*Lzts1* fluorescence signals at the cellular junction (*) where the both neighboring cells were GFP⁺

in ISVZ and compared the values to the neighboring cells (x) to examine the Lzts1 KO efficiency. The gRNAs of KO#1 more strongly suppressed Lzts1 expression than those of KO#2. N=34, 29 and 36 cellular junctions from 3 embryos per experiment, Steel-Dwass test, medians with Q1 and Q3 values are shown.

Methods, page 21

The guide RNAs (gRNAs) against target genes (mouse Lzts1: GCCGTTTTGGGGGTTTCAGCT; ferret Lzts1-KO#1: CCGGGCTTCACGATAACAAGT, TAGTGGGGACATAGGCGGCC, and GGAGAGTTCTCTGCGTACCA; ferret Lzts1-KO#2: GAGCTCAATCGGTATTCAGA, CAAGCTCAGGTCCTACGAGAAG, and CCCACCAGCCGTTTTGGAGGCTC; ferret negative control [intron of *GDF5*: TCCCGGGTTGACGATAAAAAT) were designed using the Zhang lab website (<http://crispr.mit.edu>)⁵⁹. The gRNA fragment was amplified by PCR through self-amplification of the designed primer set and cloned into AflII-digested gRNA vectors (gRNA backbone-YT210)⁶⁰, which were modified from the original vector generated by the Church lab⁶¹. The primers used to construct gRNAs are listed in Supplementary Table 1. A mixture of gRNA vectors, pCAX::hCas9, and pCAG::EGFP(3NLS), pCAG::Strawberry, or pPB-CAG-2EGFP-3NLS with pCAG-hyPBBase, was transfected into the embryonic brain by *in vivo* electroporation³⁶ (Supplementary Table 2).

page 22

...For the *in vivo* electroporation experiments in ferrets⁶⁰ shown in Fig. 10 and Supplementary Fig. 11, E32 pregnant ferrets were anesthetized with isoflurane, the embryonic brain hemispheres were injected with 2 μ l of the DNA solution along with 0.0002% Fast Green FCF, and then the embryos were placed between the paddles of the 10-mm electrodes and subjected to 5 \times 100 ms 45 V electric pulses.

2. Generally speaking, the authors invoke many mechanisms to explain the role of Lzts1 in lamination. Further, the paper is a bit disorganized-as it jumps between knockdown and overexpression making the central conclusions difficult to interpret. As a result the paper suffers somewhat from "data dump" without a clear idea of what is most relevant, making it overly speculative. Figure 9 attempts to bring all of this together but is itself extremely confusing, again perhaps due to inclusion of multiple datasets. Some of this may be related to the over-reliance on GOF studies, which as I suggest in point 1 may not be physiologically relevant. The manuscript would be more impactful and stronger if the authors carefully edit experiments (perhaps a sole focus on lamination), and consider reorganizing the paper to

focus on knockdown and overexpression separately, with further focus on a central mechanism.

We sincerely appreciate the reviewer's insightful comments on the framework of our manuscript. According to the reviewer's suggestion, we have extensively edited our manuscript to present our results fluently, which is now composed of two major stories (one describing neuronal delamination, Figs 2 to 7; and the other describing oRG generation, Figs. 8 to 10) with a conclusion panel for each story (Fig. 7i and Fig. 10e).

Furthermore, we have revised the figures to present the data from the loss-of-function and gain-of-function studies separately. In the revised manuscript, the loss-of-function studies are shown in Figs. 2, 3, 4 9, and 10, while the gain-of-function studies are shown in Figs. 5 to 8.

3. Figures 2, 3, and 8 include loss of function data which show that Lzts1 controls: more basal location of newborn cells (figs 2 and 3) and spindle orientation (fig 8). Generally these data support their conclusion that Lzts1 controls delamination. However they need to be better fleshed out.

A) Figure 2: Data mentioned in lines 155-156: the authors need to show the data that EGFP+ cells in the VZ after overexpression of Neurog1/2 together with Lzts1 shRNA-knockdown are Neurog2+, Sox2- and Pax6-. They should also use a different marker (such as NeuroD2 or Tbr1) to assess differentiated fates (as Neurog2 was actually overexpressed in that experiment). They also need to show that the total number of GFP+ cells are not affected in these experiments and given that Lzts1 is expressed in neurons, that there is no change in neuron generation in Lzts1 loss of function - this is only shown for overexpression currently.

We agree with the reviewer's point that we should examine the effects of Lzts1 LOF on the differentiation of the cells.

First, we have confirmed Tbr2, a marker for the neuronally differentiating cells (Figure A), was expressed in Lzts1 KD Neurog1/2-overexpressing cells. We have revised our description in the Results.

Figure A: Bar, 50 μ m

Then, we performed additional experiments to examine whether Lzts1 KD affected the proliferation and differentiation of the

electroporated cells. Since the total number of GFP+ cells varied among the samples in the electroporation studies, this marker was inappropriate for use as an indicator of the differentiation status. Therefore, we examined the percentages of cells expressing progenitor/neuronal markers (Pax6 and Tbr2), BrdU incorporation and the mitotic index (MI) of the electroporated cells: MI indicates the percentage of Ki67+ cells among BrdU+ GFP+ cells that received BrdU 20 hrs before fixation and represents the percentage of proliferating daughter cells. We have examined these parameters in the control-siRNA or Lzts1-siRNA electroporated cells, and the results are shown in new Fig. 3a-d, overall suggesting that Lzts1 KD does not primarily affect the differentiation status of the cells.

Therefore, we have revised the text and Figure legends as follows:

Result, page 6

...These EGFP+ VZ cells were Tbr2+, Sox2⁻ and Pax6⁻ (RG markers)^{34, 35}, indicating that Lzts1 KD did not inhibit neurogenin-induced differentiation.

Page 6

In separate experiments, Lzts1 KD did not significantly alter the percentage of Pax6+ cells (Fig. 3a), BrdU incorporation after 30 min of labeling (Fig. 3b), Tbr2+ cells (Fig. 3c) or the mitotic index in the electroporated cells (Fig. 3d), suggesting that the loss of *Lzts1* did not primarily affect cellular differentiation.

Figure 3

a-d) Lzts1-siRNA#1 or a negative control siRNA was *in vivo* electroporated at E13, and sections of the E14 (c) or E15 (a, b, and d) brain were examined using IHC with anti-Pax6 (a), anti-BrdU (30-min BrdU pulse labeling) (b), anti-Tbr2 (c), anti-Ki67 and anti-BrdU (d) antibodies. The mitotic index (MI) (d) indicates the percentage of Ki67+ cells among BrdU+ GFP+ cells that received BrdU 20 h before fixation. Means ± s.d., (a) N=9, 9; (b) N=10, 9; (d) N=8, 6; (d) N=9, 9 sections from 3 embryos per experiment, Wilcoxon rank sum test.

B) The graph of GFP distribution is difficult to interpret relative to the figure. Why not represent in bins as in Figure 3B?

We appreciate the comment on this point. We have added figures to show the distribution of the cells in bins (Fig 2b). Since we feel that the inclusion of figures to show multiple comparisons of the number of cells in each bin among the seven experimental conditions would be somewhat busy and unclear, we extracted the data of '%cells within 0-60 μm from the apical surface' (orange-colored in Fig 2b) and performed the multiple comparison test in

Fig 2c.

The figure legend was revised, as described in our response to minor point (3) from Reviewer #2.

C) For Figure 3B, the result showing more apically positioned cells in the Lzts1-KO is not convincing and lacks error bars and statistical tests.

We agree with this point. To show the distribution of GFP+ control and KO cells, we revised the figures to use 10 bins with error bars and described the statistical tests (new Fig 3f in mouse and Fig 10d in ferret).

D) The authors need to include siRNA rescue in the absence of Ngn2 (currently only shown in presence of Ngn2 overexpression).

We agree with the reviewer's point and performed the siRNA rescue experiments in Fig. 3i and Fig. 9g. We revised the text as follows:

Result, page 7

Moreover, compared with the negative control, both Lzts1 KD and Lzts1 KO increased the number of apically positioned Tbr2+ EGFP+ cells, and this KD-induced phenotype was rescued by the expression of siRNA-resistant Lzts1, but not wild-type Lzts1 (Fig. 3g-i).

Result, page14

Then, we examined the effect of the loss of function of Lzts1 on the number of newly generated oRG-like cells in the SVZ. We electroporated Lzts1-siRNA#1 or NC-siRNA along with an EGFP-3NLS expression vector into the E13 or E14 cerebrum. After 2 days, the percentage of Sox2+ cells among EGFP+ SVZ cells was significantly reduced by Lzts1 KD in animals electroporated at E14 ($p=0.0091$) (Fig. 9g-i). More Sox2+ GFP+ SVZ cells were observed at E14→E16 than at E13→E15, consistent with previous reports of the temporal pattern of oRG distribution⁷. The observed decrease induced by Lzts1-siRNA#1 at E14→E16 was rescued by the siRNA-resistant mutant form of Lzts1, but not wild-type Lzts1, further supporting the hypothesis that this decrease was induced by the loss of Lzts1 expression.

4. The authors state that Lzts1 functions to "enhance actomyosin system" but it is not clear what they mean by this, as it relies on experiments with blebbistatin (showing incomplete rescue). Figure 6c uses an atomic force microscope to measure stiffness in fibroblasts, but I am not sure how much this adds- is this outcome really relevant in a tissue context where

there are vastly different pulling forces on the cell? There is also heavy speculation as to how Lzts1 intersects with microtubules but this cytoskeletal component is never evaluated formally. If acto-myosin activation is important, additional experiments could be included to assess activation of myosin light chain kinase and other differences in contraction via phosphorylation events and perhaps fret? Can they include some experiments to support the interesting idea that Lzts1 mediates these activities are via interface with microtubules?

We appreciate the Reviewer's thoughtful comment.

We agree that an experiment designed to show the activation of myosin light chain (MLC) kinase by Lzts1 *in vivo* would be better. The requirement for myosin activity at the apical contraction ('apical abscission') has been shown in chick neuroepithelial cells by using live imaging of the localization of the active form of MRLC2 (myosin regulatory light chain 2), inhibitor experiments with blebbistatin and ML-7, and rescue experiments using the active form of MRLC2 (Das and Storey, 2014). We have tried to detect the pMLC immunoreactivity at the apical contraction of the neuronally differentiating cells and the Lzts1-overexpressing cells, but we were unable to observe enhanced staining in these cells by using either *en face* observations of brain tissues or ICC of the primary cultured neural cells. We postulated that the discrepancy from the previous study was the technical difficulty in immunostaining the phosphorylated form of the protein and the possibility that a small additive activation of MLC may be sufficient to evoke the apical contraction.

Since we already collected AFM data from NIH3T3 cells, we next examined pMLC immunoreactivity in NIH3T3 cells, and the results suggested that these cells show a relatively lower level of endogenous pMLC expression than primary cultured neural cells. Then, forced expression of Lzts1 in NIH3T3 cells induced morphological changes, such as the narrowed shape shown Fig 7g, which was accompanied by increased immunoreactivity for pMLC. Based on these results, at least in NIH3T3 cells, exogenous Lzts1 activates the actomyosin system to change cell morphology.

Regarding the AFM experiments, we have used NIH3T3 cells, not primary cultured neural cells, to minimize the noise based on the cellular heterogeneity in cellular stiffness. AFM measurements were performed using isolated NIH3T3 cells, which had been plated on the dish 30 min to 2 hrs before starting the measurements. At this time point, all cells showed the round shape, and thus the effect of the variation in cell morphology on the data could be ignored.

In tissue culture, the effects of cytoskeletal inhibitors on the vast majority of the cells were not only abnormal cellular movement but also disorganization of the tissue morphology. These environmental changes prevented us from easily assessing the effect of inhibitors on

the target cells. In this sense, AFM measurements of isolated cells provide information about the mechanisms by which Lzts1 and inhibitors alter the properties of the examined cells as a compartment of the tissues. Indeed, in the experiment in which Taxol, which stabilizes the microtubules, was administered and cells were analyzed using AFM, the Taxol-induced increase in stiffness was attenuated by the addition of blebbistatin to the Lzts1-overexpressing cells (new Fig. 7h). Thus, Lzts1 may have a general function to enhance the activity of the actomyosin system, contributing to cellular stiffness or contraction, which does not necessarily require Lzts1-mediated inhibition of microtubule assembly.

Again, these results were obtained from isolated cells. Moreover, we analyzed the Lzts1-overexpressing fibroblasts, not the neural progenitors. Thus, our AFM data from NIH3T3 cells (Fig. 7g and h) only provide general information about the effect of Lzts1 on the cytoskeleton, which does not directly extrapolate to the cellular behaviors in the neural tissues.

Therefore, we agree that further efforts to visualize Lzts1 interacting with microtubules and F-actin, such as EB3-EGFP and F-Actin-RFP (Kasioulis et al., 2017), possibly combined with the FRET technique would be needed to determine how Lzts1 controls cytoskeletal dynamics during delamination and oRG generation. While we believe that these experiments are beyond the scope of this manuscript, this point should be addressed in the future.

Therefore, we have revised our manuscript as follows:

Results, page 11

To further clarify the effect of Lzts1 on the cytoskeletons at the cellular level, we used the NIH3T3 mouse fibroblast cell line to minimize the noise based on the cellular heterogeneity. NIH3T3 cells overexpressing Lzts1 exhibited increased immunoreactivity for phospho-myosin light chain 2 (pMLC) accompanied by a change in morphology to a narrow shape (Fig 7g). Our atomic force microscope (AFM) indentation measurement⁴⁰ indicated that exogenous Lzts1 expression increased the stiffness of the NIH3T3 cells compared with control cells ($p=2.2 \times 10^{-4}$), and blebbistatin antagonized this effect (Fig. 7h). These results suggest that the exogenous Lzts1 activates the actomyosin system in NIH3T3 cells to increase the stiffness of the cells compared with the control cells.

Then, to examine whether the reported inhibitory effect of Lzts1 on microtubule assembly²⁸ was necessary for the Lzts1-mediated actomyosin activation, we added Taxol to the

medium and measured the cellular stiffness by using AFM. The Taxol treatment itself increased cellular stiffness both in the control and the Lzts1-overexpressing NIH3T3 cells. In Lzts1-overexpressing cells, this Taxol-induced increase in stiffness was attenuated by the addition of blebbistatin, suggesting that Lzts1 activates myosin II even in response to microtubule stabilization (Fig. 7h). Therefore, Lzts1 may have a general function to enhance the activity of the actomyosin system, contributing to cellular stiffness or contraction, which might not necessarily require the inhibitory effect of Lzts1 on microtubule assembly.

Taken together with the *in vivo* expression pattern (Fig. 1) and results from the loss-of-function studies (Fig. 2–4), we proposed a model that Lzts1 in neuronally differentiating cells modulates the microtubule-actin-AJ system at the apical endfeet to evoke apical contraction and reduce N-Cadherin expression, both of which ensure the quick delamination from the apical surface (Fig. 7i).

Discussion, page 17

.... We speculate that this modulation would disintegrate AJs of differentiating cells with the downregulation of N-cadherin. Further studies are needed to understand the precise molecular mechanism by which Lzts1 activates the actomyosin system and inhibits the microtubule system.

5. Figure 5 includes live imaging which correlates levels of Lzts1 with different behaviors of RGs-but this is all qualitative. A more comprehensive quantitative analysis of the different behaviors is needed to more clearly assess the effect of Lzts1 expression. Of note there is no wild-type control included here making it difficult to interpret. Again these experiments rely upon overexpression in apical RGs, making the physiological relevance questionable (given low levels of Lzts1 expression in these cells).

As shown in our response to major point 1 from Reviewer #2, we added a figure showing the normal INM in the control experiments (Fig. 6d) and the quantitative analysis of the percentages of cells exhibiting each departure pattern to new Fig. 6e. Furthermore, we have revised the text, including the description on the control cases. The 63 control cases we observed were categorized into the normal departure type, as shown in Fig. 6d (these 63 'normal' cases included the 6 cases in which the short daughter cells divided in the SVZ within our 20-hr imaging time, suggesting that they are IPs). As the Reviewer's mentioned, these studies all employed overexpression, and we agree with the Reviewer's point that the LOF phenotype is more important to confirm the physiological function of Lzts1 in cell departure behaviors.

Minor comments:

1. In Figure 1, do the authors observe any co-localization of Lzts1 with microtubules by EM?

We agree with the reviewer that an ideal experiment would be to examine the co-localization of Lzts1 with microtubules using EM. However, this experiment is technically challenging because strong fixation is needed to observe microtubules using EM, preventing us from performing immuno-EM.

2. For Figure 3A, please show a high mag of the Lzts1 expression in GFP knockout cells to convince the reader that the antibody is specific.

We have added a figure to show the magnified view (new Fig. 3e').

3. In Figure 4a the authors need to clarify why they use Lyn-EGFP and what 1.0 and 0.2 are in the figure.

We appreciate the reviewer's comment. We have added the note of Lyn-EGFP to the new Fig. 5 legend and a panel to show the experimental design as new Fig. 5a.

Result, page 8

Next, we performed an *lzts1* gain-of-function study. We forcibly expressed Lzts1 (at a plasmid concentration of $1.0 \mu\text{g } \mu\text{l}^{-1}$, indicated as 'Lzts1 1.0' in the figures) with Lyn-EGFP, a membrane-targeted EGFP used to visualize cell morphology, under the control of a ubiquitous CAG promoter by performing *in vivo* electroporation at E13 (Fig. 5a).

Fig5. legend

a) Experimental design.

b) Lzts1 overexpression ($1.0 \mu\text{g } \mu\text{l}^{-1}$) promotes the detachment of cells from the VZ surface. Cell morphology was visualized by Cre-mediated Lyn-EGFP (membrane-targeted form of EGFP), which was co-electroporated (2 days after E13 electroporation).

4. Lines 191-194: It is not clear why the authors mention *Scrt1* and what the experiments related to this gene bring to the study. Consider removing this result or be more specific on the rationale for this piece of data.

We appreciate the reviewer's comment on this point. We have added a description of why we examined the relationship between *Scrt1* and Lzts1 to the Results. Since we felt that Fig 4 was crowded, we moved the panels related to *Scrt1* to Supplementary Fig. 5.

Result, page 8

We also examined the effect of *Scrt1* on *Lzts1* expression. *Scrt1* is an EMT-related transcription factor that induces neuronal delamination by downregulating E-cadherin (*Cdh1*) expression¹⁸. Since *Scrt1* expression is upregulated throughout neuronal differentiation, it is possible that *Lzts1* expression is induced by *Scrt1* during neuronal differentiation. To examine this possibility, we overexpressed *Scrt1* in the brain and examined *Lzts1* expression using IHC. As reported in a previous study¹⁸, *Scrt1* overexpression resulted in cells positioned outside the VZ (Supplementary Fig. 5). However, we did not detect increased *Lzts1* immunoreactivity in *Scrt1*-overexpressing cells, suggesting that *Scrt1* does not induce *Lzts1* expression.

5. Figure 4e-g: It seems that a lot of the ectopic progenitor cells are not EGFP+, suggesting a non-cell autonomous role for *Lzts1*. The authors should comment on this result.

We appreciate the reviewer's insightful comment and agree with the point.

We have revised the text as follows:

Result, page 8

In the *Lzts1*-overexpressing brains, we also noticed an increase in the number of EGFP⁻ SVZ progenitors, which included both Sox2⁺ and Tbr2⁺ cells (Fig. 5e–h). These observations suggest that *Lzts1* overexpression exerts a non-cell autonomous effect on the positioning of the cell, possibly through the homophilic cell adhesion among neural progenitor cells, in addition to the possible cell-extrinsic factors that function in regulating the proliferation and maintenance of the surrounding cells.

6. Figure 4E would benefit from graph of bins to support the conclusion that there are more cells in the SVZ/IZ following overexpression.

We have revised this figure to show the distribution of GFP+ cells in bins (new figure, Fig 5d).

7. Figures 6C,G: instead of comparing electroporated cells with the surrounding non electroporated cells, it would be better to use electroporated cells without any overexpression plasmid as a control.

We appreciate the reviewer's comment. We performed additional experiments with the EGFP-electroporated cells as a control and revised these figures (new figure Fig. 7d and Fig. 4b). The new Fig. 7d shows that the overall conclusions are the same as those listed in the

previous version of the manuscript.

The new Fig. 4 summarizes the results from *en face* observations of control or Lzts1 KO cells in the Gadd45g-d4Venus Tg mice. In Fig. 4b, we also increased the sample size of Lzts1 KO cells and performed the multiple comparison test (Steel-Dwass test). The main conclusion was the same as the previous result in which Lzts1 was dispensable for the apical contraction in the differentiating cells: even in the Lzts1 KO cells, the AJ ring length of the differentiating Venus+ cells was significantly smaller than the Venus- cells ($p < 1.0 \times 10^{-7}$). A significant change in the AJ ring length was not observed between control and Lzts1-KO Venus+ cells ($p = 0.76$) (this point was different from the previous result).

To examine whether loss of function of Lzts1 affected the delamination of the differentiating cells, we performed additional analyses of the percentage of Venus+ apical endfeet among the apical endfeet of the electroporated cells. Based on the *en face* observations, Lzts1 KO increased the percentage of the Venus+ apical endfeet compared to the control cases (new Fig. 4c). Because LOF of Lzts1 did not primarily affect the differentiation status of the cells (Fig. 3a-d), this observation suggests that the loss of function of Lzts1 retards the delamination of the apical process of the neuronally differentiating cells. This interpretation is consistent with the results shown in Fig. 2d, in which many of the Lzts1 KD cells retained long apical processes that were attached to the apical surface even following Neurog1/2-induced differentiation.

Therefore, we have revised the Results and Figures as follows:

Results: page 7

Since neuronally differentiating cells are known to delaminate from the apical surface with contraction of their AJ rings at the apical endfeet^{16, 17}, we performed an *en face* observation of the apical endfeet of the Lzts1 KO cells. We expressed a gRNA for Lzts1, hCas9 and RFP (dsRed) into the E13 Gadd45::d4Venus mice, and after two days, we fixed the brain tissues and stained them with an anti-ZO1 antibody to visualize the AJ ring. As shown in Fig. 4a-c, the size of the AJ ring in the differentiating Venus+ cells was significantly smaller than the Venus- cells ($p < 1.0 \times 10^{-7}$), even in the Lzts1 KO cells, while a significant difference in the AJ ring length was not observed between control and Lzts1 KO Venus+ cells ($p = 0.76$) (Fig. 4b). These results indicated that Lzts1 KO was insufficient to inhibit the contraction of the AJ ring in the differentiating cells.

On the other hand, we noticed that Lzts1 KO significantly increased the percentage of Venus+ apical endfeet among the RFP+ endfeet ($p = 0.0072$) (Fig. 4c). Because the loss of

function of *Lzts1* did not increase neuronal differentiation (Fig. 3a–d), this result suggests that the detachment of the apical process from the apical surface is retarded by *Lzts1* KO, consistent with the results of the KD experiments with *Ngn1/2* co-overexpression (Fig. 2d).

Together, these results indicate that *Lzts1* is necessary for appropriate neuronal delamination from the apical surface.

Fig. 4

***Lzts1* is dispensable for apical contraction but ensures rapid delamination.**

a–c) CRISPR/Cas9-mediated *Lzts1* KO was performed by electroporating the plasmids into E13 *Gadd45g::d4Venus Tg* mice *in vivo*, and the apical endfeet were examined at E15 by performing *en face* observations.

a) Arrows indicate the RFP⁺ (electroporated) cells that are Venus⁺ (neuronally differentiating).

b) *Lzts1* KO did not change the distribution of the circumferential length of AJ rings in both Venus⁺ and Venus⁻ cell populations. N=445, 124, 404, and 162 cells from 5 (control) and 6 (KO) embryos, respectively, Steel-Dwass test.

c) *Lzts1* KO increased the percentage of Venus⁺ apical endfeet among RFP⁺ endfeet. N=13 and 13 fields from 6 embryos per experiment, Wilcoxon rank sum test.

Bar, 5 μm. Medians with Q1 and Q3 values are reported.

8. Line 287: The authors should clarify what they mean with "anchoring of chromosomes to the apical portion of the process forming the AJ"

As written in our response to the point (10) from Reviewer #2, we have revised the text as follows:

Result, page 12

The Type C departure, i.e., 'MST from the apical surface' induced by *Lzts1* overexpression (Fig. 6c), indicates that *Lzts1* inhibits the anchoring of centrosomes to the apical portion of the process forming the AJs during M phase. Thus, we speculated that weak expression of *Lzts1* in aRGs might weaken the anchoring machinery of centrosomes and give rise to oblique aRG division, which might mimic the physiological condition of oRG production.

9. Figure 7e: How many brains were analyzed in this experiment?

We examined 3 and 4 brains (embryos) in the control and OE experiments, respectively. We have added this information to the legend (new Fig. 8e).

10. The statement that Ngn2 upregulates Ltz is not supported by data in figure 2, which only support a genetic relationship between the two.

We agree with this point and have revised the text as follows:

Discussion, page 18

... Since *Lzts1* expression is induced by neurogenin, the unified regulation of cell departure, i.e., neuronal delamination and oRG generation, by *Lzts1* ensures the generation of oRGs with neurogenic potential within the time window of neurogenesis^{6, 7, 12}.

11. In figure 8K they should quantify spindle orientation as in Figure 7E.

We agree with this point; however, the 3D measurements of the spindle orientation of ~400 mitotic cells in the *en face*-observed movies require considerable effort and are practically impossible. Thus, we separately examined the spindle orientation in fixed sections of the control and *Lzts1*-KO brains using a 3D measurement. We examined 193 mitotic cells in each condition from 9 control and 10 KO embryos and added the results to Fig. 9e. The difference in the spindle orientation between control and KO cells was significant ($p=0.016$), but small.

We have revised the text and figures as follows:

Results, page 13

To investigate whether *Lzts1* is involved in oRG generation *in vivo*, we examined the effects of the loss of function of *Lzts1* on spindle orientation, 'asymmetric' division of aRGs and the emergence of Sox2⁺ oRG-like cells in SVZ.

First, to examine the spindle orientation, we silenced *lzts1* expression and introduced an H2B-EGFP expression vector to visualize DNA by using *in vivo* electroporation at E13. After 2 days, brains were fixed, stained with anti-ZO1 and anti- γ tubulin antibodies, and the mitotic spindle orientation of aRGs was measured by using a 3D calculation method⁴¹. As the result, we found that *Lzts1* KO significantly decreased the angle (θ) of spindle orientation in the mitotic aRGs ($p=0.016$); however, the difference in the angles between control and KO mitotic cells was small (Fig. 9e).

Therefore, we next examined whether *Lzts1* KO actually affected the frequency of the 'asymmetric' division of aRGs (Fig. 9f).

Figure 9, legend

e) *Lzts1* KO decreases the division angles (θ) of aRGs. N=193 and 193 cells from 10 and 9

embryos, respectively, Brunner-Munzel test, medians with Q1 and Q3 values are shown.

12. General comment: For many results, the authors did not mention the number of samples/animals used for the statistical comparison. Please add those.

We have added a description of the sample size and the numbers of animals used in each experiment to the legends.

Reviewers' comments:

Reviewer #2 (Remarks to the Author):

The authors have done substantial revisions, but I am unsure if they can confidently say that their manipulation results in generation of oRGs or if they are simply inducing premature progenitor delamination when *Lzts1* is overexpressed. My reservations are based on the following observations:

1. They use the marker *Hes1* to identify oRGs in ferret and mouse, but *Hes1* is expressed in both oRGs and aRGs. The use of an oRG-specific panel of markers would be more convincing. While the existence of oRG in rodents is unclear, Pollen et al. found that human oRG markers *TNC* and *PTPRZ1* were both expressed in the mouse in oRG regions but not the apical ones.
2. The interaction between *Hes1* and *Lzts1* is unclear and may cloud interpretation of this study, as the levels of *Hes1* are inversely correlated with *Lzts1* at E14.5 in their screen of transcriptome data in aRGs. However, loss of *Lzts1* decreases *Hes1* expression in the authors' ferret studies. This raises the possibility of a reciprocal interaction between *Hes1* and *Lzts1*, and again raises the point that the identification of oRG should be done by additional markers in this case - that is if for some reason labeling the changes in *Lzts1* overexpressing cortices as increase in bona fide oRGs. This over-stretching in the opinion of this reviewer is unnecessary and distracts the readers who are experts in cortical development.
3. The "Type C" behavior observed in the time lapse movies is not the same as the behaviors that are observed in gyrencephalic species when oRGs are generated, and may be indicative of a mislocalization of aRGs during mitosis since the apical membrane is maintained.
4. There is no evidence provided to indicate that there is an expansion of cortical neurons upon overexpression of *Lzts1*. Longer timepoints to determine if neurogenesis (particularly layers 2/3) are affected by *Lzts1* overexpression would be of interest to support their conclusion that they are generating neuron-producing oRG rather than mislocalizing aRG.

Minor points:

Line 127: A brief justification for why *Gadd45g:d4Venus+* expression helps to identify nascent differentiating cells would be instructive for the reader (i.e. *Gadd45g*, a negative cell cycle regulator and pro-neural gene...)

Lines 151-170: The authors use multiple abbreviations for *Neurog1/2* and should make it more consistent throughout this paragraph.

Lines 195-205: The authors do not indicate the conditions for the control experiments (what plasmids were used in the control experiments?). Further, the description of results in lines 200-205 is confusingly written and the summary of results should not be a double negative (insufficient to inhibit). The paragraph should be rewritten to clearly indicate that the contraction of AJ rings in cells that are differentiating is not inhibited in the absence of *Lzts1*.

Lines 246-255: The justification for the *Scrt1* experiments is unclear. Is *Scrt1* upregulated on a similar time scale as *Lzts1*? Or were the authors also interested in *Scrt1* because it downregulates a component of AJs?

Lines 364-370: The authors state "The Type C departure...indicates that *Lzts1* inhibits the anchoring of centrosomes to the apical portion of the process forming the AJs during M phase" but they do not investigate centrosomes specifically, I would recommend rephrasing this paragraph to reflect that this is a hypothesis, not a demonstrated phenomenon.

Lines 420-427: The authors should consistently use either “oblique” or “asymmetric” to define their analysis. Given that this is a controversial concept in the field, I suggest sticking to observational conclusions rather than interpretive ones with little evidence.

Lies 438-445: Related to the last comment, since the authors did not track the progeny of divisions that were asymmetric or symmetric, they can only say that changes to division angle correlate with *Lzts1* levels, as it is possible the fate of the progeny may be decided before spindle angle.

Lines 452-454: The authors should include in their justification that the ferret brain is a better representation of oRG generation and therefore is better suited to studying the role of *Lzts1* in oRG generation.

Line 580: The title “*Lzts1* function underlies neocortical evolution” is a stretch.

Reviewer #3 (Remarks to the Author):

The authors added a considerable amount of experiments that firm up their initial conclusions and greatly enhance the quality of the study. I just have a few minor concerns which should be addressed.

- Referencing Figure 2, the authors state that “EGFP+ VZ cells were *Tbr2*+, *Sox2*– and *Pax6*–, indicating that *Lzts1* KD did not inhibit neurogenin-induced differentiation.” These data should be shown for both KD and control (to also provide evidence that differentiation is actually induced in this experimental paradigm of *Ngn1,2* overexpression).
- Figure 2a: from this set of images, it actually looks like that LZTS1 levels are lower after overexpression of *Neurog1/2*. Please clarify.
- Figure 2b and associated text. The use of the term “mutant” to describe the siRNA-resistant form of *Lzts1* is confusing. It would be more appropriate to use the term “siRNA-resistant *Lzts1*” in the text and Figure.
- Figures 5i and 6e. The authors need to perform Chi-square analyses with Bonferroni correction to assess the statistical significance of their comparisons.
- Line 306: “examine”. There is probably a typo here, the authors should have used the past tense to be consistent with the rest of the manuscript.
- Line 502: “Thus, neuronal delamination and oRG generation, the two major cell behaviors for departure from the apical surface (Fig. 1a), are unified because they are actually two aspects of the same, continuously variable cellular dynamics controlled by *Lzts1*.” This is a very strong assertion that deserves to be slightly toned down. The authors could introduce this sentence by “Altogether our data supports the hypothesis by which...”
- It would be helpful to add in dotted lines into the new high mag images of Figure 3e, and Fig. S11, to outline GFP+ cells. This may better convince the reader that there is depletion of *Lzts1* expression in knockout cells, which is currently difficult to see.
- The data in Figure 10b depict a very minor but significant increase in apically located *celsl1* but there is no difference represented in other bins. Which bins were used to quantify data in Figure 10d.

25th Mar 2019

Dear Reviewers,

We wish to express our appreciation for the additional instructive comments to improve the manuscript. Our point-by-point responses are provided below.

According to the manuscript style format, we changed the title to avoid using an abbreviation; the new title is 'Lzts1 controls both neuronal delamination and outer radial glial cell generation during mammalian cerebral development.' We have also revised the subheadings and legends to fit the length limitation.

Sincerely yours,
Ayano Kawaguchi

RESPONSES

Reviewer #2 (Remarks to the Author):

The authors have done substantial revisions, but I am unsure if they can confidently say that their manipulation results in generation of oRGs or if they are simply inducing premature progenitor delamination when Lzts1 is overexpressed. My reservations are based on the following observations:

1. They use the marker Hes1 to identify oRGs in ferret and mouse, but Hes1 is expressed in both oRGs and aRGs. The use of an oRG-specific panel of markers would be more convincing. While the existence of oRG in rodents is unclear, Pollen et al. found that human oRG markers TNC and PTPRZ1 were both expressed in the mouse in oRG regions but not the apical ones.

We understand the reviewer's point that we would ideally examine oRGs using oRG-specific markers. However, unfortunately, the mentioned human oRG-specific markers Hopx, Tnc and Ptpz1 (Pollen *et al.*, 2015)¹ are not oRG-specific in the mouse and ferret brain; they are all expressed in aRGs, at least during the developmental stages we examined. To clarify this point, we have presented additional information in a new Supplementary Fig. 13.

Before explaining our results, we would like to suggest that the reviewer may have misunderstood Pollen's paper. This previous paper (Pollen et al., Cell, 2015)¹ showed that *Hopx*, *Tnc* and *PTPRZ1* mRNAs and proteins are specifically expressed in human oRGs

and can be used as human oRG markers. In the mouse brain, however, Pollen *et al.* showed that these mRNAs and proteins are expressed in radial glia in both the VZ and SVZ (Fig. 7 and Supplementary Fig. 7 in Pollen *et al.*, 2015¹; Fig. 7 shows the TNC and PTPRZ1 immunoreactivity in the mouse VZ, and Supplementary Fig. 7 ISH shows the *Hopx*, *Ptprz1*, and *Tnc* mRNA expression in the mouse VZ). Therefore, these human oRG-specific markers are not specific for oRGs in the mouse brain as they are also expressed in aRGs.

Therefore, we examined whether these human oRG-specific markers are specific for oRGs in the ferret brain. We examined the expression patterns of these molecules by IHC and ISH and have added our results to a new Supplemental Fig. 13. As shown in the figure, *Hopx*, *Tnc* (mRNA), and *Ptprz1* are expressed in the ferret VZ, suggesting that these molecules are also expressed in aRGs. The anti-*Tnc* immunoreactivity was strongly observed in layer I, similar to the immunoreactivity in the rat brain (Malvarez-Dolado *et al.*, 1998)²; however, we did not observe clear staining in the VZ/SVZ, although we used the same anti-TNC antibody as the one used by Pollen *et al.* in mice.

We appreciate the reviewer's comment and agree with the importance of clarifying this point. We have added these data in Supplementary Fig. 13a and d-f.

Supplementary Figure 13 (related to Figure 10)

The human oRG-related molecules *Hopx*, *Tnc*, and *Ptprz1*¹ are not oRG-specific in ferret brains.

a) Anti-Hes1 and anti-Hopx IHC of the ferret E38 cerebral wall. In addition to some OSVZ cells, VZ cells are also immunoreactive for the anti-Hopx antibody, indicating that aRGs also expressed Hopx. Note that weak and broad anti-Hopx immunoreactivity was observed in the ISVZ, which contains many neuronally differentiating cells.

d, e) *Tnc* mRNA ISH (d) and anti-*Tnc* IHC (e) in the ferret brain. Strong immunoreactivity was observed in layer I (asterisk in [e]).

f) Anti-*Ptprz1* IHC of the ferret brain. Immunoreactivity of the VZ suggests that the aRGs express *Ptprz1*. Bars, 100 μ m in (a, d-f) and 30 μ m in (b).

2. The interaction between *Hes1* and *Lzts1* is unclear and may cloud interpretation of this study, as the levels of *Hes1* are inversely correlated with *Lzts1* at E14.5 in their screen of transcriptome data in aRGs. However, loss of *Lzts1* decreases *Hes1* expression in the authors' ferret studies. This raises the possibility of a reciprocal interaction between *Hes1*

and *Lzts1*, and again raises the point that the identification of oRG should be done by additional markers in this case - that is if for some reason labeling the changes in *Lzts1* overexpressing cortices as increase in bonafide oRGs. This over-stretching in the opinion of this reviewer is unnecessary and distracts the readers who are experts in cortical development.

From our understanding, this comment includes two points, i.e., (1) The concern that *Lzts1* KO may decrease *Hes1* expression levels in oRGs, which might cause underestimation of the number of bonafide oRGs in the ferret KO brain, and (2) the usage of molecular markers other than *Hes1* to identify oRGs.

(1) *Lzts1* KO does not reduce *Hes1* expression in oRGs

This conclusion is mainly supported by the fact that *Lzts1* KO does not reduce %*Hes1*⁺/*Hopx*⁺ cells in the OSVZ in the ferret brain.

Hopx expression in the OSVZ has often been used as a marker of oRGs in humans and mice. In the ferret brain, *Hopx* immunoreactivity is observed in the VZ and in some cells in the OSVZ; however, weak and broad *Hopx* immunoreactivity is also found in the ISVZ (Supplementary Fig. 13a). This ISVZ immunoreactivity suggests that the neuronally differentiating cells are also immunoreactive for the anti-*Hopx* antibody in the E38 ferret brain. On the other hand, *Hes1* immunoreactivity is almost confined to the VZ and some cells in the ISVZ/OSVZ. This expression pattern suggests that *Hes1* immunoreactivity is rapidly downregulated through neuronal differentiation and is exclusively observed in undifferentiated cells. Consistent with these observations, only a portion of *Hopx*⁺ cells (~70%) are *Hes1*⁺ in the OSVZ of the E38 ferret brains (while we confirmed that all *Hes1*⁺*GFP*⁺ OSVZ cells are *Hopx*⁺ both in the Cas9 and KO#1 cases).

Then, if the decrease in the number of *Hes1*⁺oRGs in *Lzts1* KO cases (Fig. 10) is caused by a possible *Lzts1* KO effect that reduces *Hes1* expression in 'bonafide' oRGs, *Lzts1* KO cases would have a lower %*Hes1*⁺/*Hopx*⁺ OSVZ cells than control cases. We examined this frequency (%*Hes1*⁺/*Hopx*⁺) in our samples and confirmed no significant difference between control (Cas9) and KO#1 cases (%*Hes1*⁺/*Hopx*⁺*GFP*⁺ OSVZ cells=26/40=65% and 38/57=67%, respectively, each from 5 hemispheres). These results suggest that *Lzts1* KO does not reduce *Hes1* expression itself in individual oRGs but does reduce the number of newly generated oRGs. Furthermore, because *Hes1* is a transcription factor that maintains the neural progenitor cells in the undifferentiated state, this conclusion is consistent with the fact that gain-of-function and loss-of-function of *Lzts1* do not primarily affect the differentiation status of the electroporated cells in the mouse brain (Fig. 3 and 5).

(2) Hes1 expression in OSVZ is the best currently available oRG marker in the ferret brain

As mentioned above, Hopx is not an oRG-specific marker in the mouse and ferret brain because it is also expressed in aRGs. In the ferret brain, Hopx immunoreactivity is also observed in neuronally differentiating cells. On the other hand, IHC with the anti-Hes1 antibody suggests that Hes1 expression is more confined to undifferentiated neural progenitors (both in oRGs and aRGs). Because Hes1 is a repressor-type bHLH transcription factor and functions to maintain neural progenitor cells in the undifferentiated state (Kageyama et al., 2015)³, Hes1 expression itself guarantees that the cells are undifferentiated. Moreover, as Hes1 is a nuclear protein, its nuclear staining pattern is suitable for the precise and reliable counting of the number of immunoreactive cells, especially when there are many GFP⁺ cells in close proximity to one another (in contrast, Hopx and Ptprz1 immunoreactivity is observed in the cytoplasm). In the mouse brain, other nuclear proteins, Sox2 and Pax6, are often used as markers for undifferentiated cells, but in the E38 ferret brain, a subset of neurons also express these proteins. Therefore, we consider that Hes1 in the OSVZ is currently the best choice as a marker for oRGs in the ferret brain at the developmental stage we examined. Because ferret is a useful animal model to investigate gyrencephalic brain formation, we continue to search for oRG-specific markers available in the ferret brain.

As indicated by the reviewer, our data show that Lzts1⁺ aRGs expressed a low level of *Hes1* mRNA (Fig. 9b), which might somehow confuse the readers. Because this transcriptome profile provides a snapshot of the cells at a certain time point, the low level of *Hes1* mRNA expression in aRGs does NOT mean that the cells do not express Hes1 and does not mean that such cells are differentiating. A certain range of fluctuation of Hes1 expression is reportedly necessary for the proper proliferation of neural progenitor cells (Baek et al., 2006)⁴. Therefore, we have added a description of this point to our manuscript.

We appreciate the reviewer's comment on this issue and believe that these responses and additional information will facilitate the readers' understanding. We have added information on Hes1⁺ oRGs to Supplementary Fig. 13a-c and revised the text as follows:

Supplementary Figure 13 (related to Figure 10)

The human oRG-related molecules Hopx, Tnc, and Ptprz1¹ are not oRG-specific in ferret brains.

a) Anti-Hes1 and anti-Hopx IHC of the ferret E38 cerebral wall. In addition to some OSVZ cells,

VZ cells are also immunoreactive for the anti-Hopx antibody, indicating that aRGs also expressed Hopx. Note that weak and broad anti-Hopx immunoreactivity was observed in the ISVZ, which contains many neuronally differentiating cells.

b) Both in the Cas9 and *Lzts1* KO#1 cases, all *Hes1*⁺GFP⁺ cells in the OSVZ were Hopx⁺ (yellow arrows) (32 and 33 *Hes1*⁺GFP⁺ cells examined from 5 hemispheres).

c) *Lzts1* KO does not decrease %*Hes1*⁺/*Hopx*⁺GFP⁺ OSVZ cells, suggesting that *Lzts1* KO does not reduce *Hes1* expression itself in the individual oRGs; rather, *Lzts1* KO reduces the number of newly generated oRGs (40 and 57 Hopx⁺GFP⁺ OSVZ cells examined, from each 5 hemispheres; two-sided Fisher's exact test) (see also Fig. 10).

Results, Page 13

Because these transcriptome profiles provide a snapshot of the cells at a certain time point, the low level of *hes1* expression in aRGs does not necessarily mean that the cells do not express *Hes1* or are differentiating. Rather, *hes1* has been reported to exhibit oscillatory expression in single aRGs⁴³ and variable expression in the aRG population²⁵. Because *Hes1* represses neurogenin expression, which induces *Lzts1* expression (Fig. 2a), we speculated that the variable *hes1* expression explains the variable *lzts1* expression in the aRG population (discussed below).

Page 16

The observed distribution of EGFP⁺ cells suggests that *Lzts1* KO delays cellular migration from the apical surface (Fig. 10d), similar to that observed in mice (Fig. 3f). Then, we examined whether *Lzts1* KO affects oRG production. In this experiment, we used *Hes1* as a marker of undifferentiated cells. *Lzts1* KO (KO#1) significantly (and KO#2, more moderately) reduced the percentage of *Hes1*⁺ undifferentiated neural progenitor cells among EGFP⁺ cells in the OSZ, which represent the newly generated oRGs (Fig. 10c and d). We also confirmed that these *Hes1*⁺GFP⁺ OSVZ cells were also Hopx⁺ and that *Lzts1* KO did not reduce the expression of *Hes1* in individual oRGs (Supplementary Figure 13). Based on these results, *Lzts1* positively controls oRG generation in the ferret brain.

3. The “Type C” behavior observed in the time lapse movies is not the same as the behaviors that are observed in gyrencephalic species when oRGs are generated, and may be indicative of a mislocalization of aRGs during mitosis since the apical membrane is maintained.

We understand the reviewer's criticism that *if* ‘MST-from-apical surface’ in the human and ferret brain is defined only as a somal departure of VZ progenitors that occurs without maintenance of the apical process (those after or in the process of losing the apical process), our type C MST that occurs while the apical process is maintained should be considered a

different behaviour from the MST of oRG cells in ferrets and humans.

As one approach to clarify this issue, we carefully re-examined our movies, focusing on the retention of the apical processes during MST in Type C. In all cases categorized into the 'MST-from-apical surface' type, the thin apical process was retained during MST. In this experiment, we sparsely labelled aRGs by their expression of both EGFP and Lyn-EGFP, a membrane target form of EGFP, at the same time. Because the apical process during MST in Type C is very thin, both sparse labelling (Cre-loxp system we used) and usage of the membrane-targeted fluorescent protein are necessary to visualize the processes.

Applying this technical limitation that we experienced, i.e., whether such a thin apical process can clearly be recognized with membrane-directed GFP, to the previous paper that reported 'MST-from-apical surface' in the gyrencephalic brains (Gertz et al., 2014)⁵, we can see that it is notable that these authors used a CMV-GFP adenovirus, not a membrane-directed one, to visualize the cell morphology. Therefore, it is currently impossible to determine whether the thin apical processes are maintained (such as in the Type C behaviour) during the 'MST-from-apical surface' behaviour in the human and ferret brain.

Moreover, as we described in response 1 and 2, we cannot currently distinguish mislocalized aRGs and oRGs in our gain-of-function studies because we have no oRG-specific marker in the mouse or ferret brains at the developmental stages examined.

Considering these technical limitations at this point, we believe that simply suggesting a similarity of the Lzts1-induced type C behaviour to the behaviour of oRG cells in ferrets and humans is important for future comparative studies with better spatiotemporal resolutions. Future loss-of-function studies may also clarify whether Lzts1 is also involved in a physiologically rare type of oRG production. 'MST-from-apical surface', in the gyrencephalic brains.

Regarding the visualization of the retained thin apical processes, we have revised our text as follows:

Results, Page 9

To further understand the abnormal cellular positioning after Lzts1 overexpression, we performed live imaging to observe the behaviours of aRG-like cells expressing exogenous Lzts1 (at the concentration 1.0–2.0 $\mu\text{g } \mu\text{l}^{-1}$) with EGFP and Lyn-EGFP (Fig. 6a–c).

This pattern of behaviour resembles that of some aRGs in ferrets and humans that undergo MST to generate an oRG daughter cell following INM to the apical surface during G2²². Although our observations apparently differed from these reported cell behaviours in that the apical daughter cell of division retained the thin apical process for a while, use of a membrane-targeted form of fluorescent protein, such as Lyn-EGFP, may be able to be used to visualize potentially retained apical processes in ferrets/humans.

4. There is no evidence provided to indicate that there is an expansion of cortical neurons upon overexpression of *Lzts1*. Longer timepoints to determine if neurogenesis (particularly layers 2/3) are affected by *Lzts1* overexpression would be of interest to support their conclusion that they are generating neuron-producing oRG rather than mislocalizing aRG.

Lzts1 overexpression in the mouse brain by *in vivo* electroporation did not induce the expansion of the CP or increase in layer 2/3 neurons. We consider this result reasonable for the following reasons.

When the experimental preparation induces SVZ neural progenitor cells (oRG-like cells) from aRGs, layer 2/3 expansion would only be observed if the SVZ neural progenitors are more proliferative and contribute more neuronal production than the physiological aRGs. In this sense, the previously reported genetic manipulation of mouse oRG-like cells to induce cortical expansion and/or folding indeed enhances proliferation of oRG-like cells in the SVZ, such as by downregulating DNA-associated protein *Trnp1* (Stahl et al., 2013)⁶, overexpressing cell cycle regulatory proteins *Cdk4/Cyclin D1* (Nonaka-Kinoshita et al., 2013)⁷, and forcing the expression of primate-specific *TMEM14B* (Liu et al., 2017)⁸, human-specific *TBC1D3* (Ju et al., 2016)⁹ and *ARHGAP11B* (Florio et al., 2015)¹⁰.

However, as previously reported (Wang et al., 2011; Pollen et al., 2015)^{1,11}, the mouse oRG-like cells in mice have more limited potential for proliferation and neuronal production than the oRGs in the gyrencephalic brain. Therefore, the generation of oRG-like cells mediated by the simple repositioning of neural progenitors to the SVZ does not result in the expansion of the CP in the mouse brain (Konno et al., 2008; Shitamukai et al., 2011)^{12,13}. In these papers, perturbing LGN function induced oblique divisions of aRGs to increase the number of SVZ neural progenitors but did not enhance the proliferation of the SVZ progenitors. Similarly, *Lzts1* induces the production of SVZ progenitors from aRGs but does not induce over-proliferation of the SVZ progenitors.

[REDACTED]

Regarding the oRGs in mice, we are uncertain whether the SVZ neural progenitors in the mouse brain should be called oRGs. Considering this discussion point, we have carefully revised our description of mouse SVZ progenitors and used the term 'oRG-like cells' rather than 'oRGs,' including in the subheading of the Results section.

Minor points:

Line 127: A brief justification for why Gadd45g:d4Venus+ expression helps to identify nascent differentiating cells would be instructive for the reader (i.e. Gadd45g, a negative cell cycle regulator and pro-neural gene...)

We appreciate the suggestion. We have revised the text as follows:

Results, page 5

These Lzts1⁺ VZ cells were Tbr2 (Eomes)::EGFP⁺ ¹⁴ (Fig. 1e) and Gadd45g::d4Venus⁺ ¹⁵ (Fig. 1f). The simultaneous expression of both a transcription factor, *tbr2*, and a negative cell- cycle regulator, *gadd45g*, in neuronally differentiating cells (nascent neurons and IPs) ¹⁶ suggests that the Lzts1⁺ VZ cells were nascent differentiating cells.

Lines 151-170: The authors use multiple abbreviations for Neurog1/2 and should make it more consistent throughout this paragraph.

We have corrected this mistake.

Lines 195-205: The authors do not indicate the conditions for the control experiments (what plasmids were used in the control experiments?). Further, the description of results in lines 200-205 is confusingly written and the summary of results should not be a double negative (insufficient to inhibit). The paragraph should be rewritten to clearly indicate that the contraction of AJ rings in cells that are differentiating is not inhibited in the absence of Lzts1.

We agree with this point and have revised as follows:

Results, page 7

Because neuronally differentiating cells are known to delaminate from the apical surface with contraction of their AJ rings at the apical endfeet ^{16, 17}, we performed an *en face* observation of the apical endfeet of the Lzts1 KO cells. We introduced a guide RNA (gRNA) for *lzts1*, hCas9 and RFP (dsRed) into E13 Gadd45::d4Venus mice. As a control experiment, we expressed hCas9 and RFP.

After two days, the brain tissues were fixed and stained with an anti-ZO1 antibody to visualize the AJ ring. As shown in Fig. 4a–c, the size of the AJ ring in the differentiating Venus⁺ cells was significantly smaller than that in the Venus⁻ cells both in the control and the Lzts1 KO cells ($p < 1.0 \times 10^{-7}$ and $p < 1.0 \times 10^{-7}$). Furthermore, no significant difference in the AJ ring length was observed between control and Lzts1 KO Venus⁺ cells ($p = 0.76$) (Fig. 4b). These results suggest that the contraction of the AJ ring in the differentiating cells occurs even in the absence of Lzts1.

Lines 246-255: The justification for the Sct1 experiments is unclear. Is Sct1 upregulated on a similar time scale as Lzts1? Or were the authors also interested in Sct1 because it downregulates a component of AJs?

We performed Sct1 experiments because a previous study (Itoh *et al.*, 2013) showed that (1) Sct1 KD impairs neuronal delamination and that (2) this phenotype suggests that Sct1 is first expressed at an early time point of neuronal differentiation, similar to the timing of Lzts1 expression.

Therefore, we have revised the text in the Results, page 9, as follows:

We also examined the effect of Sct1, an EMT-related transcription factor, on Lzts1 expression. A previous study suggested that Sct1 expression begins at an early time point of neuronal differentiation and promotes apical process detachment¹⁷. Therefore, Sct1 may induce Lzts1 expression to evoke neuronal delamination. To examine this possibility, we overexpressed Sct1 in the brain and examined Lzts1 expression using IHC. As reported¹⁷, Sct1 overexpression resulted in cells positioned outside the VZ (Supplementary Fig. 6).

Lines 364-370: The authors state “The Type C departure...indicates that Lzts1 inhibits the anchoring of centrosomes to the apical portion of the process forming the AJs during M phase” but they do not investigate centrosomes specifically, I would recommend rephrasing this paragraph to reflect that this is a hypothesis, not a demonstrated phenomenon.

We agree with this point and have revised the text as follows:

Results, page 12

The Type C departure, i.e., the MST from the apical surface induced by Lzts1 overexpression (Fig. 6c), suggests that Lzts1 inhibits the anchoring of centrosomes to the apical portion of the process forming the AJs during M phase. If this is the case, weak expression of Lzts1 in aRGs will weaken the anchoring machinery of centrosomes and give rise to oblique aRG division, which may mimic the physiological condition of oRG production.

Lines 420-427: The authors should consistently use either “oblique” or “asymmetric” to

define their analysis. Given that this is a controversial concept in the field, I suggest sticking to observational conclusions rather than interpretive ones with little evidence.

We appreciate this comment. As the reviewer mentioned, the term 'asymmetric division' may be confusing, and we have revised our description.

In this case, we merely meant 'asymmetric division' to describe the 'morphologically asymmetric division' that we observed, in which only one of the progeny inherited the apical membrane after oblique division (previous studies have shown that this type of oblique division predicts oRG-producing division [Shitamukai et al., 2011; LaMonica et al., 2013])^{13, 18}. However, as the reviewer mentioned, the term 'asymmetric division' has often been used in the case of 'asymmetric cell-fate' between the two daughter cells. Therefore, to avoid misunderstanding, we have revised our description of the 'asymmetric' division to read as 'oblique' division in our text and figure.

Results, Page 14

To investigate whether *Lzts1* is involved in the generation of oRG-like cells *in vivo*, we examined the effects of the loss of function of *Lzts1* on spindle orientation, oblique division of aRGs that induces the asymmetrical inheritance of the apical junction, and the emergence of Sox2⁺ oRG-like cells in SVZ.

Therefore, we next examined whether *Lzts1* KO actually affected the frequency of the oblique division of aRGs that induces asymmetrical inheritance of the apical junction by one of the daughter cells¹² (Fig. 9f).

Lies 438-445: Related to the last comment, since the authors did not track the progeny of divisions that were asymmetric or symmetric, they can only say that changes to division angle correlate with *Lzts1* levels, as it is possible the fate of the progeny may be decided before spindle angle.

As described above, to avoid the misunderstanding of the meaning of 'asymmetric,' we have revised our text.

Lines 452-454: The authors should include in their justification that the ferret brain is a better representation of oRG generation and therefore is better suited to studying the role of *Lzts1* in oRG generation.

We appreciate the suggestion. We have revised the text as follows:

Results, Page 15:

Lzts1 is a highly evolutionarily conserved protein from mice to ferrets and humans. Because ferret has a gyrencephalic brain and oRGs are more abundant in ferret than in mouse^{10,11}, the developing ferret brain is a suitable organism to study Lzts1 function in oRG generation (Fig. 10a).

Line 580: The title “Lzts1 function underlies neocortical evolution” is a stretch.

According to the manuscript style format, we deleted the subheadings in the Discussion part.

=====

Reviewer #3 (Remarks to the Author):

The authors added a considerable amount of experiments that firm up their initial conclusions and greatly enhance the quality of the study. I just have a few minor concerns which should be addressed.

- Referencing Figure 2, the authors state that “EGFP+ VZ cells were Tbr2+, Sox2- and Pax6-, indicating that Lzts1 KD did not inhibit neurogenin-induced differentiation.” These data should be shown for both KD and control (to also provide evidence that differentiation is actually induced in this experimental paradigm of Ngn1,2 overexpression).

We agree with this point and have added the data as new Supplementary Fig. 3.

- Figure 2a: from this set of images, it actually looks like that LZTS1 levels are lower after overexpression of Neurog1/2. Please clarify.

To clearly show Neurog1/2 co-expression induces Lzts1 expression, we have revised Fig. 2a. The new Fig. 2a shows the apico-basal expansion of the Lzts1⁺⁺ area (=reduction of the apico-basal width of Lzts1⁺⁺⁺ VZ area) in the Neurog1/2 co-electroporated region compared to the expansion in the neighbouring non-electroporated region.

We have revised the legend of Fig. 2 as follows:

Figure 2

a) Lzts1 expression is upregulated throughout neuronal differentiation following the forced co-expression of neurog1 and neurog2. Neurog1/2 and GFP were co-expressed at E13 by *in vivo* electroporation, and sections were examined after 18 h. Neurog1/2 co-expression expands the apico-basal width of the Lzts1⁺ area and reduces the depth of the VZ (shown by asterisk) compared to that in the non-electroporated region. (a') Magnified view. Note that the Neurog1/2-expressing EGFP⁺ cells are Lzts1⁺. Arrows indicate the GFP⁺Lzts1⁺ apical processes.

- Figure 2b and associated text. The use of the term “mutant” to describe the siRNA-resistant form of Lzts1 is confusing. It would be more appropriate to use the term “siRNA-resistant Lzts1” in the text and Figure.

We corrected the term in the text (pages 1 and 20), Fig. 1 and 3 and Supplementary Fig. 2.

- Figures 5i and 6e. The authors need to perform Chi-square analyses with Bonferroni correction to assess the statistical significance of their comparisons.

We appreciate the comment. In these cases, the data show a highly unequal distribution (Fig. 5i), and the sample sizes are relatively small (Fig. 6e); therefore, we performed Fisher’s exact test with Bonferroni-adjusted *P* values.

- Line 306: “examine”. There is probably a typo here, the authors should have used the past tense to be consistent with the rest of the manuscript.

We have corrected this typo.

- Line 502: “Thus, neuronal delamination and oRG generation, the two major cell behaviours for departure from the apical surface (Fig. 1a), are unified because they are actually two aspects of the same, continuously variable cellular dynamics controlled by Lzts1.” This is a very strong assertion that deserves to be slightly toned down. The authors could introduce this sentence by “Altogether our data supports the hypothesis by which...”

We agree with this comment and have revised this part. As there is a similar description in the Introduction, we have changed it, too.

Introduction, Page 4:

Here, we report that Lzts1 positively controls not only neuronal delamination but also oRG generation in an expression level-dependent manner. Our findings support the hypothesis that these two different events are both aspects of the same process, continuously varying cellular dynamics controlled by Lzts1. Therefore, we propose that Lzts1 functions as a master modulator of the cytoskeleton, including both the actomyosin system and microtubules, to produce diverse cell behaviours in the cell departure processes.

Discussion, Page 16

Altogether, our data support the hypothesis that neuronal delamination and oRG generation, the two major cell behaviours for departure from the apical surface (Fig. 1a), are two aspects of the same process, continuously variable cellular dynamics controlled by Lzts1 (Supplementary Fig. 16).

- It would be helpful to add in dotted lines into the new high mag images of Figure 3e, and Fig. S11, to outline GFP+ cells. This may better convince the reader that there is depletion of Lzts1 expression in knockout cells, which is currently difficult to see.

We agree with this point and have revised the figures (Fig. 3e and new Supplementary Fig. 12). In Fig. 3e, outlining all GFP+ cells made seeing the Lzts1 image difficult; therefore, as examples, we marked some GFP+ cells to show the Lzts1⁻ cell processes.

- The data in Figure 10b depict a very minor but significant increase in apically located cells but there is no difference represented in other bins. Which bins were used to quantify data in Figure 10d.

For Fig. 10d, we examined the GFP+ cells in a 150- μ m-depth area from the basal side of the ISVZ. This region approximately corresponds to Bin 8 and the apical half of Bin 7. We have added this information to the Fig. 10 legend.

Fig. 10 legend:

Lzts1 KO (KO#1, and KO#2 more moderately) reduced the percentage of Hes1⁺ cells among the electroporated cells in the OSVZ (we examined a 150- μ m-depth area from the basal side of the ISVZ, approximately corresponding to Bin 8 and the apical half of Bin 7).

REFERENCES

1. Pollen, A.A., *et al.* Molecular Identity of Human Outer Radial Glia during Cortical Development. *Cell* **163**, 55-67 (2015).
2. Alvarez-Dolado, M., González-Sancho, J.M., Bernal, J. & Muñoz, A. Developmental expression of the tenascin-C is altered by hypothyroidism in the rat brain. *Neuroscience* **84**, 309-322 (1998).
3. Kageyama, R., Shimojo, H. & Imayoshi, I. Dynamic expression and roles of Hes factors in neural development. *Cell Tissue Res* **359**, 125-133 (2015).
4. Baek, J.H., Hatakeyama, J., Sakamoto, S., Ohtsuka, T. & Kageyama, R. Persistent and high levels of Hes1 expression regulate boundary formation in the developing central nervous system. *Development* **133**, 2467-2476 (2006).
5. Gertz, C.C., Lui, J.H., LaMonica, B.E., Wang, X. & Kriegstein, A.R. Diverse behaviors of outer radial glia in developing ferret and human cortex. *J Neurosci* **34**, 2559-2570 (2014).
6. Stahl, R., *et al.* Trnp1 regulates expansion and folding of the mammalian cerebral cortex by

control of radial glial fate. *Cell* **153**, 535-549 (2013).

7. Nonaka-Kinoshita, M., *et al.* Regulation of cerebral cortex size and folding by expansion of basal progenitors. *EMBO J* **32**, 1817-1828 (2013).
8. Liu, J., *et al.* The Primate-Specific Gene TMEM14B Marks Outer Radial Glia Cells and Promotes Cortical Expansion and Folding. *Cell Stem Cell* **21**, 635-649.e638 (2017).
9. Ju, X.C., *et al.* The hominoid-specific gene TBC1D3 promotes generation of basal neural progenitors and induces cortical folding in mice. *Elife* **5** (2016).
10. Florio, M., *et al.* Human-specific gene ARHGAP11B promotes basal progenitor amplification and neocortex expansion. *Science* **347**, 1465-1470 (2015).
11. Wang, X., Tsai, J.W., LaMonica, B. & Kriegstein, A.R. A new subtype of progenitor cell in the mouse embryonic neocortex. *Nat Neurosci* **14**, 555-561 (2011).
12. Konno, D., *et al.* Neuroepithelial progenitors undergo LGN-dependent planar divisions to maintain self-renewability during mammalian neurogenesis. *Nat Cell Biol* **10**, 93-101 (2008).
13. Shitamukai, A., Konno, D. & Matsuzaki, F. Oblique radial glial divisions in the developing mouse neocortex induce self-renewing progenitors outside the germinal zone that resemble primate outer subventricular zone progenitors. *J Neurosci* **31**, 3683-3695 (2011).
14. Gong, S., *et al.* A gene expression atlas of the central nervous system based on bacterial artificial chromosomes. *Nature* **425**, 917-925 (2003).
15. Kawaue, T., *et al.* Neurogenin2-d4Venus and Gadd45g-d4Venus transgenic mice: visualizing mitotic and migratory behaviors of cells committed to the neuronal lineage in the developing mammalian brain. *Dev Growth Differ* **56**, 293-304 (2014).
16. Kawaguchi, A., *et al.* Single-cell gene profiling defines differential progenitor subclasses in mammalian neurogenesis. *Development* **135**, 3113-3124 (2008).
17. Itoh, Y., *et al.* Scratch regulates neuronal migration onset via an epithelial-mesenchymal transition-like mechanism. *Nat Neurosci* **16**, 416-425 (2013).
18. LaMonica, B.E., Lui, J.H., Hansen, D.V. & Kriegstein, A.R. Mitotic spindle orientation predicts outer radial glial cell generation in human neocortex. *Nat Commun* **4**, 1665 (2013).

REVIEWERS' COMMENTS:

Reviewer #2 (Remarks to the Author):

The authors have done an excellent job responding to the past critiques carefully. I still think the claims regarding oRGs is an over stretching of interpretations by the authors and ultimately unnecessary since the global outcome of the effects on this population is not demonstrated in this work. I'll leave the final decision in this regard to the editors.

Reviewer #3 (Remarks to the Author):

The authors have resolved all the issues I had in my previous round of review. Further they have been generally responsive to the concerns of Reviewer 2. This is an interesting study that I favorably recommend for publication.

Response to Reviewer's comments:

REVIEWERS' COMMENTS:

Reviewer #2 (Remarks to the Author):

The authors have done an excellent job responding to the past critiques carefully. I still think the claims regarding oRGs is an over stretching of interpretations by the authors and ultimately unnecessary since the global outcome of the effects on this population is not demonstrated in this work. I'll leave the final decision in this regard to the editors.

We understand the reviewer's concern and have revised the main title as suggested by the editors.

Reviewer #3 (Remarks to the Author):

The authors have resolved all the issues I had in my previous round of review. Further they have been generally responsive to the concerns of Reviewer 2. This is an interesting study that I favorably recommend for publication.